# Fairness Through Matching

**Kunwoong Kim**  *kwkim.online@gmail.com*
*Department of Statistics*
*Seoul National University*

**Insung Kong**[*]  *insung.kong@utwente.nl*
*Department of Applied Mathematics*
*University of Twente*

**Jongjin Lee**  *jjlee__.lee@samsung.com*
*Samsung Research*

**Minwoo Chae**  *mchae@postech.ac.kr*
*Department of Industrial and Management Engineering*
*Pohang University of Science and Technology (POSTECH)*

**Sangchul Park**  *parks@snu.ac.kr*
*School of Law*
*Seoul National University*

**Yongdai Kim**[†]  *ydkim0903@gmail.com*
*Department of Statistics*
*Interdisciplinary Program in Artificial Intelligence*
*Seoul National University*

**Reviewed on OpenReview:** *https://openreview.net/forum?id=dHljjaNHh1*

## Abstract

Group fairness requires that different protected groups, characterized by a given sensitive attribute, receive equal outcomes overall. Typically, the level of group fairness is measured by the statistical gap between predictions from different protected groups. In this study, we reveal an implicit property of existing group fairness measures, which provides an insight into how the group-fair models behave. Then, we develop a new group-fair constraint based on this implicit property to learn group-fair models. To do so, we first introduce a notable theoretical observation: every group-fair model has an implicitly corresponding transport map between the input spaces of each protected group. Based on this observation, we introduce a new group fairness measure termed Matched Demographic Parity (MDP), which quantifies the averaged gap between predictions of two individuals (from different protected groups) matched by a given transport map. Then, we prove that any transport map can be used in MDP to learn group-fair models, and develop a novel algorithm called Fairness Through Matching (FTM)[1], which learns a group-fair model using MDP constraint with an user-specified transport map. We specifically propose two favorable types of transport maps for MDP, based on the optimal transport theory, and discuss their advantages. Experiments reveal that FTM successfully trains group-fair models with certain desirable properties by choosing the transport map accordingly.

---

[1]The source code of FTM is publicly available at *https://github.com/kwkimonline/FTM*.
[*]This research was conducted while the author was affiliated with Seoul National University.
[†]Corresponding author.

# 1 Introduction

Artificial Intelligence (AI) technologies based on machine learning algorithms have become increasingly prevalent as crucial decision-making tools across diverse areas, including credit scoring, criminal risk assessment, and college admissions. However, when observed data contains unfair biases, the resulting trained models may produce discriminatory decisions (Calders et al., 2009; Feldman et al., 2015; Angwin et al., 2016; Barocas & Selbst, 2016; Chouldechova, 2016; Kleinberg et al., 2018; Mehrabi et al., 2019; Zhou et al., 2021). For instance, several cases of unfair preferences favoring specific groups, such as white individuals or males, have been reported (Angwin et al., 2016; Ingold & Soper, 2016; Dua & Graff, 2017). In response, there is a growing trend in non-discrimination laws that calls for the consideration of fair models (Hellman, 2019).

Under this social circumstance, ensuring algorithmic fairness in AI-based decision-making has become a crucial mission. Among several notions of algorithmic fairness, the notion of *group fairness* is the most explored one, which requires that certain statistics of each protected group should be similar. For example, the ratio of positive predictions should be similar across each protected group (Calders et al., 2009; Barocas & Selbst, 2016; Zafar et al., 2017; Donini et al., 2018; Agarwal et al., 2018).

Various algorithms have been proposed to learn models achieving group fairness. Existing methods for group fairness are roughly categorized into three: pre-processing, in-processing and post-processing. Pre-processing approaches (Zemel et al., 2013; Feldman et al., 2015; Webster et al., 2018; Xu et al., 2018; Madras et al., 2018; Creager et al., 2019; Kim et al., 2022a) aim to debias a given dataset, typically by learning fair representations whose distribution is independent of a given sensitive attribute. The debiased data (or fair representation) is then used to learn models. In-processing approaches (Kamishima et al., 2012; Goh et al., 2016; Zafar et al., 2017; Agarwal et al., 2018; Wu et al., 2019; Cotter et al., 2019; Celis et al., 2019; Zafar et al., 2019; Jiang et al., 2020a; Kim et al., 2022b) train models by minimizing a given objective function under a specified group fairness constraint. Post-processing approaches (Kamiran et al., 2012; Hardt et al., 2016b; Fish et al., 2016; Corbett-Davies et al., 2017; Pleiss et al., 2017; Chzhen et al., 2019; Wei et al., 2020; Jiang et al., 2020a) transform given prediction scores, typically provided by an unfair model, to satisfy a certain fairness level.

Most group-fair algorithms correspond to specific group fairness measures, typically defined by explicit quantities such as prediction scores and sensitive attributes. For example, demographic parity (Calders et al., 2009; Feldman et al., 2015; Angwin et al., 2016) considers the gap between two protected groups in terms of the positive prediction ratio.

However, a shortcoming of such group fairness measures is that they only concern statistical disparities without accounting for implicit mechanisms about how a given model achieves group fairness. Consequently, models can achieve high levels of group fairness in very undesirable ways (see Section B of Appendix for an example). Dwork et al. (2012) also noted that focusing solely on group fairness can lead to issues such as subset targeting and the self-fulfilling prophecy. These observations serve as the motivation of this study. Moreover, our empirical investigations on real-world datasets reveal that unfairness on subsets can be observed in group-fair models learned by existing algorithms, which are designed to achieve group fairness merely (see Section 5).

In this paper, we first propose a new group fairness measure that reveals implicit behaviors of group-fair models. Based on the proposed measure, we develop an in-processing algorithm to learn group-fair models with less undesirable properties (e.g., unfairness on subsets), that cannot be explored or controlled by existing fairness measures.

To do so, we begin by introducing a notable mathematical finding: every group-fair model implicitly corresponds to a transport map, which moves the measure of one protected group to another. Building on this observation, we propose a new measure for group fairness called Matched Demographic Parity (MDP), which quantifies the average prediction gap between two individuals from different protected groups matched by a given transport map. We further prove that the reverse of this finding also holds: any transport map can be used in MDP to learn a group-fair model. Consequently, we develop an algorithm called Fairness Through Matching (FTM), designed to learn a group-fair model under a fairness constraint based on MDP with a given transport map. FTM can provide group-fair models having specific desirable properties (e.g., higher fairness on subsets) by selecting a transport map accordingly. Note that FTM is not designed to achieve the

optimal fairness-prediction trade-off, rather, the focus is on mitigating undesirable properties when achieving group fairness (e.g., unfairness on subsets).

In contrast, existing algorithms that focus solely on group fairness may lack such flexibility. In fact, FTM with a transport map that is not carefully selected could result in unreasonable group-fair models, even though group fairness is achieved.

Therefore, the key to effectively using FTM lies in selecting a good transport map. To this end, we propose two specific options for the transport map: one is the optimal transport (OT) map in the input space, and the other is the OT map in the product space of input and output. Each is designed to achieve specific goals. For example, the former achieves higher fairness levels on subsets, while the latter yields better prediction performance and attains higher levels of equalized odds compared to the former. Experiments on real benchmark datasets support our theoretical findings, showing that FTM successfully learns group-fair models with more desirable properties, than those learned by existing algorithms for group fairness.

**Main contributions**

1. We introduce a notable mathematical observation: every group-fair model has a corresponding implicit transport map. Building on this finding, we present a novel measure of group fairness called **Matched Demographic Parity** (MDP).

2. We prove that any transport map can be used in MDP to learn group-fair models. Subsequently, we devise a novel algorithm called **Fairness Through Matching** (FTM), designed to find a group-fair model using a constraint based on MDP with a given transport map. We propose two favorable transport maps tailored for specific purposes.

3. Experiments on benchmark datasets illustrate that FTM successfully learns group-fair models. Furthermore, we examine the benefits of the two proposed transport maps in learning more desirable group-fair models, compared to those learned by existing algorithms that focus solely on achieving group fairness.

## 2 Preliminaries

### 2.1 Notations & Problem setting

In this section, we outline the mathematical notations used throughout this paper. We focus on binary classification task in this study. We denote $\mathbf{X} \in \mathcal{X} \subset \mathbb{R}^d$ and $Y \in \mathcal{Y} = \{0, 1\}$ as the $d$-dimensional input vector and the binary label, respectively. Whenever necessary, we split $\mathcal{X}$, i.e., the domain of $\mathbf{X}$, with respect to $S$ and write $\mathcal{X}_s$ as the domain of $\mathbf{X}|S = s$. For given $s \in \{0, 1\}$, we let $s' = 1 - s$. Assuming the pre-defined sensitive attribute is binary, we denote $S \in \{0, 1\}$ as the binary sensitive attribute. For a given $s \in \{0, 1\}$, the realization of $S$, we write $s' = 1 - s$. For the probability distributions of these variables, let $\mathcal{P}$ and $\mathcal{P}_{\mathbf{X}}$ represent the joint distribution of $(\mathbf{X}, Y, S)$ and the marginal distribution of $\mathbf{X}$, respectively. Furthermore, let $\mathcal{P}_s = \mathcal{P}_{\mathbf{X}|S=s}, s \in \{0, 1\}$ be the conditional distributions of $\mathbf{X}$ given $S = s$. We write $\mathbb{E}$ and $\mathbb{E}_s$ as the corresponding expectations of $\mathcal{P}$ and $\mathcal{P}_s$, respectively. For observed data, denote $\mathcal{D} = \{(\mathbf{x}_i, y_i, s_i)\}_{i=1}^n$ as the training dataset, consisting of $n$ independent copies of the random tuple $(\mathbf{X}, Y, S) \sim \mathcal{P}$. Let $\|\cdot\|_p$ denote the $L_p$ norm. We also write the $L_2$ norm by $\|\cdot\|$, i.e., $\|\cdot\| = \|\cdot\|_2$ for simplicity.

We denote the prediction model as $f = f(\cdot, s), s \in \{0, 1\}$, which is an estimator of $\mathcal{P}_s(Y = 1|\mathbf{X} = \cdot)$, belonging to a given hypothesis class $\mathcal{F} \subset \{f : \mathcal{X} \times \{0, 1\} \to [0, 1]\}$. Note that the output of $f$ is restricted to the interval $[0, 1]$, making $f$ bounded. For simplicity, we sometimes write $f(\cdot, s) = f_s(\cdot)$ whenever necessary. For given $f$ and $s \in \{0, 1\}$, we denote $\mathcal{P}_{f_s}$ as the conditional distribution of $f(\mathbf{X}, s)$ given $S = s$. Furthermore, let $C_{f_s} := \mathbb{I}(f(\cdot, s) \geq \tau)$ be the classification rule based on $f$, where $\tau$ is a specific threshold (typically, $\tau = 0.5$).

### 2.2 Measures for group fairness & Definition of group-fair model

In the context of group fairness, various measures have been introduced to quantify the gap between predictions of each protected group. The original measure for Demographic Parity (DP), $\Delta \mathrm{DP}(f) :=$

$|\mathcal{P}(C_{f_0}(\mathbf{X}) = 1|S = 0) - \mathcal{P}(C_{f_1}(\mathbf{X}) = 1|S = 1)|$, has been initially considered in various studies (Calders et al., 2009; Feldman et al., 2015; Donini et al., 2018; Agarwal et al., 2019; Zafar et al., 2019). Its relaxed version, $\Delta\overline{\mathrm{DP}}(f) := |\mathbb{E}(f_0(\mathbf{X}))|S = 0) - \mathbb{E}(f_1(\mathbf{X}))|S = 1)|$, has also been explored widely (Madras et al., 2018; Chuang & Mroueh, 2021; Kim et al., 2022a).

However, $\Delta\mathrm{DP}(f)$ has a limitation as it relies on a specific threshold $\tau$ (Silvia et al., 2020). To overcome this issue, the concept of strong DP (which requires similarity in the distributions of prediction values of each protected group) along with several measures for quantifying the discrepancy between $\mathcal{P}_{f_0}$ and $\mathcal{P}_{f_1}$, has been considered (Jiang et al., 2020b; Chzhen et al., 2020; Silvia et al., 2020; Barata et al., 2021). $\Delta\mathrm{WDP}(f) := \mathcal{W}(\mathcal{P}_{f_0}, \mathcal{P}_{f_1})$, $\Delta\mathrm{TVDP}(f) := \mathrm{TV}(\mathcal{P}_{f_0}, \mathcal{P}_{f_1})$, and $\Delta\mathrm{KSDP}(f) := \mathrm{KS}(\mathcal{P}_{f_0}, \mathcal{P}_{f_1})$ are the examples of the measure for strong DP. Here, $\mathcal{W}, \mathrm{TV}$, and KS represent the Wasserstein distance, the Total Variation, and the Kolmogorov-Smirnov distance, respectively. For given two probability distributions $\mathcal{Q}_1$ and $\mathcal{Q}_2$, the three distributional distances are defined as follows. The (1-)Wasserstein distance is defined as $\mathcal{W}(\mathcal{Q}_1, \mathcal{Q}_2) := \inf_{\gamma \in \Gamma(\mathcal{Q}_1, \mathcal{Q}_2)} \mathbb{E}_{(x,y) \sim \gamma} \|x - y\|_1$, where $\Gamma(\mathcal{Q}_1, \mathcal{Q}_2)$ is the set of joint probability distributions of marginals $\mathcal{Q}_1$ and $\mathcal{Q}_2$. The Total Variation is defined as $\mathrm{TV}(\mathcal{Q}_1, \mathcal{Q}_2) := \sup_{A \in \mathcal{A}} |\mathcal{Q}_1(A) - \mathcal{Q}_2(A)|$, where $\mathcal{A}$ is the collection of measurable sets. The Kolmogorov-Smirnov distance is defined as $\mathrm{KS}(\mathcal{Q}_1, \mathcal{Q}_2) := \sup_{x \in \mathbb{R}} |F_{\mathcal{Q}_1}(x) - F_{\mathcal{Q}_2}(x)|$, where $F_{\mathcal{Q}}(x) := \mathcal{Q}(\mathbf{X} \le x)$ is the cumulative distribution function of $\mathcal{Q}$.

Let $\Delta = \Delta(\cdot)$ be a given fairness measure. A given model $f$ is said to be *group-fair* (with level $\epsilon$) if it satisfies $\Delta(f) \le \epsilon$. Furthermore, if $\Delta(f) = 0$, we say $f$ is *perfectly group-fair* (with respect to $\Delta$).

### 2.3 Optimal transport

The concept of the Optimal Transport (OT) provides an approach for geometric comparison between two probability measures. For given two probability measures $\mathcal{Q}_1$ and $\mathcal{Q}_2$, a map $\mathbf{T}$ from $\mathrm{Supp}(\mathcal{Q}_1)$ to $\mathrm{Supp}(\mathcal{Q}_2)$ is called a *transport map* from $\mathcal{Q}_1$ to $\mathcal{Q}_2$ if the push-forward measure $\mathbf{T}_{\#}\mathcal{Q}_1$ is equal to $\mathcal{Q}_2$ (Villani, 2008). Here, the push-forward measure is defined by $\mathbf{T}_{\#}\mathcal{Q}_1(A) = \mathcal{Q}_1(\mathbf{T}^{-1}(A))$ for any measurable set $A$. In other words, $\mathbf{T}_{\#}\mathcal{Q}_1$ is the measure of $\mathbf{T}(\mathbf{X}), \mathbf{X} \sim \mathcal{Q}_1$. For given source and target distributions $\mathcal{Q}_1$ and $\mathcal{Q}_2$, the OT map from $\mathcal{Q}_1$ to $\mathcal{Q}_2$ is the optimal one among all transport maps from $\mathcal{Q}_1$ to $\mathcal{Q}_2$. In this context, 'optimal' means minimizing transport cost, such as $L_p$ distance in Euclidean space.

Monge (1781) originally formulates this OT problem: for given source and target distributions $\mathcal{Q}_1, \mathcal{Q}_2$ in $\mathbb{R}^d$ and a cost function $c$ (e.g., $L_2$ distance), the OT map from $\mathcal{Q}_1$ to $\mathcal{Q}_2$ is defined by the solution of $\min_{\mathbf{T}:\mathbf{T}_{\#}\mathcal{Q}_1 = \mathcal{Q}_2} \mathbb{E}_{\mathbf{X} \sim \mathcal{Q}_1}(c(\mathbf{X}, \mathbf{T}(\mathbf{X})))$. If both $\mathcal{Q}_1$ and $\mathcal{Q}_2$ are discrete with the same number of support points, the OT map exists as a one-to-one mapping. For the case when $\mathcal{Q}_1$ and $\mathcal{Q}_2$ are discrete but have different numbers of supports, Kantorovich relaxed the Monge problem by seeking the optimal coupling between two distributions (Kantorovich, 2006). The Kantorovich problem is formulated as $\inf_{\pi \in \Pi(\mathcal{Q}_1, \mathcal{Q}_2)} \mathbb{E}_{\mathbf{X}, \mathbf{Y} \sim \pi}(c(\mathbf{X}, \mathbf{Y}))$, where $\Pi(\mathcal{Q}_1, \mathcal{Q}_2)$ is the set of all joint measures of $\mathcal{Q}_1$ and $\mathcal{Q}_2$. Note that this formulation can be also applied to the case of $\mathcal{Q}_1$ and $\mathcal{Q}_2$ with an identical number of support. See Section C of Appendix for more details about the Kantorovich problem.

Various feasible estimators for the OT map have been developed (Cuturi, 2013; Genevay et al., 2016), and applied to diverse tasks such as domain adaptation (Damodaran et al., 2018; Forrow et al., 2019) and computer vision (Su et al., 2015; Li et al., 2015; Salimans et al., 2018), to name a few.

### 2.4 Related works

**Algorithmic fairness** *Group fairness* is a fairness notion aimed at preventing discriminatory predictions over protected (demographic) groups divided by pre-defined sensitive attributes. Among various notions of group fairness, Demographic Parity (DP) (Calders et al., 2009; Feldman et al., 2015; Agarwal et al., 2019; Jiang et al., 2020b; Chzhen et al., 2020) quantifies the statistical gap in predictions between two different protected groups. Other measures, including Equal opportunity (Eqopp) and Equalized Odds (EO), consider groups conditioned on both the label and the sensitive attribute (Hardt et al., 2016a). Various algorithms have been proposed to learn group-fair model satisfying these group fairness notions (Zafar et al., 2017; Donini et al., 2018; Agarwal et al., 2018; Madras et al., 2018; Zafar et al., 2019; Chuang & Mroueh, 2021; Kim et al., 2022a).

*Individual fairness*, initially introduced by Dwork et al. (2012), is another fairness notion based on the philosophy of treating similar individuals similarly. It is noteworthy that Dwork et al. (2012) highlighted that focusing solely on group fairness is not always a complete answer, pointing out issues such as subset targeting and the self-fulfilling prophecy. Subsequent researches (Yona & Rothblum, 2018; Yurochkin et al., 2020; Yurochkin & Sun, 2021; Petersen et al., 2021) have been studied in both theories and methodologies. However, individual fairness has two bottlenecks: (i) it strongly depends on the choice of the similarity metric, which hinders its practical applicability, and (ii) by itself, it does not ensure group fairness when the distributions of protected groups differ significantly. See Section 3.3 for detailed comparison between individual fairness and our proposed framework.

*Counterfactual fairness*, which requires treating a counterfactual individual similarly to the original individual, has been studied by Kusner et al. (2017); Chiappa & Gillam (2018); von Kügelgen et al. (2022); Nilforoshan et al. (2022). Its application, however, is limited since it requires causal models, which are difficult to obtain only with observed data. Instead of using graphical models to define counterfactuals, this paper suggests using a transport map from $\mathcal{X}_s$ to $\mathcal{X}_{s'}$ having certain desirable properties. See Proposition 4.3 in Section 4.1 for the relationship between counterfactual fairness and our proposed algorithm.

On the other hand, in presence of multiple sensitive attributes, the concept of *subgroup fairness* can be considered (Kearns et al., 2018a;b; Shui et al., 2022; Mehrotra et al., 2022; Molina & Loiseau, 2022; Carvalho et al., 2022), emphasizing the need for models that satisfy fairness for all subgroups defined over the multiple sensitive attributes. In addition, Wachter et al. (2020); Simons et al. (2021); Mougan et al. (2024) have suggested that not only statistically equal outcome but also the notion of *equal treatment*, i.e., treating individuals with equal reasons, should be considered in algorithmic fairness.

**Fair representation learning (FRL)** FRL algorithm aims at searching a fair representation space, in the sense that the conditional distributions of the encoded representation with respect to the sensitive attribute are similar (Zemel et al., 2013). After learning the fair representation, FRL constructs a fair model by applying a supervised learning algorithm to the fair representation space. Initiated by Edwards & Storkey (2016), various FRL algorithms have been developed (Madras et al., 2018; Zhang et al., 2018) motivated by the adversarial-learning technique used in GAN (Goodfellow et al., 2014). See Section 3.3 for the detailed comparison between FRL and our proposed framework.

**Applications of the OT map to algorithmic fairness** Several studies, including Gordaliza et al. (2019); Jiang et al. (2020b); Chzhen et al. (2020); Silvia et al. (2020); Buyl & Bie (2022), have employed the OT map for algorithmic fairness. Gordaliza et al. (2019) introduced a fair representation learning method that aligns inputs from different protected groups using the OT map. Jiang et al. (2020b); Chzhen et al. (2020); Silvia et al. (2020) proposed aligning prediction scores from different protected groups using the OT map or OT-based barycenter. Buyl & Bie (2022) developed a method that projects prediction scores onto a fair space by optimizing the projection through minimizing the transport cost calculated on all pairs of inputs.

Most of these algorithms (e.g., Jiang et al. (2020b); Chzhen et al. (2020); Silvia et al. (2020)) focus on applying the OT map in the prediction space, i.e., aligning two conditional distributions $\mathcal{P}_{f_0}$ and $\mathcal{P}_{f_1}$. These methods fundamentally differ from our proposed approach, which focuses on applying the OT map in the input space. On the other hand, Gordaliza et al. (2019) and our approach are particularly similar in the sense that both apply the OT map on the input space. See the detailed comparison between Gordaliza et al. (2019) and our approach in Section 3.3.

It is worth noting that our proposed algorithm becomes the first to define a group fairness measure based on the transport map in the input space and to propose reasonable choices for the transport map.

## 3 Learning group-fair model through matching

The goal of this section is to explore and specify the correspondence between group-fair models and transport maps. In Section 3.1, we show that *every group-fair model has a corresponding implicit transport map* in the input space that matches two individuals from different protected groups. We then introduce a new fairness measure based on the correspondence. In Section 3.2, we show the reverse, i.e., *any given*

***transport map can be used to learn a group-fair model***, then present our proposed algorithm for learning group-fair models under a fairness constraint based on a given transport map.

### 3.1 Matched Demographic Parity (MDP)

Proposition 3.1 below shows that for a given perfectly group-fair model $f$ (i.e., $\mathcal{P}_{f_0} = \mathcal{P}_{f_1}$ or equivalently $\Delta = 0$), there exists an corresponding transport map in the input space that matches two individuals from different protected groups. Its proof is in Section A of Appendix. Throughout this section, we assume a condition (C): $\mathcal{P}_s, s = 0, 1$, are absolutely continuous with respect to the Lebesgue measure. This regularity condition is assumed to simplify the discussion, since it guarantees the existence of transport maps between two distributions. See Proposition A.1 in Section A of Appendix, which presents a similar result for the case where $\mathcal{P}_s, s = 0, 1$ are discrete. Let $\mathcal{T}_s^{\text{trans}}$ be the set of all transport maps from $\mathcal{P}_s$ to $\mathcal{P}_{s'}$.

**Proposition 3.1** (Fair model $\Rightarrow$ Transport map: perfect fairness case). *For any perfectly group-fair model $f$, i.e., $\mathcal{P}_{f_0} = \mathcal{P}_{f_1}$, there exists a transport map $\mathbf{T}_s \in \mathcal{T}_s^{\text{trans}}$ satisfying $f(\mathbf{X}, s) = f(\mathbf{T}_s(\mathbf{X}), s'), a.e.$*

The key implication of this mathematical proposition is that all perfectly group-fair models are not the same and the differences can be identified by their corresponding transport maps. We can similarly define a transport map corresponding to a given not-perfectly group-fair model (i.e., $\Delta > 0$), by using a novel fairness measure termed **Matched Demographic Parity** (MDP) defined as below.

**Definition 3.2** (Matched Demographic Parity). *For a given model $f \in \mathcal{F}$ and a transport map $\mathbf{T}_s \in \mathcal{T}_s^{\text{trans}}$, the measure for MDP is defined as*

$$\Delta \text{MDP}(f, \mathbf{T}_s) := \mathbb{E}_s \left| f(\mathbf{X}, s) - f(\mathbf{T}_s(\mathbf{X}), s') \right|. \tag{1}$$

The idea behind MDP is that two individuals from different protected groups are matched by $\mathbf{T}_s$, and $\Delta \text{MDP}(f, \mathbf{T}_s)$ quantifies the similarity between the predictions of these two matched individuals. Subsequently, Theorem 3.3 below presents a relaxed version of Proposition 3.1, showing that any group-fair model $f$ has a transport map $\mathbf{T}_s$ such that $\Delta \text{MDP}(f, \mathbf{T}_s)$ is small. Refer to Section A of Appendix for the proof of Theorem 3.3 and see Figure 1 for the illustration of MDP.

**Theorem 3.3** (Fair model $\Rightarrow$ Transport map: relaxed fairness case). *Fix a fairness level $\delta \geq 0$. For any given group-fair model $f$ such that $\Delta \text{TVDP}(f) \leq \delta$, there exists a transport map $\mathbf{T}_s \in \mathcal{T}_s^{\text{trans}}$ satisfying $\Delta \text{MDP}(f, \mathbf{T}_s) \leq 2\delta$.*

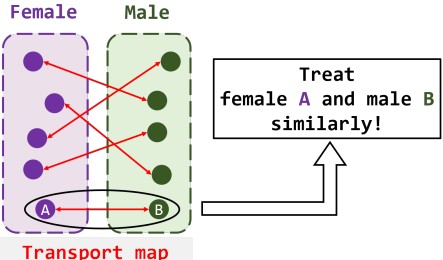

Figure 1: Simplified illustration of MDP. Once two individuals A and B are matched ($\leftrightarrow$), a model treats the pair of matched individuals A and B similarly (as well as all the other pairs). This implicit mechanism contributes to making the model fair.

We define the **fair matching function** for a given $f$ as the transport map that attains the infimum of equation (1), as formulated in Definition 3.4 below. The term 'fair' is used because MDP is minimized by this transport map. Through investigating the fair matching function, we can understand the mechanism behind how a group-fair model behaves. That is, the fair matching function specifically reveals which pairs of individuals from two protected groups are treated similarly by the given group-fair model.

**Definition 3.4** (Fair matching function of $f$). *For a given $f$, denote $\mathbf{T}_s^f := \arg\min_{\mathbf{T}_s \in \mathcal{T}_s^{\text{trans}}} \Delta\text{MDP}(f, \mathbf{T}_s), s \in \{0, 1\}$. For $\hat{s} := \arg\min_{s \in \{0,1\}} \Delta\text{MDP}(f, \mathbf{T}_s^f)$, the fair matching function of $f$ is defined as $\mathbf{T}^f := \mathbf{T}_{\hat{s}}^f$.*

Note that the existence of this fair matching function is guaranteed by Brenier's theorem (Villani, 2008; Hütter & Rigollet, 2021). To be specific, since $\mathcal{P}_s, s \in \{0, 1\}$ are absolutely continuous by (C), and the cost function in MDP, i.e., $c(\mathbf{x}, \mathbf{y}) := |f(\mathbf{x}, s) - f(\mathbf{y}, s')|$, is lower semi-continuous and bounded from below, the minimizer of $\Delta\text{MDP}(f, \mathbf{T}_s)$ uniquely exists.

**Practical computation of the fair matching function** In practice, we estimate the fair matching function using the observed data $\mathcal{D} = \{(\mathbf{x}_i, y_i, s_i)\}_{i=1}^n$. Let $\mathcal{D}_0 = \{\mathbf{x}_i : s_i = 0\} = \{\mathbf{x}_i^{(0)}\}_{i=1}^{n_0}$ and $\mathcal{D}_1 = \{\mathbf{x}_j : s_j = 1\} = \{\mathbf{x}_j^{(1)}\}_{j=1}^{n_1}$ be the set of inputs given the sensitive attribute, where $n_0 + n_1 = n$. We here introduce practical methods for computing the fair matching function in Definition 3.4, by considering two cases:

1. Case of $n_0 = n_1$: When the sizes of two protected groups are equal ($n_0 = n_1$), we can easily find the fair matching function by quantile matching. We first sort the scores in $\{f_s(\mathbf{x})\}_{\mathbf{x} \in \mathcal{D}_s}$ for each group $s \in \{0, 1\}$. Then, we match the individuals having the same rank (i.e., quantile) in each set, thereby obtaining the fair matching function of $f$. This straightforward procedure is theoretically guaranteed by the definition of 1-Wasserstein distance, which is calculated by quantile matching (Rachev & Rüschendorf, 1998; Chzhen et al., 2020; Jiang et al., 2020b). We formally present this procedure in Proposition 3.5 below. Its proof is provided in Section A of Appendix. Let $F_s$ represents the cumulative distribution function of $f_s(\mathbf{x}), \mathbf{x} \in \mathcal{D}_s$. For technical simplicity, we assume that there is no tie in $\{f_s(\mathbf{x}) : \mathbf{x} \in \mathcal{D}_s\}$.

   **Proposition 3.5.** *Let $\mathbf{T}^f$ be the fair matching function of $f$ on $\mathcal{D}_0$ and $\mathcal{D}_1$ (where the empirical distributions with respect to $\mathcal{D}_0$ and $\mathcal{D}_1$ are used in Definition 3.4). Then, the matched individual $\mathbf{T}^f(\mathbf{x})$ of any $\mathbf{x} \in \mathcal{D}_s$ is obtained by $\mathbf{T}^f(\mathbf{x}) = f_{s'}^{-1} \circ F_{s'}^{-1} \circ F_s \circ f_s(\mathbf{x})$.*

2. Case of $n_0 \neq n_1$: When the sizes differ, we can consider a stochastic fair matching function (a stochastic transport map that minimizes $\Delta\text{MDP}$), which matches individuals with probability, not deterministically. In fact, a stochastic transport map corresponds to a joint distribution between two protected groups, i.e., a stochastic transport map can be defined by a joint distribution, as follows:

   Denote $\mathbb{Q}$ as a joint distribution between $\mathcal{D}_0$ and $\mathcal{D}_1$, and let $\mathbf{X}_0$ and $\mathbf{X}_1$ be the random variables following the empirical distributions on $\mathcal{D}_0$ and $\mathcal{D}_1$, respectively. For a given joint distribution $\mathbb{Q}$, the stochastic transport map $\mathbf{T}_s$ (corresponding to the $\mathbb{Q}$) is defined by $\mathbf{T}_s(\mathbf{x}_i^{(s)}) = \mathbf{x}_j^{(s')}$ with probability $\mathbb{Q}(\mathbf{X}_s = \mathbf{x}_i^{(s)}, \mathbf{X}_{s'} = \mathbf{x}_j^{(s')})$. Once we find the stochastic fair matching function, i.e., the stochastic transport map (joint distribution $\mathbb{Q}$) minimizing $\Delta\text{MDP}(f, \mathbb{Q}) = \mathbb{E}_{(\mathbf{X}_0, \mathbf{X}_1) \sim \mathbb{Q}} |f(\mathbf{X}_0, 0) - f(\mathbf{X}_1, 1)|$, we can compute its transport cost as $\mathbb{E}_{(\mathbf{X}_0, \mathbf{X}_1) \sim \mathbb{Q}} \|\mathbf{X}_0 - \mathbf{X}_1\|^2$. Note that the minimization of $\Delta\text{MDP}$ with respect to the stochastic transport map is technically equivalent to solving the Kantorovich problem, which can be easily solved by the use of linear programming. See Section A and Section C for more details about the stochastic transport map and the Kantorovich problem, respectively.

   Alternatively, in practice, we can apply a mini-batch sampling technique. We first sample two random mini-batches $\tilde{\mathcal{D}}_0 \subset \mathcal{D}_0$ and $\tilde{\mathcal{D}}_1 \subset \mathcal{D}_1$ with identical size $m$. Then, we follow the process in '(1) Case of $n_0 = n_1$' above. The transport cost of the fair matching function can be then estimated by the average transport costs computed on many random mini-batches. See Remark A.2 in Section A for the statistical validity of this mini-batch technique.

Furthermore, Remark 3.6 below explains how the transport cost can serve as a metric for assessing the desirability of given group-fair models.

**Remark 3.6** (Usage of the transport cost of the fair matching function). *Furthermore, the transport cost of the fair matching function can serve as a measure to assess whether a given group-fair model is desirable. For example, when choosing a model between two group-fair models with similar levels of group fairness or/and prediction accuracies, the model with the lower transport cost would be preferred. In Section 5.3.2, we compare transport costs of fair matching functions for two different group-fair models, using the mini-batch technique.*

### 3.2 Fairness Through Matching (FTM): learning a group-fair model with a transport map

The goal of this section is to formulate our proposed algorithm. Before introducing it, we provide a theoretical support, which shows that a group-fair model can be constructed by MDP using any transport map. Theorem 3.7 below, which is the reverse of Theorem 3.3, shows that any transport map in the input space can construct a group-fair model. Precisely, for a given transport map, if a model provides similar predictions for two individuals who are matched by the transport map, then it is group-fair. The proof is given in Section A of Appendix.

**Theorem 3.7** (Transport map $\Rightarrow$ Group-fair model). *For a given $\mathbf{T}_s \in \mathcal{T}_s^{\mathrm{trans}}$, if $\Delta\mathrm{MDP}(f, \mathbf{T}_s) \le \delta$, then we have $\Delta\mathrm{WDP}(f) \le \delta$ and $\Delta\overline{\mathrm{DP}}(f) \le \delta$.*

Again, it is remarkable that a group-fair model and its corresponding transport map are closely related, i.e., ***every group-fair model has its corresponding implicit transport map, and vice versa***. This finding can be mathematically expressed as follows. Let $\Delta$ be a given (existing) fairness measure, and define $\mathcal{F}_\Delta(\delta) \coloneqq \{f \in \mathcal{F} : \Delta(f) \le \delta\}$ as the set of group-fair models of level $\delta$ (with respect to $\Delta$). Similarly, for MDP, define $\mathcal{F}_{\Delta\mathrm{MDP}}(\mathbf{T}_s, \delta) \coloneqq \{f \in \mathcal{F} : \Delta\mathrm{MDP}(f, \mathbf{T}_s) \le \delta\}$. Then, following from Theorem 3.3 and 3.7, we can conclude that the three measures (i.e., TVDP, WDP, and MDP) are closely related: $\mathcal{F}_{\Delta\mathrm{TVDP}}(\delta) \subseteq \cup_{\mathbf{T}_s \in \mathcal{T}_s^{\mathrm{trans}}} \{f : \Delta\mathrm{MDP}(f, \mathbf{T}_s) \le 2\delta\} \subseteq \mathcal{F}_{\Delta\mathrm{WDP}}(2\delta)$.

As discussed in Section 1 as well as several previous works (e.g., Dwork et al. (2012)), there exist group-fair models having undesirable properties such as subset targeting or self-fulfilling prophecy. The advantage of MDP is that we can screen out such undesirable group-fair models during the learning phase, to consider desirable group fair models only. And, using MDP achieves this goal by considering group-fair models whose transport maps have low transport costs. That is, we can search for a group-fair model only on $\cup_{\mathbf{T}_s \in \mathcal{T}_s^{\mathrm{good\ trans}}} \{f : \Delta\mathrm{MDP}(f, \mathbf{T}_s) \le 2\delta\}$, where $\mathcal{T}_s^{\mathrm{good\ trans}} \subseteq \mathcal{T}_s^{\mathrm{trans}}$ is a set of specified 'good' transport maps. See Section 4 for such good transport maps that we specifically propose. It is worth noting that such screening would be challenging with existing group-fair measures.

**FTM algorithm**  Based on Theorem 3.7, we develop a learning algorithm named **Fairness Through Matching** (FTM), which learns a group-fair model subject to MDP being small with a given transport map. FTM consists of two steps. First, we select a (good) transport map. Then, we learn a model under the MDP constraint with the selected transport map. The precise objective of FTM is formulated below.

Suppose a transport map $\mathbf{T}_s$ is selected (see Section 4 for the proposed transport maps). FTM solves the following objective for a given loss function $l$ (e.g., cross-entropy) and a pre-defined fairness level $\delta \ge 0$:

$$f^{\mathrm{FTM}}(\mathbf{T}_s) \coloneqq \arg\min_{f \in \mathcal{F}} \mathbb{E}l(Y, f(\mathbf{X}, S)) \text{ subject to } \min_{s \in \{0,1\}} \Delta\mathrm{MDP}(f, \mathbf{T}_s) \le \delta. \tag{2}$$

Unless there is any confusion, we write $f^{\mathrm{FTM}}$ instead of $f^{\mathrm{FTM}}(\mathbf{T}_s)$ for simplicity. By Theorem 3.7, it is clear that $f^{\mathrm{FTM}}$ is fair (i.e., $\Delta\mathrm{WDP}(f^{\mathrm{FTM}}), \Delta\overline{\mathrm{DP}}(f^{\mathrm{FTM}}) \le \delta$) for any transport map $\mathbf{T}_s$. In practice, we estimate $f^{\mathrm{FTM}}$ with observed data $\mathcal{D}$ using mini-batch technique along with a stochastic gradient descent based algorithm (see Section 4 for details).

### 3.3 Conceptual comparison of existing approaches and FTM

**Individual fairness**  FTM and individual fairness are similar in the sense that they try to treat similar individuals similarly. A difference is that FTM aims to treat two individuals from different protected groups similarly, while the individual fairness tries to treat similar individuals similarly regardless of sensitive attribute (even when it is unknown). That is, similar individuals in FTM could be dissimilar in view of individual fairness, especially when the two protected groups are significantly different.

On the other hand, a limitation of individual fairness is that group fairness is not guaranteed. FTM can be also understood as a tool to address this limitation by finding models with higher individual fairness among group-fair models. Empirical results support this conjecture that FTM improves individual fairness compared to baseline methods for group fairness (see Table 4 in Section 5.4.2).

**Fair representation learning** FTM and FRL (Fair Representation Learning) are similar in the sense that they align two conditional distributions. However, the role of the alignment in FTM and FRL are different: FTM builds a prediction model in the original input space, using transport map in the MDP constraint only. In contrast, FRL methods first learn fair representation by aligning the conditional distributions of representation, and then use the learned fair representation as input.

Furthermore, FTM and FRL methods are motivated by fundamentally different objectives. FTM is designed to learn group-fair models with desirable properties (e.g., higher fairness on subsets), whereas FRL methods aim to obtain fair representations which can be used for downstream tasks requiring fairness. The FRL approach proposed by Gordaliza et al. (2019) is particularly similar to FTM, as both apply the OT map on the input space. However, unlike FTM, it trains a prediction model on the (pre-trained) fair representation space derived from the OT map. On the contrary, FTM builds the prediction model on the original input space and utilizes the OT map solely within the fairness constraint. As a result, the prediction model of FTM is not tied to the OT map, making it more flexible and leading to improved prediction accuracy. Empirical evidence supporting this claim is provided in Table 5 in Section 5.4.3, which demonstrates the superior performance of FTM compared to several FRL methods, including Gordaliza et al. (2019).

## 4 Empirical algorithm with transport map selection

The implication of Theorems 3.3 and 3.7 is that any transport map corresponds to a group-fair model. However, an improperly chosen transport map can result in undesirable outcomes, even if group fairness is satisfied, as shown in Theorem 3.7 (see Section B of Appendix for an example of a problematic group-fair model due to an unreasonable transport map). Thus, selecting an appropriate transport map is crucial when using FTM, which is the primary focus of this section.

Specifically, we propose two favorable choices of the transport map $\mathbf{T}_s$ used in FTM. In Section 4.1, we suggest using the OT map in the input space $\mathcal{X}$, resulting in a group-fair model with a higher fairness level on subsets. In Section 4.2, we propose using the OT map in the product space $\mathcal{X} \times \mathcal{Y}$, to improve prediction performance and the level of equalized odds, when compared to the OT map in the input space.

### 4.1 OT map on $\mathcal{X}$

For a given $\mathbf{T}_s \in \mathcal{T}_s^{\text{trans}}$, the *transport cost* (on $\mathcal{X}$) of $\mathbf{T}_s$ is defined by $\mathbb{E}_s \|\mathbf{X} - \mathbf{T}_s(\mathbf{X})\|^2$. First, we propose using the OT map between the two input spaces, which is the minimizer of the transport cost on $\mathcal{X}$ among all transport maps. From now on, we call this OT map on $\mathcal{X}$ as the **marginal OT map**.

In this section, we explore a benefit of using the marginal OT map (i.e., low transport cost) by showing a theoretical relationship between the transport cost and fairness on subsets. Many undesirable behaviors of group-fair models have been recognized and discussed (Dwork et al., 2012; Kearns et al., 2018a; Wachter et al., 2020; Mougan et al., 2024). *Subset fairness*, which is a similar concept to subset targeting in Dwork et al. (2012), is one of such undesirable behaviors. We say that a group-fair model is subset-unfair if it is not group-fair against a certain subset (e.g., aged over 60s) even if it is group-fair overall. The definition of subset fairness can be formulated as follows.

**Definition 4.1** (Subset fairness). *Let $A$ be a subset of $\mathcal{X}$. The level of subset fairness over $A$ is defined as*

$$\Delta\overline{\text{DP}}_A(f) \coloneqq |\mathbb{E}(f(\mathbf{X}, 0)|S = 0, \mathbf{X} \in A) - \mathbb{E}(f(\mathbf{X}, 1)|S = 1, \mathbf{X} \in A)|.$$

Intuitively, we expect that a group-fair model with a low transport cost would exhibit a high level of subset fairness. This is because the chance of two matched individuals (from different protected groups) belonging to the same subset $A$ tends to be higher when the transport cost is smaller. Theorem 4.2 theoretically supports this conjecture, whose proof is provided in Section A of Appendix.

**Theorem 4.2** (Low transport cost benefits subset fairness). *Suppose $\mathcal{F}$ is the collection of $L$-Lipschitz[2] functions. Let $A$ be a given subset in $\mathcal{X}$. Then, for all $f$ satisfying $\Delta\text{MDP}(f, \mathbf{T}_s^f) \le \delta$, we have*

$$\Delta\overline{\text{DP}}_A(f) \le L\left(\mathbb{E}_s\|\mathbf{X} - \mathbf{T}_s^f(\mathbf{X})\|^2\right)^{\frac{1}{2}} + \text{TV}(\mathcal{P}_{0,A}, \mathcal{P}_{1,A}) + U\delta, \tag{3}$$

---

[2]A function $g : \mathcal{X} \to \mathbb{R}$ is $L$-Lipschitz if $|g(\mathbf{x}_1) - g(\mathbf{x}_2)| \le L\|\mathbf{x}_1 - \mathbf{x}_2\|$ for all $\mathbf{x}_1, \mathbf{x}_2 \in \mathcal{X}$.

*where $\mathcal{P}_{s,A}$ is the distribution of $\mathbf{X}|S = s, \mathbf{X} \in A$, and $U > 0$ is a constant only depending on $A$ and $\mathcal{P}_s, s = 0, 1$.*

The first term of RHS, $L\left(\mathbb{E}_s\|\mathbf{X} - \mathbf{T}_s^f(\mathbf{X})\|^2\right)^{1/2}$, implies that using a transport map with a small transport cost helps improve the level of subset fairness. The uncontrollable term $\mathrm{TV}(\mathcal{P}_{0,A}, \mathcal{P}_{1,A})$ can be small for certain subsets. For example, for disjoint sets $A_1, \cdots, A_K$ of $A$, suppose that $\mathcal{P}_s$ is a mixture of uniform distribution given as $\mathcal{P}_s(\cdot) = \sum_{k=1}^K p_{sk} \mathbb{I}(\cdot \in A_k)$ with $p_{sk} \geq 0$ and $\sum_{k=1}^K p_{sk} = 1$ (e.g., the histogram). Then, $\mathrm{TV}(\mathcal{P}_{0,A}, \mathcal{P}_{1,A})$ becomes zero for all $A_k, k \in [K]$. The last term $U\delta$ becomes small when $\delta$ is small.

**Connection to counterfactual fairness**   The concept of FTM is also related to counterfactual fairness (Kusner et al., 2017). In specific, under a simple Structural Causal Model (SCM), the input transported by the marginal OT map is equivalent to the counterfactual input. Particularly, let $\mathbf{X}_s = \mathbf{X}|S = s$ for $s \in \{0, 1\}$ and consider an SCM

$$\mathbf{X}_s = \mu_s + A\mathbf{X}_s + \epsilon_s \tag{4}$$

for given $A \in \mathbb{R}^{d\times d}$, $\mu_s \in \mathbb{R}^d$ with Gaussian random noise $\epsilon_s \sim \mathcal{N}(0, \sigma_s^2 D)$ of a diagonal matrix $D \in \mathbb{R}^{d\times d}$ and variance scaler $\sigma_s^2$. An example DAG (Directed Acyclic Graph) for equation (4) is Figure 2.

Let $\mathbf{x}_s$ be a realization of $\mathbf{X}_s$ and assume $(\mathbf{I}_d - A)$ has its inverse $B :=$ $(\mathbf{I}_d - A)^{-1}$, where $\mathbf{I}_d \in \mathbb{R}^{d\times d}$ is the identity matrix. Then, its counterfactual becomes $\tilde{\mathbf{x}}_s^{\mathrm{CF}} = B\mu_{s'} + \sigma_{s'}\sigma_s^{-1}\mathbf{I}_d(\mathbf{x}_s - B\mu_s)$ (proved with Proposition 4.3 together in Section A of Appendix). On the other hand, by Lemma A.3 in Section A of Appendix, the image of $\mathbf{x}_s$ by the marginal OT map is given as $\tilde{\mathbf{x}}_s^{\mathrm{OT}} = B\mu_{s'} + \mathbf{W}_s(\mathbf{x}_s - B\mu_s)$ for some $\mathbf{W}_s \in \mathbb{R}^{d\times d}$. Proposition below 4.3 shows $\tilde{\mathbf{x}}_s^{\mathrm{CF}} = \tilde{\mathbf{x}}_s^{\mathrm{OT}}$, whose proof is deferred to Section A of Appendix.

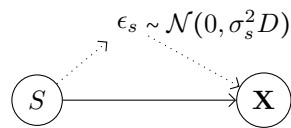

Figure 2: An example DAG of the SCM in equation (4).

**Proposition 4.3** (Counterfactual fairness and the marginal OT map). *For all $A$ having $(\mathbf{I}_d - A)^{-1}$, $\mathbf{W}_s$ becomes $\sigma_{s'}\sigma_s^{-1}\mathbf{I}_d$. That is, $\tilde{\mathbf{x}}_s^{CF} = \tilde{\mathbf{x}}_s^{OT}$.*

We further present an example in Section B of Appendix showing that a transport map with a high transport cost can lead to a problematic group-fair model. Moreover, Section 5.2.2 empirically shows that group-fair models learned by FTM with the marginal OT map attain higher fairness levels on various subsets that are not explicitly considered in the training phase, compared to group-fair models learned by existing algorithms.

**Estimation of the marginal OT map**   To estimate the marginal OT map in practice, we sample two random mini-batches $\tilde{\mathcal{D}}_0 = \{\mathbf{x}_i^{(0)}\}_{i=1}^m \subset \mathcal{D}_0$ and $\tilde{\mathcal{D}}_1 = \{\mathbf{x}_j^{(1)}\}_{j=1}^m \subset \mathcal{D}_1$ with an identical size $m \leq n$. For given two empirical distributions on $\tilde{\mathcal{D}}_0$ and $\tilde{\mathcal{D}}_1$, the cost matrix between the two is defined by $\mathbf{C} := [c_{i,j}] \in \mathbb{R}_+^{m\times m}$ where $c_{i,j} = \|\mathbf{x}_i^{(0)} - \mathbf{x}_j^{(1)}\|^2$. The optimal coupling is then defined by the matrix $\Gamma = [\gamma_{i,j}] \in \mathbb{R}_+^{m\times m}$, which solves the following objective:

$$\min_\Gamma \|\mathbf{C} \odot \Gamma\|_1 = \min_{\gamma_{i,j}} c_{i,j}\gamma_{i,j} \text{ s.t. } \sum_{i=1}^m \gamma_{i,j} = \sum_{j=1}^m \gamma_{i,j} = \frac{1}{m}, \gamma_{i,j} \geq 0. \tag{5}$$

Due to the equal sample sizes of $\tilde{\mathcal{D}}_s, s \in \{0, 1\}$, the optimal coupling has only one non-zero (positive) entry for each row and column (i.e., $\Gamma$ is a doubly stochastic matrix scaled by a constant factor of $1/m$). Then, the marginal OT map for each $\mathbf{x}_i^{(0)} \in \tilde{\mathcal{D}}_0$ is defined by $\mathbf{T}_{0,\tilde{\mathcal{D}}}(\mathbf{x}_i^{(0)}) = \mathbf{x}_j^{(1)}\mathbb{I}(\gamma_{i,j} > 0)$ and $\mathbf{T}_{1,\tilde{\mathcal{D}}}$ is defined similarly.

## 4.2   OT map on $\mathcal{X} \times \mathcal{Y}$

One might argue that using the marginal OT map could degrade the prediction performance much, since the matchings done by the marginal OT map do not consider the similarity in $Y$. As a remedy for this issue, we consider incorporating the label $Y$ into the cost matrix calculation to avoid substantial degradation in prediction performance.

For this purpose, we define a new cost function on $\mathcal{X} \times \mathcal{Y}$. Let $\alpha$ be a given positive constant. The new cost function $c^\alpha : \mathbb{R}^{d+1} \times \mathbb{R}^{d+1} \to \mathbb{R}_+$ is defined by: $c^\alpha((\mathbf{x}_1, y_1), (\mathbf{x}_2, y_2)) := \|\mathbf{x}_1 - \mathbf{x}_2\|^2 + \alpha|y_1 - y_2|$, for given

$\mathbf{x}_1, \mathbf{x}_2 \in \mathbb{R}^d$ and $y_1, y_2 \in \mathbb{R}$. Among all transport maps from the distribution of $\mathbf{X}, Y | S = s$ to the distribution of $\mathbf{X}, Y | S = s'$, we find the OT map that minimizes this modified transport cost on $\mathcal{X} \times \mathcal{Y}$ (i.e., the expected value of $c^\alpha$, where the expectation is taken over the distribution of $\mathbf{X}, Y | S = s$). We call this OT map on $\mathcal{X} \times \mathcal{Y}$ as the **joint OT map**.

Using this joint OT map with a positive $\alpha$ can contribute to the improvement in prediction accuracy compared to the marginal OT map. This is because, while the marginal OT map does not care labels when matching individuals, the joint OT map tends to match individuals with the same label as much as possible when $\alpha$ is sufficiently large.

Not only the prediction accuracy, but also the level of equalized odds (i.e., demographic parities on the subsets consisting of those with $Y = 0$ and $Y = 1$, respectively) can be improved. This is because, FTM with the joint OT map tends to predict similarly for individuals with the same labels but from different protected groups, which directly aligns with the concept of equalized odds. We can theoretically prove this claim as follows. First, we decompose $\Delta\mathrm{MDP}(f, \mathbf{T}_s)$ as $\Delta\mathrm{MDP}(f, \mathbf{T}_s) = w_0 \mathbb{E}_s \left( |f(\mathbf{X}, s) - f(\mathbf{T}_s(\mathbf{X}), s')| \big| Y = 0 \right) + w_1 \mathbb{E}_s \left( |f(\mathbf{X}, s) - f(\mathbf{T}_s(\mathbf{X}), s')| \big| Y = 1 \right)$, where $w_y := \mathbb{P}(Y = y | S = s), y \in \{0, 1\}$. Note that by the definition of the joint OT map, we have for almost all $\mathbf{x}$ with respect to the distribution of $\mathbf{X}|S = s$, $\mathcal{P}(Y = y | \mathbf{X} = \mathbf{x}, S = s) = \mathcal{P}(Y = y | \mathbf{X} = \mathbf{T}_s(\mathbf{x}), S = s')$ for all $y \in \{0, 1\}$ when $\alpha \to \infty$. Then, we have that $C \max\{\Delta\overline{\mathrm{TPR}}(f), \Delta\overline{\mathrm{FPR}}(f)\} \le w_0 \Delta\overline{\mathrm{TPR}}(f) + w_1 \Delta\overline{\mathrm{FPR}}(f) \le \Delta\mathrm{MDP}(f, \mathbf{T}_s)$, where $C = \min\{w_0, w_1\}$ is a constant. Here, $\Delta\overline{\mathrm{TPR}}(f) = |\mathbb{E}(f_0(\mathbf{X})|Y = 1, S = 0) - \mathbb{E}(f_1(\mathbf{X})|Y = 1, S = 1)|$ and $\Delta\overline{\mathrm{FPR}}(f) = |\mathbb{E}(f_0(\mathbf{X})|Y = 0, S = 0) - \mathbb{E}(f_1(\mathbf{X})|Y = 0, S = 1)|$ are the smooth versions of $\Delta\mathrm{TPR}(f)$ and $\Delta\mathrm{FPR}(f)$, respectively. Hence, we can conclude that using the joint OT map with large $\alpha$ can improve EO, compared to the marginal OT map.

We empirically confirm this claim in Section 5.3, by showing that group-fair models learned by FTM with the joint OT map offer improved prediction accuracies as well as improved levels of equalized odds, when compared to FTM with the marginal OT map.

**Estimation of the joint OT map**   We apply a similar technique to the marginal OT map case, starting by sampling two random mini-batches $\tilde{\mathcal{D}}_0$ and $\tilde{\mathcal{D}}_1$ with an identical size $m \le n$. Let $y_i^{(0)}$ and $y_j^{(1)}$ be the corresponding labels of $\mathbf{x}_i^{(0)} \in \tilde{\mathcal{D}}_0$ and $\mathbf{x}_j^{(1)} \in \tilde{\mathcal{D}}_1$, respectively. For a given $\alpha \ge 0$, we modify the cost matrix as follows: $\mathbf{C}^\alpha := [c_{i,j}^\alpha] \in \mathbb{R}_+^{m \times m}$ where $c_{i,j}^\alpha = \|\mathbf{x}_i^{(0)} - \mathbf{x}_j^{(1)}\|^2 + \alpha |y_i^{(0)} - y_j^{(1)}|$. Note that when $\alpha = 0$, this problem becomes equivalent to the case of the marginal OT map in equation (5). We similarly calculate the optimal coupling the matrix by solving the following objective:

$$\min_\Gamma \|\mathbf{C}^\alpha \odot \Gamma\|_1 = \min_{\gamma_{i,j}} c_{i,j}^\alpha \gamma_{i,j} \text{ s.t. } \sum_{i=1}^m \gamma_{i,j} = \sum_{j=1}^m \gamma_{i,j} = \frac{1}{m}, \gamma_{i,j} \ge 0. \tag{6}$$

Then, the joint OT map for each $\mathbf{x}_i^{(0)} \in \tilde{\mathcal{D}}_0$ is defined by $\mathbf{T}_{0,\tilde{\mathcal{D}}}(\mathbf{x}_i^{(0)}) = \mathbf{x}_j^{(1)} \mathbb{1}(\gamma_{i,j} > 0)$ and $\mathbf{T}_{1,\tilde{\mathcal{D}}}$ is defined similarly.

### 4.3   Empirical algorithm for FTM

In practice, we learn $f$ with a stochastic gradient descent algorithm. For each update, to calculate the expected loss, we sample a random mini-batch $\mathcal{D}' \subset \mathcal{D}$ of size $n' \le n$. Then, we update the solution using the gradient of the following objective function

$$\mathcal{L}(f) := \frac{1}{n'} \sum_{(\mathbf{x}_i, y_i, s_i) \in \mathcal{D}'} l(y_i, f(\mathbf{x}_i, s_i)) + \lambda \frac{1}{m} \sum_{\mathbf{x}_i^{(s)} \in \tilde{\mathcal{D}}_s} \left| f(\mathbf{x}_i^{(s)}, s) - f(\mathbf{T}_{s,\tilde{\mathcal{D}}}(\mathbf{x}_i^{(s)}), s') \right| \tag{7}$$

for any $s \in \{0, 1\}$, where $\lambda > 0$ is the Lagrange multiplier and $\mathbf{T}_{s,\tilde{\mathcal{D}}}$ is a pre-specified transport map from $\tilde{\mathcal{D}}_s$ to $\tilde{\mathcal{D}}_{s'}$ (e.g., the marginal OT map from Section 4.1 or the joint OT map from Section 4.2).

# 5 Experiments

This section presents our experimental results, showing that FTM with the proposed transport maps in Section 4 empirically works well to learn group-fair models. The key findings throughout this section are summarized as follows.

- FTM with the marginal OT map successfully learns group-fair models that exhibit (i) competitive prediction performance (Section 5.2.1) and (ii) higher levels of subset fairness (Section 5.2.2), when compared to other group-fair models learned by existing baseline algorithms. Beyond subset fairness, we further evaluate the self-fulfilling prophecy (Dwork et al., 2012) as an additional benefit of low transport cost (see Table 10 and 11 in Section E of Appendix).

- FTM with the joint OT map has the ability to learn group-fair models with improved prediction performance as well as improved levels of equalized odds, when compared to FTM with the marginal OT map (Section 5.3).

## 5.1 Settings

**Datasets** We use four real-world benchmark tabular datasets in our experiments: ADULT (Dua & Graff, 2017), GERMAN (Dua & Graff, 2017), DUTCH (Van der Laan, 2001), and BANK (Moro et al., 2014). The basic information about these datasets is provided in Table 6 in Section D.1 of Appendix. We randomly partition each dataset into training and test datasets with the 8:2 ratio. For each split, we learn models using the training dataset and evaluate the models on the test dataset. This process is repeated 5 times, and the average performance on the test datasets is reported.

**Baseline algorithms and implementation details** For the baseline algorithms, we consider three most popular state-of-the-art methods: Reduction (Agarwal et al., 2018), Reg (minimizing cross-entropy $+ \lambda \Delta \overline{\mathrm{DP}}^2$, which is considered in Donini et al. (2018); Chuang & Mroueh (2021)), and Adv (adversarial learning for ensuring that the model's predictions are independent of sensitive attributes, which is proposed by Zhang et al. (2018)). Additionally, we consider the unfair baseline (abbr. Unfair), the ERM model trained without any fairness regularization or constraint. For the measure of prediction performance, we use the classification accuracy (abbr. `Acc`). For fairness measures, we consider $\Delta \mathrm{DP}$, $\Delta \overline{\mathrm{DP}}$ and $\Delta \mathrm{WDP}$, which are defined in Section 2.2.

For all algorithms, we employ MLP networks with ReLU activation and two hidden layers, where the hidden size is equal to the input dimension. We run all algorithms for 200 epochs and report their final performances on the test dataset. The Adam optimizer (Kingma & Ba, 2014) with the initial learning rate of 0.001 is used. To obtain the OT map for each mini-batch, we solve the linear program by using the POT library (Flamary et al., 2021). We utilize several Intel Xeon Silver 4410Y CPU cores and RTX 3090 GPU processors. More implementation details with `Pytorch`-style psuedo-code are provided in Section D.2 and D.3 of Appendix.

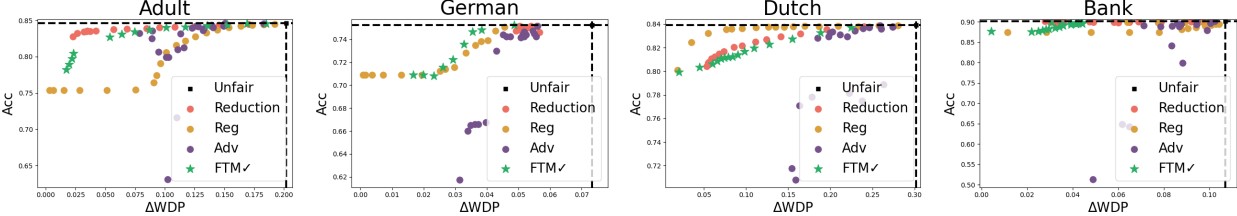

Figure 3: Fairness-prediction trade-offs: $\Delta \mathrm{WDP}$ vs. `Acc` on (Left to right) ADULT, GERMAN, DUTCH, BANK datasets. Similar results are observed for $\Delta \mathrm{DP}$ and $\Delta \overline{\mathrm{DP}}$ in Figure 8 in Section E of Appendix.

## 5.2 FTM with the marginal OT map

This section presents the performance of FTM with the marginal OT map, in terms of (i) fairness-prediction trade-off and (ii) improvement in subset fairness.

### 5.2.1 Fairness-prediction trade-off

In this section, we empirically verify that learned models by FTM successfully achieves group fairness (demographic parity). Figure 3 below shows that FTM successfully learns group-fair models for various fairness levels.

Another main implication is that using the marginal OT map does not hamper prediction performance much. Figure 3 supports this assertion in terms of fairness-prediction trade-off, that is, FTM is competitive with the three baselines. In most datasets, the performance of FTM is not significantly worse than that of the top-performing algorithm (i.e., Reduction). Notably, on GERMAN dataset, FTM performs the best, whereas Reduction fails to learn group-fair models with fairness level under 0.06. Additionally, FTM mostly outperforms the other two baseline algorithms, Reg and Adv. Hence, we can conclude that FTM is also a promising algorithm for achieving group fairness.

### 5.2.2 Improvement in subset fairness

This section highlights the key advantages of using the marginal OT map in terms of subset fairness, which is theoretically supported by Theorem 4.2. We examine two scenarios for the subset $A$ in Definition 4.1: (1) random subsets and (2) subsets defined by specific input variables.

**Random subsets**   First, we generate a random subset $\mathcal{D}_{\mathrm{sub}}$ from the test data defined as $\mathcal{D}_{\mathrm{sub}} = \{i : \mathbf{v}^\top \mathbf{x}_i \geq 0\}$, using a random vector $\mathbf{v}$ drawn from the uniform distribution on $[-1, 1]^d$. Then, we calculate $\Delta \overline{\mathrm{DP}}$ on $\mathcal{D}_{\mathrm{sub}}$. Figure 4 presents boxplots of the $\Delta \overline{\mathrm{DP}}$ values calculated on 1,000 randomly generated $\mathcal{D}_{\mathrm{sub}}$. Outliers in the boxplots (points in red boxes) represent the example instances of subset unfairness. For a fair comparison, we evaluate under a given $\Delta \overline{\mathrm{DP}}$ for each dataset: 0.06 for ADULT, 0.01 for GERMAN, 0.07 for DUTCH, and 0.04 for BANK. Notably, FTM consistently has the fewest outliers than all the baselines, indicating that FTM achieves higher fairness on random subsets.

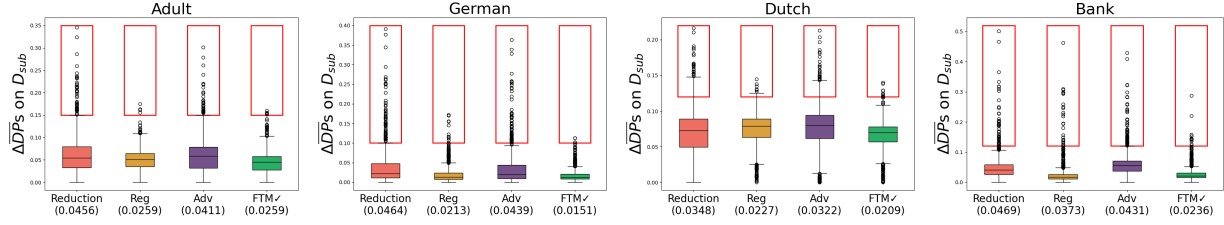

Figure 4: Fairness on random subsets: Boxplots of the levels of $\Delta \overline{\mathrm{DP}}$ on 1,000 randomly generated subsets $\mathcal{D}_{\mathrm{sub}}$ of test datasets. (Left to right) ADULT, GERMAN, DUTCH and BANK. The values presented under the algorithm name (e.g., 0.0151 for FTM in GERMAN) are the standard deviations.

**Subsets defined by specific input variables**   Second, we focus on subsets defined by a specific input variable. For this scenario, we construct two subsets by binarizing a specific input variable using its median value. Note that we learn models considering only gender as the sensitive attribute. Table 1 presents the fairness levels (with respect to gender) in the two subsets of the learned models. We consider GERMAN and DUTCH datasets for this analysis. For GERMAN dataset, the two subsets are defined by $\{\mathbf{x}$ of high age$\}$ and $\{\mathbf{x}$ of low age$\}$. For DUTCH dataset, the two subsets are defined by $\{\mathbf{x}$ who is married$\}$ and $\{\mathbf{x}$ who is not married$\}$. The results highlight the superiority of FTM in achieving higher fairness in these specific subsets.

## 5.3 FTM with the joint OT map

This section shows the effect of using the joint OT map, in terms of improvement in (i) prediction accuracy, level of equalized odds (Section 5.3.1), and (ii) the transport cost (Section 5.3.2). We use ADULT dataset for this analysis. For FTM with the joint OT map, we fix $\alpha = 100$, because the results with $\alpha > 100$ are almost identical to those with $\alpha = 100$. In other words, 100 is the minimum value for $\alpha$ where $\alpha|y_i^{(0)} - y_j^{(1)}|$ fully dominates the transport cost $\|\mathbf{x}_i^{(0)} - \mathbf{x}_j^{(1)}\|^2$.

Table 1: Fairness levels on the subsets defined by specific input variables: **Bold** faced ones highlight the best results, and underlined ones are the second best ones. Standard errors are provided in Table 8 and 9 in Section E of Appendix. (Left) The subsets are defined by the input variable 'age' on GERMAN dataset under a given $\Delta\overline{\text{DP}} = 0.045$. (Right) The subsets are defined by the input variable 'marital status' on DUTCH dataset under a given $\Delta\overline{\text{DP}} = 0.120$.

| Measure ‖ | | GERMAN | | | | DUTCH | | | | |
|---|---|---|---|---|---|---|---|---|---|---|
| Measure ‖ | Subset | Reduction | Reg | Adv | FTM ✓ | Subset | Reduction | Reg | Adv | FTM ✓ |
| $\Delta\text{DP}$ | | 0.073 | 0.077 | 0.048 | **0.045** | | 0.258 | 0.372 | 0.237 | **0.204** |
| $\Delta\overline{\text{DP}}$ | High age | 0.049 | 0.029 | 0.028 | **0.026** | Married | 0.182 | 0.164 | 0.187 | **0.152** |
| $\Delta\text{WDP}$ | | 0.053 | 0.039 | 0.042 | **0.038** | | 0.183 | 0.172 | 0.193 | **0.152** |
| $\Delta\text{DP}$ | | 0.118 | 0.116 | 0.122 | **0.077** | | **0.061** | 0.131 | 0.095 | 0.068 |
| $\Delta\overline{\text{DP}}$ | Low age | **0.047** | 0.050 | 0.053 | **0.047** | Not married | 0.045 | 0.062 | 0.098 | **0.036** |
| $\Delta\text{WDP}$ | | 0.058 | 0.059 | 0.061 | **0.054** | | **0.045** | 0.072 | 0.098 | **0.045** |

### 5.3.1 Improvement in prediction accuracy and equalized odds

Table 2 shows that FTM with the joint OT map can provide higher prediction performance than FTM with the marginal OT map in certain scenarios, where more accurate group-fair models than FTM with the marginal OT map exist (e.g., $\Delta\text{DP} \leq 0.06$ in Figure 3). Furthermore, we observe that the level of equalized odds can also be improved in these scenarios. To assess the level of equalized odds, we basically use the differences in TPR and FPR, defined as $\Delta\text{TPR} := |\mathbb{P}(C_{f_0}(\mathbf{X}) = 1|S = 0, Y = 1) - \mathbb{P}(C_{f_1}(\mathbf{X}) = 1|S = 1, Y = 1)|$ and

Table 2: Comparison between (i) the marginal OT map and (ii) the joint OT map in terms of prediction accuracy and level of equalized odds, with the two fixed fairness level $\Delta$DPs at 0.033 and 0.054.

| $\Delta\text{DP} = 0.033$ ‖ | Acc (↑) | $\Delta\text{TPR}$ (↓) | $\Delta\text{FPR}$ (↓) | $\Delta\text{EO}$ (↓) |
|---|---|---|---|---|
| Marginal OT map | 0.806 | 0.052 | 0.012 | 0.032 |
| Joint OT map | 0.810 | 0.043 | 0.013 | 0.028 |
| $\Delta\text{DP} = 0.054$ ‖ | Acc (↑) | $\Delta\text{TPR}$ (↓) | $\Delta\text{FPR}$ (↓) | $\Delta\text{EO}$ (↓) |
| Marginal OT map | 0.826 | 0.031 | 0.023 | 0.027 |
| Joint OT map | 0.830 | 0.021 | 0.019 | 0.020 |

$\Delta\text{FPR} := |\mathbb{P}(C_{f_0}(\mathbf{X}) = 1|S = 0, Y = 0) - \mathbb{P}(C_{f_1}(\mathbf{X}) = 1|S = 1, Y = 0)|$, respectively. Note that $\Delta\text{TPR}$ is also a measure for equal opportunity. We additionally use an overall measure defined as $\Delta\text{EO} := \frac{1}{2}\sum_{y=0,1}|\mathbb{P}(C_{f_0}(\mathbf{X}) = 1|S = 0, Y = y) - \mathbb{P}(C_{f_1}(\mathbf{X}) = 1|S = 1, Y = y)|$, which is also considered in previous works (Donini et al., 2018; Chuang & Mroueh, 2021).

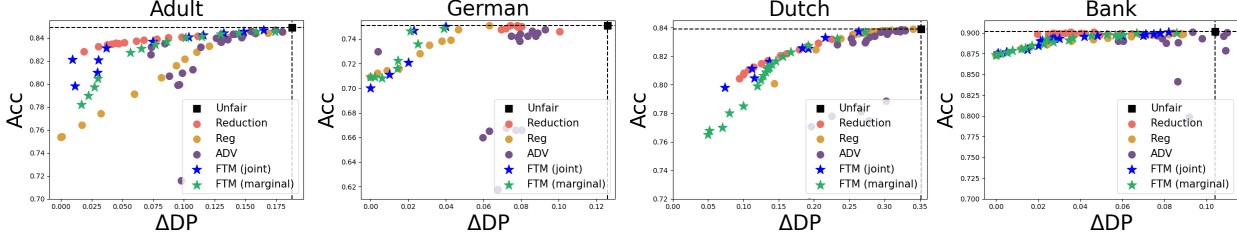

Figure 5: Fairness-prediction trade-offs: $\Delta\text{DP}$ vs. Acc on (Left to right) ADULT, GERMAN, DUTCH, BANK datasets. FTM (joint) = FTM with the joint OT map. FTM (marginal) = FTM with the marginal OT map.

In addition, we compare FTM using the marginal OT map, FTM using the joint OT map, and other baseline methods, in terms of the overall fairness-prediction trade-off. As shown in Figure 5, FTM with the joint OT map achieves performance comparable to the best-performing baseline. This result suggests that using FTM with an appropriate transport map can empirically achieve the optimal trade-off, highlighting the flexibility of FTM in controlling the fairness-prediction trade-off, by selecting an appropriate transport map.

### 5.3.2 Increase in transport cost

On the other hand, using the joint OT map can result in a higher transport cost (of the fair matching function) compared to the marginal OT map. To compute the transport cost of the fair matching function, we follow the mini-batch technique introduced in Section 3.1. We use 100 random mini-batches with batch size of $m = 1024$. The left panel of Figure 6 illustrates that increasing $\alpha$ can improve prediction accuracy, though this improvement comes with a higher transport cost, especially when group-fair models more accurate than FTM with the marginal OT map exist. That is, at $\Delta DP = 0.025$, a point where a group-fair model more accurate than FTM with the marginal OT map exists (e.g., Reduction in Figure 3), both accuracy and transport cost increase, as $\alpha$ increases.

In contrast, the right panel of Figure 6 shows that increasing $\alpha$ does not significantly improve prediction accuracy while still incurring a higher transport cost, when FTM with the marginal OT map is competitive with other group-fair models in terms of accuracy. That is, at $\Delta DP = 0.073$, a point where the accuracy of FTM with the marginal OT map is similar to that of other group-fair models, increasing $\alpha$ does not yield notably beneficial results.

Overall, FTM with the joint OT map using an appropriately tuned $\alpha$ could be a desirable solution, especially when seeking for a group-fair model that is more accurate than a group-fair model learned by FTM with the marginal OT map. However, tuning $\alpha$ can be challenging, and compromising subset fairness would be generally not advisable. Additionally, as discussed in Section 5.2.1, using the marginal OT map is also competitive with other baselines in terms of prediction accuracy. Therefore, we basically recommend using the marginal OT map for FTM, while considering the joint OT map is particularly useful when the prediction accuracy of a group-fair model learned by FTM with the marginal OT map is suboptimal.

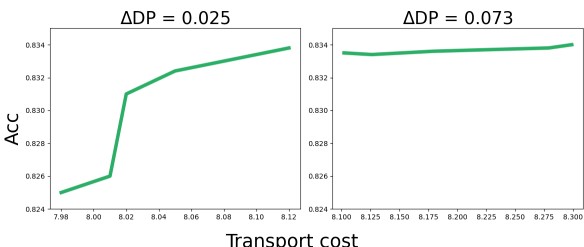

Figure 6: Transport cost-prediction trade-offs: Transport cost of the fair matching function vs. `Acc` of FTM with the joint OT map of $\alpha \in \{0, 1, 5, 10, 50, 100\}$ on ADULT dataset.

## 5.4 Further comparisons

To further assess the validity of FTM from various perspectives, we consider three scenarios for empirical comparison with baseline algorithms: (i) FTM is robust under a group imbalance setting (i.e., $\mathcal{P}(S = 0) << \mathcal{P}(S = 1)$). (ii) FTM can improve the level of individual fairness, compared to existing group fairness algorithms. (iii) FTM outperforms several existing FRL methods, by achieving higher prediction accuracy for a given level of fairness. For simplicity, we employ the marginal OT map in this section for these analyses.

### 5.4.1 Comparison of stability under imbalance setting

We evaluate the stability of FTM and existing algorithms under an imbalanced scenario for the sensitive attribute. For this purpose, after training/test data split, we construct an imbalanced training dataset with a 5:95 ratio (5% for $S = 0$ and 95% for $S = 1$) by randomly sampling data from $\mathcal{D}_0$ and fully using $\mathcal{D}_1$. Then, we measure the performances on test dataset, where the models are learned on this imbalanced training dataset. We use ADULT dataset for this analysis. For a fixed fairness level of $\Delta \overline{DP} \approx 0.064$, we calculate the corresponding prediction accuracy. This procedure is repeated five times, and we calculate the average as well as the standard deviation.

In Table 3, we present the average, standard deviation, and coefficient of variation (standard deviation ÷ average) for each algorithm. The results indicate that FTM offers comparable stability to the baselines, showing that FTM is not particularly unstable under this group imbalance setting.

Table 3: Stability under imbalance setting: For a fixed fairness level $\Delta\overline{\text{DP}} \approx 0.064$, average (Avg), standard deviation (Std), and coefficient of variation (CV) of the prediction accuracy (`Acc`) over five random training imbalanced datasets are provided.

| Measure | Reduction | Reg | Adv | FTM ✓ |
|---|---|---|---|---|
| Avg | 0.835 | 0.836 | 0.819 | 0.827 |
| Std | 0.003 | 0.003 | 0.021 | 0.002 |
| CV | 0.004 | 0.004 | 0.026 | 0.002 |

Table 4: Comparison in view of individual fairness: Given a fixed $\Delta$DP, the level of individual fairness (i.e., `Con`) and corresponding `Acc` are provided. The bold faced values are the best values of `Con`.

| Dataset ($\Delta$DP) | Con (Acc) | | | |
|---|---|---|---|---|
| | Reduction | Reg | Adv | FTM ✓ |
| ADULT ($\approx 0.08$) | 0.915 (0.839) | 0.902 (0.802) | 0.899 (0.829) | **0.935** (0.838) |
| GERMAN ($\approx 0.06$) | 0.950 (0.743) | 0.932 (0.745) | 0.944 (0.672) | **0.965** (0.744) |
| DUTCH ($\approx 0.17$) | 0.952 (0.824) | 0.914 (0.815) | 0.921 (0.778) | **0.980** (0.824) |
| BANK ($\approx 0.04$) | 0.927 (0.885) | 0.978 (0.885) | 0.911 (0.884) | **0.972** (0.885) |

### 5.4.2 Comparison in view of individual fairness

As discussed in Section 3.3, as FTM is conceptually similar to the individual fairness, we here compare FTM to baseline algorithms for group fairness, in view of individual fairness. For the measure of individual fairness, we use the consistency (`Con`) from Yurochkin et al. (2020); Yurochkin & Sun (2021), which is the ratio of consistently predicted labels when we only flip the sensitive variable among the input variables. Table 4 presents the results, showing that FTM consistently achieves higher individual fairness than baselines.

### 5.4.3 Comparison with FRL methods

Table 5: Comparison between FTM and FRL methods: Given a fixed $\Delta\overline{\text{DP}}$, the prediction accuracy (`Acc`) is provided. The bold faced values are the best values of `Acc`.

| Dataset ($\Delta$DP) | Acc | | | |
|---|---|---|---|---|
| | LAFTR | sIPM-LFR | Gordaliza et al. (2019) | FTM ✓ |
| ADULT ($\approx 0.06$) | 0.823 | 0.820 | 0.815 | **0.835** |
| GERMAN ($\approx 0.04$) | 0.721 | 0.741 | 0.728 | **0.743** |
| DUTCH ($\approx 0.15$) | 0.810 | 0.813 | 0.811 | **0.820** |
| BANK ($\approx 0.02$) | 0.877 | 0.878 | 0.876 | **0.878** |

In this section, we experimentally compare FTM with several existing FRL methods. The baseline FRL methods include three approaches: two adversarial learning-based FRL methods, LAFTR (Madras et al., 2018) and sIPM-LFR (Kim et al., 2022a), as well as the FRL method proposed by Gordaliza et al. (2019), which is particularly similar to FTM in that both utilize the OT map for aligning distributions. Note that in FRL, fair representations are first obtained (via barycentric mapping for Gordaliza et al. (2019) and adversarial learning for LAFTR and sIPM-LFR), and then we learn a prediction model on the fair representation space.

The results are presented in Table 5, showing the superior prediction performance of FTM compared to the baseline FRL methods. This outperformance of FTM is due to the fact that FRL methods are pre-processing approaches (i.e., using fair representation as a new input feature for the prediction model), whereas FTM is an in-processing approach.

# 6 Conclusion and discussion

In this paper, we have discussed the existence of implicit transport maps corresponding to each group-fair model. Specifically, we have introduced a novel group fairness measure named MDP. Building upon MDP, we propose a novel algorithm, FTM, designed for learning group-fair models with high levels of subset fairness. Experimental results demonstrate that FTM with the marginal OT map effectively produces group-fair models with improved levels of subset fairness on various subsets compared to baseline models, while maintaining reasonable prediction performance. Moreover, we have proposed to use the joint OT map to improve the prediction accuracy and equalized odds, compared to the marginal OT map.

We suggest several promising topics for future research: (1) Expanding FTM to incorporate other fairness notions would be a valuable avenue for future work. For example, while this paper has focused DP (demographic parity) for simplicity and clarity, but applying FTM to Eqopp (equal opportunity) would be straightforward by restricting the calculation of MDP to instances where $Y = 1$. (2) Exploring scenarios with multiple sensitive attributes, a topic frequently studied in group fairness research, is another valuable direction. Such cases require matching individuals from more than two protected groups, which would be challenging and is therefore left for future work. (3) There may be other transport maps that yield different types of group-fair models, while this paper has only considered two transport maps. Exploring other useful transport maps would yield interesting insights and applications.

**Broader Impact Statement**

The broad goal of this study is to caution users of fair AI models – such as social planners and courts – against focusing solely on group fairness without accounting for the risks of potential discrimination, such as subset fairness. Additionally, our study aims to equip them with tools to address these aspects. Even though our study is rather technical, we believe that it can provide a new perspective on algorithmic fairness and could potentially impact policy-making and regulation in related fields.

A key social benefit of the proposed methods is that we can train group-fair models with higher levels of subset fairness without needing to collect and process additional sensitive data. By doing so, the proposed algorithm is expected to transcend the fairness-privacy trade-off, making it practical for use without conflicting with data protection laws.

Another social impact of our study is that the relationship between transport maps and group-fair models may help us form a new concept of fairness that can be readily accepted by society. Our approach explores the micro-level behavior of a given group-fair model (i.e., how the model matches individuals rather than simply looking at the statistics), which could facilitate finding reasonable compromises for existing fairness concepts that may appear paradoxical.

**Acknowledgments**

YK was partly supported by the National Research Foundation of Korea(NRF) grant funded by the Korea government(MSIT)(No. 2022R1A5A7083908), Institute of Information & communications Technology Planning & Evaluation (IITP) grant funded by the Korea government(MSIT) (No.RS-2022-II220184, Development and Study of AI Technologies to Inexpensively Conform to Evolving Policy on Ethics), and Institute of Information & communications Technology Planning & Evaluation (IITP) grant funded by the Korea government(MSIT) [NO.RS-2021-II211343, Artificial Intelligence Graduate School Program (Seoul National University)]. MC was supported by a Korea Institute for Advancement of Technology (KIAT) grant funded by the Korea Government (MOTIE) (RS-2024-00409092, 2024 HRD Program for Industrial Innovation).

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

## A  Proofs of theorems

Let $\mathbb{N}$ be the set of natural numbers. For $s = 0, 1$ and any measurable set $A \subseteq [0, 1]$, we denote $f_s^{-1}(A) \coloneqq \{\mathbf{x} \in \mathcal{X}_s : f(\mathbf{x}, s) \in A\}$. Recall that $\mathcal{X}_s$ is the domain of $\mathbf{X}|S = s$.

**Proposition 3.1** For any perfectly group-fair model $f$, i.e., $\mathcal{P}_{f_0} = \mathcal{P}_{f_1}$, there exists a transport map $\mathbf{T}_s = \mathbf{T}_s(f)$ satisfying $f(\mathbf{X}, s) = f(\mathbf{T}_s(\mathbf{X}), s')$, $a.e.$

*Proof of Proposition 3.1.* By letting $\delta = 0$, Theorem 3.3 below implies $\mathbb{E}_s |f(\mathbf{X}, s) - f(\mathbf{T}_s(\mathbf{X}), s')| = 0$. This implies that $|f(\mathbf{X}, s) - f(\mathbf{T}_s(\mathbf{X}), s')| = 0$ almost everywhere, which concludes the proof. □

**Theorem 3.3** Fix a fairness level $\delta \geq 0$. For any given group-fair model $f$ such that $\Delta \mathrm{TVDP}(f) \leq \delta$, there exists a transport map $\mathbf{T}_s \in \mathcal{T}_s^{\mathrm{trans}}$ satisfying $\Delta \mathrm{MDP}(f, \mathbf{T}_s) \leq 2\delta$.

*Proof of Theorem 3.3.* Without loss of generality, let $s = 0$ and $s' = 1$. Let $F_0 : [0, 1] \to [0, 1]$ and $F_1 : [0, 1] \to [0, 1]$ denote the cumulative distribution functions (CDFs) of $f(\mathbf{X}, 0)|S = 0$ and $f(\mathbf{X}, 1)|S = 1$, respectively. Note that $F_0$ and $F_1$ have at most countably many discontinuity points.

Define $D_s$ as the set of all discontinuity points of $F_s$, which is countable. For each $t \in D_s$, define the jump at $t$ as $\Delta F_s(t) \coloneqq F_s(t) - F_s(t-)$, where $F_s(t-)$ denotes the left limit of $F_s$ at $t$.

Define the sub-CDF (i.e., the continuous part of $F_s$) as: $F_s^{cont}(v) \coloneqq F_s(v) - \sum_{t \in D_s, t \leq v} \Delta F_s(t)$ for $v \in [0, 1]$. Note that this function $F_s^{cont}$ is continuous and non-decreasing on $[0, 1]$. Moreover, for any interval $(a, b] \subseteq [0, 1]$, define $F_s^{cont}((a, b]) \coloneqq F_s^{cont}(b) - F_s^{cont}(a)$.

We here prove for the case when $F_1^{cont}(1) \leq F_0^{cont}(1)$. The other case $F_1^{cont}(1) > F_0^{cont}(1)$ can be treated similarly.

1. (Constructing subsets based on $F_s^{cont}$) Since $F_1^{cont}(1) \leq F_0^{cont}(1)$, there exists $z \leq 1$ such that $F_1^{cont}(1) = F_0^{cont}(z)$. We partition the interval $[0, z]$ into subintervals of length at most $\delta$. Let $v_0 = 0$ and define $v_k \coloneqq \min\{v_{k-1} + \delta, z\}$ for $k = 1, \ldots, m$. Here, $m \in \mathbb{N}$ is the number that satisfies $v_m = z$. Note that $F_1^{cont}(1) = \sum_{k=1}^m F_1^{cont}((v_{k-1}, v_k])$.

   For each $k \in \{1, \ldots, m\}$, we compare the measures of the intervals as follows.

   If $F_1^{cont}((v_{k-1}, v_k]) \leq F_0^{cont}((v_{k-1}, v_k])$, then there exists $z_k \in [v_{k-1}, v_k]$ such that $F_1^{cont}((v_{k-1}, v_k]) = F_0^{cont}((v_{k-1}, z_k])$. Define $\mathcal{X}_{0,k} \coloneqq f_0^{-1}((v_{k-1}, z_k] \smallsetminus D_0)$ and $\mathcal{X}_{1,k} \coloneqq f_1^{-1}((v_{k-1}, v_k] \smallsetminus D_1)$.
   If $F_1^{cont}((v_{k-1}, v_k]) > F_0^{cont}((v_{k-1}, v_k])$, we can define $z_k$, $\mathcal{X}_{0,k}$ and $\mathcal{X}_{1,k}$ similarly.

2. (Defining probability measures and transport maps on subsets) For each $k \in \{1, \ldots, m\}$, define probability measures $\mathcal{P}_{s,k}, s \in \{0, 1\}$ such that $\mathcal{P}_{s,k}(A) \coloneqq \frac{\mathcal{P}_s(A \cap \mathcal{X}_{s,k})}{\mathcal{P}_s(\mathcal{X}_{s,k})}$ for measurable subsets $A \subseteq \mathcal{X}$.

   By Brenier's Theorem (Villani, 2008; Hütter & Rigollet, 2021), there exists a transport map $\mathbf{T}_{0,k}^{(1)}$ from $\mathcal{P}_{0,k}(\cdot)$ to $\mathcal{P}_{1,k}(\cdot)$, under (C). Since $v_k - v_{k-1} \leq \delta, \forall k$, we have that

   $$|f(\mathbf{x}, 0) - f(\mathbf{T}_{0,k}^{(1)}(\mathbf{x}), 1)| \leq \delta, \forall \mathbf{x} \in \mathcal{X}_{0,k}. \tag{8}$$

3. (Handling discontinuity points) Let $D_{0,1} \coloneqq D_0 \cap D_1$ be the intersection of $D_0$ and $D_1$, which is the set of common discontinuity points.

   Fix $d \in D_{0,1}$. Suppose that $\mathcal{P}_1(f_1^{-1}(\{d\})) \leq \mathcal{P}_0(f_0^{-1}(\{d\}))$. Then, there exists $f_0^{-1}(\{d\})' \subset f_0^{-1}(\{d\})$ such that $\mathcal{P}_0(f_0^{-1}(\{d\})') = \mathcal{P}_1(f_1^{-1}(\{d\}))$. Define $\tilde{\mathcal{X}}_{0,d} \coloneqq f_0^{-1}(\{d\})'$ and $\tilde{\mathcal{X}}_{1,d} \coloneqq f_1^{-1}(\{d\})$. We can define $\tilde{\mathcal{X}}_{0,d}$ and $\tilde{\mathcal{X}}_{1,d}$ similarly when $\mathcal{P}_1(f_1^{-1}(\{d\})) > \mathcal{P}_0(f_0^{-1}(\{d\}))$.

   For each $d \in D_{0,1}$, we define probability measures $\tilde{\mathcal{P}}_{s,d}, s \in \{0, 1\}$ such that $\tilde{\mathcal{P}}_{s,d}(A) \coloneqq \mathcal{P}_s(A \cap \tilde{\mathcal{X}}_{s,d})/\mathcal{P}_s(\tilde{\mathcal{X}}_{s,d})$ for measurable subsets $A \subseteq \mathcal{X}$. Then, there exists a transport map $\mathbf{T}_{0,d}^{(2)}$ from $\tilde{\mathcal{P}}_{0,d}(\cdot)$ to $\tilde{\mathcal{P}}_{1,d}(\cdot)$. By the definition of $\tilde{\mathcal{X}}_{0,d}$, we have that

   $$f(\mathbf{x}, 0) = f(\mathbf{T}_{0,d}^{(2)}(\mathbf{x}), 1) = d, \forall \mathbf{x} \in \tilde{\mathcal{X}}_{0,d}. \tag{9}$$

4. (Constructing the complement parts) We collect the complements as

$$\mathcal{X}_0' \coloneqq \mathcal{X}_0 \smallsetminus \left( \bigcup_{k=1}^{m} \mathcal{X}_{0,k} \cup \bigcup_{d \in D_{0,1}} \tilde{\mathcal{X}}_{0,d} \right) \text{ and } \mathcal{X}_1' \coloneqq \mathcal{X}_1 \smallsetminus \left( \bigcup_{k=1}^{m} \mathcal{X}_{1,k} \cup \bigcup_{d \in D_{0,1}} \tilde{\mathcal{X}}_{1,d} \right).$$

Because $\mathcal{P}_0(\bigcup_{k=1}^{m} \mathcal{X}_{0,k}) = \mathcal{P}_1(\bigcup_{k=1}^{m} \mathcal{X}_{1,k})$ and $\mathcal{P}_0(\bigcup_{d \in D_{0,1}} \tilde{\mathcal{X}}_{0,d}) = \mathcal{P}_1(\bigcup_{d \in D_{0,1}} \tilde{\mathcal{X}}_{1,d})$, we have $\mathcal{P}_0(\mathcal{X}_0') = 1 - \mathcal{P}_0(\bigcup_{k \in \{1,\dots,m\}} \mathcal{X}_{0,k}) - \mathcal{P}_0(\bigcup_{d \in D_{0,1}} \tilde{\mathcal{X}}_{0,d}) = \mathcal{P}_1(\mathcal{X}_1') \le \delta$.

Define probability measures $\mathcal{P}_s', s \in \{0,1\}$ on $\mathcal{X}_s', s \in \{0,1\}$ such that $\mathcal{P}_s'(A) \coloneqq \frac{\mathcal{P}_s(A \cap \mathcal{X}_s')}{\mathcal{P}_s(\mathcal{X}_s')}$ for measurable subsets $A \subseteq \mathcal{X}$. Then, there exists a transport map $\mathbf{T}_0^{(3)}$ from $\mathcal{P}_0'(\cdot)$ to $\mathcal{P}_1'(\cdot)$.

For $\mathbf{x} \in \mathcal{X}_0'$, we have $\mathcal{P}_0(\mathcal{X}_0') = 1 - \left( \sum_{k=1}^{m} \mathcal{P}_0(\mathcal{X}_{0,k}) + \sum_{d \in D_{0,1}} \mathcal{P}_0(\tilde{\mathcal{X}}_{0,d}) \right) \le \delta$, since $\Delta \text{TVDP}(f) = \text{TV}\left( \mathcal{P}_{f(\mathbf{X},0)|S=0}, \mathcal{P}_{f(\mathbf{X},1)|S=1} \right) \le \delta$. Furthermore, since $f(\cdot) \in [0,1]$, we have that

$$\begin{aligned}
\mathbb{E}_0 \left( |f(\mathbf{X},0) - f(\mathbf{T}_0^{(3)}(\mathbf{X}),1)| \cdot \mathbb{1}(\mathbf{X} \in \mathcal{X}_0') \right) &= \int |f(\mathbf{X},0) - f(\mathbf{T}_0^{(3)}(\mathbf{X}),1)| \cdot \mathbb{1}(\mathbf{X} \in \mathcal{X}_0') d\mathcal{P}_0(\mathbf{X}) \\
&\le \int \mathbb{1}(\mathbf{X} \in \mathcal{X}_0') d\mathcal{P}_0(\mathbf{X}) = \mathcal{P}_0(\mathcal{X}_0') \le \delta.
\end{aligned} \tag{10}$$

5. (Overall transport map) Finally, combining 2 to 4 above, we define the (overall) transport map $\mathbf{T}_0$ as

$$\mathbf{T}_0(\cdot) \coloneqq \sum_{k=1}^{m} \mathbf{T}_{0,k}^{(1)}(\cdot) \mathbb{1}(\cdot \in \mathcal{X}_{0,k}) + \sum_{d \in D_{0,1}} \mathbf{T}_{0,d}^{(2)}(\cdot) \mathbb{1}(\cdot \in \tilde{\mathcal{X}}_{0,d}) + \mathbf{T}_0^{(3)}(\cdot) \mathbb{1}(\cdot \in \mathcal{X}_0'). \tag{11}$$

We note that $\left\{ \{\mathcal{X}_{0,k}\}_{k=1}^{m}, \{\tilde{\mathcal{X}}_{0,d}\}_{d \in D_{0,1}}, \mathcal{X}_0' \right\}$ and $\left\{ \{\mathcal{X}_{1,k}\}_{k=1}^{m}, \{\tilde{\mathcal{X}}_{1,d}\}_{d \in D_{0,1}}, \mathcal{X}_1' \right\}$ are partitions of $\mathcal{X}_0$ and $\mathcal{X}_1$, respectively. Moreover, $\mathcal{P}_0(\mathcal{X}_{0,k}) = \mathcal{P}_1(\mathcal{X}_{1,k}), \forall k, \mathcal{P}_0(\tilde{\mathcal{X}}_{0,d}) = \mathcal{P}_1(\tilde{\mathcal{X}}_{1,d}), \forall d,$ and $\mathcal{P}_0(\mathcal{X}_0') = \mathcal{P}_1(\mathcal{X}_1')$. Hence, $\mathbf{T}_0$ is a transport map from $\mathcal{P}_0$ to $\mathcal{P}_1$.

6. (Calculation of the bound for $\Delta \text{MDP}(f, \mathbf{T}_0)$) Using the constructed transport map $\mathbf{T}_0$, we have that

$$\begin{aligned}
\Delta \text{MDP}(f, \mathbf{T}_0) = \mathbb{E}_0 |f(\mathbf{X},0) - f(\mathbf{T}_0(\mathbf{X}),1)| &= \int |f(\mathbf{X},0) - f(\mathbf{T}_0(\mathbf{X}),1)| d\mathcal{P}_0(\mathbf{X}) \\
&= \sum_{k=1}^{m} \int_{\mathcal{X}_{0,k}} |f(\mathbf{x},0) - f(\mathbf{T}_{0,k}^{(1)}(\mathbf{x}),1)| d\mathcal{P}_0(\mathbf{x}) \\
&\quad + \sum_{d \in D_{0,1}} \int_{\tilde{\mathcal{X}}_{0,d}} |f(\mathbf{X},0) - f(\mathbf{T}_{0,d}^{(2)}(\mathbf{x}),1)| \cdot d\mathcal{P}_0(\mathbf{x}) \\
&\quad + \int_{\mathcal{X}_0'} |f(\mathbf{x},0) - f(\mathbf{T}_0^{(3)}(\mathbf{x}),1)| d\mathcal{P}_0(\mathbf{x}) \\
&\underset{(*)}{\le} \delta \sum_{k=1}^{m} \mathcal{P}_0(\mathcal{X}_{0,k}) + \int_{\mathcal{X}_0'} |f(\mathbf{x},0) - f(\mathbf{T}_0^{(3)}(\mathbf{x}),1)| d\mathcal{P}_0(\mathbf{x}) \\
&\le \delta + \delta = 2\delta,
\end{aligned} \tag{12}$$

where (*) holds by the inequalities in (8), (9), and (10).

$\square$

**Theorem 3.3 without the condition (C): the case when $\mathcal{P}_0$ and $\mathcal{P}_1$ are discrete**   The condition (C) is assumed for easier discussion involving continuous distributions. This is because the existence of (deterministic) transport maps is not guaranteed when the distributions $\mathcal{P}_0$ and $\mathcal{P}_1$ are discrete and their supports are different. However, by using the notion of stochastic transport map (defined below), we can derive a theoretical result similar to Theorem 3.3 when $\mathcal{P}_0$ and $\mathcal{P}_1$ are discrete.

Let $\mathcal{X}_0 = \{\mathbf{x}_1^{(0)}, \ldots, \mathbf{x}_{n_0}^{(0)}\}$ and $\mathcal{X}_1 = \{\mathbf{x}_1^{(1)}, \ldots, \mathbf{x}_{n_1}^{(1)}\}$. For this time only, define $\mathcal{P}_0$ and $\mathcal{P}_1$ as the empirical distributions on $\mathcal{D}_0$ and $\mathcal{D}_1$, respectively. Let $\mathbf{X}_0$ and $\mathbf{X}_1$ be the random variables following $\mathcal{P}_0$ and $\mathcal{P}_1$, respectively. Furthermore, let $f(\mathcal{X}_0) \coloneqq \{f(\mathbf{x}, 0) : \mathbf{x} \in \mathcal{X}_0\}$ and $f(\mathcal{X}_1) \coloneqq \{f(\mathbf{x}, 1) : \mathbf{x} \in \mathcal{X}_1\}$. Denote $\mathcal{P}_{f_0}$ and $\mathcal{P}_{f_1}$ be the empirical distributions on $f(\mathcal{X}_0)$ and $f(\mathcal{X}_1)$, i.e., the distributions of $f(\mathbf{X}_0), \mathbf{X}_0 \sim \mathcal{P}_0$ and $f(\mathbf{X}_1), \mathbf{X}_1 \sim \mathcal{P}_1$, respectively.

For a given joint distribution $\mathbb{Q}$ between $\mathbf{X}_0$ and $\mathbf{X}_1$, we can define $\Delta\mathrm{MDP}(f, \mathbb{Q}) \coloneqq \mathbb{E}_{(\mathbf{X}_0, \mathbf{X}_1) \sim \mathbb{Q}} |f(\mathbf{X}_0, 0) - f(\mathbf{X}_1, 1)|$, which is a general version of $\Delta\mathrm{MDP}(f, \mathbf{T}_s)$. That is, instead of the deterministic (one-to-one) transport map, we define the stochastic transport map $\mathbf{T}_s$ defined by $\mathbf{T}_s(\mathbf{x}_i^{(s)}) = \mathbf{x}_j^{(s')}$ with probability $\mathbb{Q}(\mathbf{X}_s = \mathbf{x}_i^{(s)}, \mathbf{X}_{s'} = \mathbf{x}_j^{(s')})$.

In Proposition A.1 below, we show that a stochastic transport map – a joint distribution $\mathbb{Q}$ (rather than the deterministic transport map) – exists and satisfies $\Delta\mathrm{MDP}(f, \mathbb{Q}) \le \delta$. This is analogous to the result $\Delta\mathrm{MDP}(f, \mathbf{T}_s) \le \delta$ presented in Theorem 3.3. Note that if $n_0 = n_1$, there exist transport maps (i.e., one-to-one mappings or permutations) between $\mathcal{X}_0$ and $\mathcal{X}_1$.

**Proposition A.1.** *Assume the supports of $\mathcal{P}_{f_0}$ and $\mathcal{P}_{f_1}$ share $m(\le n_0, n_1)$ number of common points. Then, for any given group-fair model $f$ such that $\Delta\mathrm{TVDP}(f) \le \delta$, there exists a joint distribution $\mathbb{Q}$ between $\mathbf{X}_0$ and $\mathbf{X}_1$ satisfying $\Delta\mathrm{MDP}(f, \mathbb{Q}) \le \delta$.*

*Proof.* If suffices to show that there exists $\mathbb{Q}$ such that $\Delta\mathrm{MDP}(f, \mathbb{Q}) \le \Delta\mathrm{TVDP}(f)$. Without loss of generality, assume $n_0 \le n_1$ and let $f(\mathbf{x}_i^{(0)}, 0) = f(\mathbf{x}_i^{(1)}, 1)$ for $i \in \{1, \ldots, m\}$ (i.e., the common points).

First, we calculate $\Delta\mathrm{TVDP}(f)$ as follows. Recall the definition of TV for discrete measures: $\mathrm{TV}(\mathcal{P}_{f_0}, \mathcal{P}_{f_1}) = \frac{1}{2}\sum_z |\mathcal{P}_{f_0}(z) - \mathcal{P}_{f_1}(z)|$. For the $m$ number of common points, the sum of differences is $m|\frac{1}{n_0} - \frac{1}{n_1}| = m\frac{n_1 - n_0}{n_0 n_1}$. For the points only in $f(\mathcal{X}_0)$ but not in $f(\mathcal{X}_1)$, the sum of differences is $(n_0 - m)/n_0$. Similarly, for the points only in $f(\mathcal{X}_1)$ but not in $f(\mathcal{X}_0)$, the sum of differences is $(n_1 - m)/n_1$. As a result, we have

$$\Delta\mathrm{TVDP}(f) = \mathrm{TV}(\mathcal{P}_{f_0}, \mathcal{P}_{f_1}) = \frac{1}{2}\left(m\frac{n_1 - n_0}{n_0 n_1} + \frac{n_0 - m}{n_0} + \frac{n_1 - m}{n_1}\right) = 1 - \frac{m}{n_1}. \tag{13}$$

Then, we construct $\mathbb{Q}$ as follows. Let $\gamma_{i,j} = \mathbb{Q}(\mathbf{X}_0 = \mathbf{x}_i^{(0)}, \mathbf{X}_1 = \mathbf{x}_j^{(1)})$ and $\Gamma = [\gamma_{i,j}]_{i \in [n_0], j \in [n_1]} \in \mathbb{R}_+^{n_0 \times n_1}$ be a matrix based on $\gamma_{i,j}$s, which we construct as below:

1. Build a diagonal matrix $\Gamma_{11} = \frac{1}{n_1}\mathbf{I}_m \in \mathbb{R}_+^{m \times m}$, where $\mathbf{I}_m$ is the identity matrix of size $m \times m$.

2. Build a zero matrix $\Gamma_{21} = \mathbf{0}_{(n_0 - m) \times m} \in \mathbb{R}_+^{(n_0 - m) \times m}$, where $\mathbf{0}$ denotes the zero matrix.

3. Build matrices $\Gamma_{12} \in \mathbb{R}_+^{m \times (n_1 - m)}$ and matrix $\Gamma_{22} \in \mathbb{R}_+^{(n_0 - m) \times (n_1 - m)}$ satisfying $\Gamma_{12}\mathbf{1}_{n_1 - m} = \left(\frac{1}{n_0} - \frac{1}{n_1}\right)\mathbf{1}_m$, $\Gamma_{22}\mathbf{1}_{n_1 - m} = \frac{1}{n_0}\mathbf{1}_{n_0 - m}$, and $\begin{pmatrix}\Gamma_{12} \\ \Gamma_{22}\end{pmatrix}^\top \mathbf{1}_{n_0} = \frac{1}{n_1}\mathbf{1}_{n_1 - m}$.

4. Complete $\Gamma \coloneqq \begin{pmatrix}\Gamma_{11} & \Gamma_{12} \\ \Gamma_{21} & \Gamma_{22}\end{pmatrix} = \begin{pmatrix}\frac{1}{n_1}\mathbf{I}_m & \Gamma_{12} \\ \mathbf{0}_{(n_0 - m) \times m} & \Gamma_{22}\end{pmatrix}$.

Last, note that the $\Gamma$ is a coupling matrix (i.e., satisfying the constraints $\sum_{i=1}^{n_0} \gamma_{i,j} = 1/n_0, \forall j \in [n_1]$ and $\sum_{j=1}^{n_1} \gamma_{i,j} = 1/n_1, \forall i \in [n_0]$). Hence, we can define the joint distribution $\mathbb{Q}$ by the constructed $\Gamma$ (i.e., $\Gamma$ is a

matrix representation of $\mathbb{Q}$, see equation (5) for the formulation). Finally, we have

$$
\begin{aligned}
\Delta\mathrm{MDP}(f,\mathbb{Q}) &= \mathbb{E}_{(\mathbf{X}_0,\mathbf{X}_1)\sim\mathbb{Q}}|f(\mathbf{X}_0,0)-f(\mathbf{X}_1,1)|\\
&= \sum_{i,j}\mathbb{Q}(\mathbf{X}_0=\mathbf{x}_i^{(0)},\mathbf{X}_1=\mathbf{x}_j^{(1)})|f(\mathbf{X}_0,0)-f(\mathbf{X}_1,1)| = \sum_{i,j}\gamma_{i,j}|f(\mathbf{X}_0,0)-f(\mathbf{X}_1,1)|\\
&= \sum_{i\in[m],j\in[m]}\gamma_{i,j}|f(\mathbf{X}_0,0)-f(\mathbf{X}_1,1)| + \sum_{i,\in\{m+1,\dots,n_0\},j\in[m]}\gamma_{i,j}|f(\mathbf{X}_0,0)-f(\mathbf{X}_1,1)|\\
&\quad + \sum_{i\in[n_0],j\in\{m+1,\dots,n_1\}}\gamma_{i,j}|f(\mathbf{X}_0,0)-f(\mathbf{X}_1,1)|\\
&= \sum_{i\in[n_0],j\in\{m+1,\dots,n_1\}}\gamma_{i,j}|f(\mathbf{X}_0,0)-f(\mathbf{X}_1,1)| \le \sum_{i\in[n_0],j\in\{m+1,\dots,n_1\}}\gamma_{i,j} = 1-\frac{m}{n_1} \le \Delta\mathrm{TVDP}(f).
\end{aligned}
\tag{14}
$$

Therefore, any $\mathbb{Q}$ constructed by the steps 1-4 above satisfies $\Delta\mathrm{MDP}(f,\mathbb{Q})\le\delta$. $\square$

**Proposition 3.5.** Let $\mathbf{T}^f$ be the fair matching function of $f$ on $\mathcal{D}_0$ and $\mathcal{D}_1$ (where the empirical distributions with respect to $\mathcal{D}_0$ and $\mathcal{D}_1$ are used in Definition 3.4). Then, the matched individual $\mathbf{T}^f(\mathbf{x})$ of any $\mathbf{x}\in\mathcal{D}_s$ is obtained by $\mathbf{T}^f(\mathbf{x}) = f_{s'}^{-1}\circ F_{s'}^{-1}\circ F_s\circ f_s(\mathbf{x})$.

*Proof of Theorem 3.5.* First, we aim to find a permutation map $M^\ast = \arg\min_M \sum_{i=1}^m |f_s(\mathbf{x}_i) - M(f_{s'}(\mathbf{x}_i))|$ subject to $\{M(f_s(\mathbf{x}_i)):\mathbf{x}_i\in\mathcal{D}_s\} = \{f_{s'}(\mathbf{x}_j):\mathbf{x}_j\in\mathcal{D}_{s'}\}$. By Rachev & Rüschendorf (1998); Chzhen et al. (2020); Jiang et al. (2020b), we have the fact that 1-Wasserstein distance is equivalent to the average of difference between two prediction scores that have same quantiles in each group. That is, $M^\star(y) = F_{s'}^{-1}\circ F_s(y)$ for any $y\in\{f_s(\mathbf{x}):\mathbf{x}\in\mathcal{D}_s\}$.

Second, we have $M^\star$ is one-to-one and $\{f_{s'}(\mathbf{x}_j):\mathbf{x}_j\in\mathcal{D}_{s'}\} = \{f_s(\mathbf{T}^f(\mathbf{x}_i)):\mathbf{x}_i\in\mathcal{D}_s\} = \{M^\star(f_s(\mathbf{x}_i)):\mathbf{x}_i\in\mathcal{D}_s\}$, since $\mathbf{T}^f$ is also an exact one-to-one map.

Therefore, we conclude that $\mathbf{T}^f(\mathbf{x}) = f_{s'}^{-1}\circ M^\star\circ f_s(\mathbf{x}) = f_{s'}^{-1}\circ F_{s'}^{-1}\circ F_s\circ f_s(\mathbf{x})$ for all $\mathbf{x}\in\mathcal{D}_s$. $\square$

**Remark A.2.** *Using Proposition 3.5 to estimate the fair matching function is equivalent to estimating the OT map between two score distributions. The statistical convergence rate for estimating the OT map between one-dimensional distributions is fast; it is of the order $\mathcal{O}(|\mathcal{D}_0|^{-1/2}+|\mathcal{D}_1|^{1/2}) = \mathcal{O}(n_0^{-1/2}+n_1^{1/2})$ (Chizat et al., 2020; Deb et al., 2021; Hütter & Rigollet, 2021). When we use two mini-batches $\tilde{\mathcal{D}}_0\subset\mathcal{D}_0$ and $\tilde{\mathcal{D}}_1\subset\mathcal{D}_1$ with identical size $m$, the order becomes $\mathcal{O}(m^{-1/2})$. This indicates that sufficiently large mini-batch size $m$ can guarantee accurate estimation.*

**Theorem 3.7** For a given $\mathbf{T}_s \in \mathcal{T}_s^{\text{trans}}$, if $\Delta\text{MDP}(f, \mathbf{T}_s) \leq \delta$, then we have $\Delta\text{WDP}(f) \leq \delta$ and $\Delta\overline{\text{DP}}(f) \leq \delta$.

*Proof of Theorem 3.7.* Fix $f \in \{f \in \mathcal{F} : \mathbb{E}_s|f(\mathbf{X}, s) - f(\mathbf{T}_s(\mathbf{X}), s')| \leq \delta\}$. Let $\mathcal{L}_1$ the set of all 1-Lipschitz functions. Using the fact that Wasserstein-1 distance is equivalent to IPM induced by set of 1-Lipschitz function (Villani, 2008), we have that

$$
\begin{aligned}
\Delta\text{WDP}(f) &= \mathcal{W}\left(\mathcal{P}_{f(\mathbf{X},0)|S=0}, \mathcal{P}_{f(\mathbf{X},1)|S=1}\right) \\
&= \sup_{u \in \mathcal{L}_1} |\mathbb{E}_s(u \circ f(\mathbf{X}, s)) - \mathbb{E}_{s'}(u \circ f(\mathbf{X}, s'))| \\
&\leq \sup_{u \in \mathcal{L}_1} |\mathbb{E}_s(u \circ f(\mathbf{X}, s)) - \mathbb{E}_s(u \circ f(\mathbf{T}_s(\mathbf{X}), s'))| + \sup_{u \in \mathcal{L}_1} |\mathbb{E}_s(u \circ f(\mathbf{T}_s(\mathbf{X}), s')) - \mathbb{E}_{s'}(u \circ f(\mathbf{X}, s'))| \\
&\leq \sup_{u \in \mathcal{L}_1} \mathbb{E}_s |u \circ f(\mathbf{X}, s) - u \circ f(\mathbf{T}_s(\mathbf{X}), s')| + \sup_{u \in \mathcal{L}_1} |\mathbb{E}_s(u \circ f(\mathbf{T}_s(\mathbf{X}), s')) - \mathbb{E}_{s'}(u \circ f(\mathbf{X}, s'))| \\
&\overset{u \in \mathcal{L}_1}{\leq} \mathbb{E}_s |f(\mathbf{X}, s) - f(\mathbf{T}_s(\mathbf{X}), s')| + \sup_{u \in \mathcal{L}_1} |\mathbb{E}_s(u \circ f(\mathbf{T}_s(\mathbf{X}), s')) - \mathbb{E}_{s'}(u \circ f(\mathbf{X}, s'))| \\
&\leq \delta + \sup_{u \in \mathcal{L}_1} |\mathbb{E}_s(u \circ f(\mathbf{T}_s(\mathbf{X}), s')) - \mathbb{E}_{s'}(u \circ f(\mathbf{X}, s'))| \\
&\leq \delta + \sup_{f \in \mathcal{F}} \sup_{u \in \mathcal{L}_1} |\mathbb{E}_s(u \circ f(\mathbf{T}_s(\mathbf{X}), s')) - \mathbb{E}_{s'}(u \circ f(\mathbf{X}, s'))| \\
&\leq \delta + \text{TV}(\mathbf{T}_{s\#}\mathcal{P}_s, \mathcal{P}_{s'}) = \delta.
\end{aligned}
$$

$$(15)$$

The last equality holds since $\text{TV}(\mathbf{T}_{s\#}\mathcal{P}_s, \mathcal{P}_{s'}) = 0$ for any transport map $\mathbf{T}_s$.
For $\Delta\overline{\text{DP}}(f)$, because the identity map is 1-Lipschitz, we have that $\Delta\overline{\text{DP}}(f) \leq \Delta\text{WDP}(f)$, which completes the proof. $\square$

**Theorem 4.2** Suppose $\mathcal{F}$ is the collection of $L$-Lipschitz functions. Let $A$ be a given subset in $\mathcal{X}$. Then, for all $f$ satisfying $\Delta\mathrm{MDP}(f, \mathbf{T}_s^f) \leq \delta$, we have

$$\Delta\overline{\mathrm{DP}}_A(f) \leq L\left(\mathbb{E}_s\|\mathbf{X} - \mathbf{T}_s^f(\mathbf{X})\|^2\right)^{\frac{1}{2}} + \mathrm{TV}(\mathcal{P}_{0,A}, \mathcal{P}_{1,A}) + U\delta, \tag{16}$$

where $\mathcal{P}_{s,A}$ is the distribution of $\mathbf{X}|S = s, \mathbf{X} \in A$, and $U > 0$ is a constant only depending on $A$ and $\mathcal{P}_s, s = 0, 1$.

*Proof.* We write $\mathbf{T}_s = \mathbf{T}_s^f$ for notational simplicity.

$$
\begin{aligned}
&|\mathbb{E}(f(\mathbf{X}, 0)|S = 0, \mathbf{X} \in A) - \mathbb{E}(f(\mathbf{X}, 1)|S = 1, \mathbf{X} \in A)| \\
&\leq |\mathbb{E}(f(\mathbf{X}, 0)|S = 1, \mathbf{X} \in A) - \mathbb{E}(f(\mathbf{T}_1(\mathbf{X}), 0)|S = 1, \mathbf{X} \in A)| \\
&+ |\mathbb{E}(f(\mathbf{T}_1(\mathbf{X}), 0)|S = 1, \mathbf{X} \in A) - \mathbb{E}(f(\mathbf{X}, 1)|S = 1, \mathbf{X} \in A)| \\
&+ |\mathbb{E}(f(\mathbf{X}, 0)|S = 0, \mathbf{X} \in A) - \mathbb{E}(f(\mathbf{X}, 0)|S = 1, \mathbf{X} \in A)|.
\end{aligned}
\tag{17}
$$

By (C1), the first term is bounded by $L\mathbb{E}_1\|\mathbf{X} - \mathbf{T}_1(\mathbf{X})\|$, which is also bounded by $L\left(\mathbb{E}_1\|\mathbf{X} - \mathbf{T}_1(\mathbf{X})\|^2\right)^{1/2}$. The second term is bounded by $\delta$ up to a constant for all $f$ satisfying $\Delta\mathrm{MDP}(f, \mathbf{T}_s) \leq \delta$. That is, we have

$$
\begin{aligned}
&|\mathbb{E}(f(\mathbf{T}_1(\mathbf{X}), 0)|S = 1, \mathbf{X} \in A) - \mathbb{E}(f(\mathbf{X}, 1)|S = 1, \mathbf{X} \in A)| \\
&= \left|\frac{\int f(\mathbf{T}_1(\mathbf{X}), 0)\mathbb{I}(\mathbf{X} \in A)d\mathcal{P}_1(\mathbf{X})}{\int \mathbb{I}(\mathbf{X} \in A)d\mathcal{P}_1(\mathbf{X})} - \frac{\int f(\mathbf{X}, 1)\mathbb{I}(\mathbf{X} \in A)d\mathcal{P}_1(\mathbf{X})}{\int \mathbb{I}(\mathbf{X} \in A)d\mathcal{P}_1(\mathbf{X})}\right| \\
&\leq \frac{1}{\int \mathbb{I}(\mathbf{X} \in A)d\mathcal{P}_1(\mathbf{X})} \int_{\mathbf{X} \in A} |f(\mathbf{T}_1(\mathbf{X}), 0) - f(\mathbf{X}, 1)|d\mathcal{P}_1(\mathbf{X}) \\
&\leq \frac{1}{\int \mathbb{I}(\mathbf{X} \in A)d\mathcal{P}_1(\mathbf{X})} \int_{\mathbf{X} \in \mathcal{X}} |f(\mathbf{T}_1(\mathbf{X}), 0) - f(\mathbf{X}, 1)|d\mathcal{P}_1(\mathbf{X}) \\
&= U'(A, \mathcal{P}_1) \times \mathbb{E}_1|f(\mathbf{T}_1(\mathbf{X}), 0) - f(\mathbf{X}, 1)| \\
&\leq U'(A, \mathcal{P}_1) \times \delta
\end{aligned}
\tag{18}
$$

where $U'(A, \mathcal{P}_1) = 1/\int \mathbb{I}(\mathbf{X} \in A)d\mathcal{P}_1(\mathbf{X}) = 1/\mathcal{P}(\mathbf{X} \in A|S = 1)$ is a constant only depending on $\mathcal{P}_1$ and $A$.

The third term $|\mathbb{E}(f(\mathbf{X}, 0)|S = 0, \mathbf{X} \in A) - \mathbb{E}(f(\mathbf{X}, 0)|S = 1, \mathbf{X} \in A)|$ is not controllable by either the transport map or $\delta$ but depends on the given distributions and $A$. That is,

$$
\begin{aligned}
&|\mathbb{E}(f(\mathbf{X}, 0)|S = 0, \mathbf{X} \in A) - \mathbb{E}(f(\mathbf{X}, 0)|S = 1, \mathbf{X} \in A)| \\
&= \left|\int_A f(\mathbf{X}, 0)d\mathcal{P}_0(\mathbf{X}) - \int_A f(\mathbf{X}, 0)d\mathcal{P}_1(\mathbf{X})\right| \\
&\leq \sup_{f \in \mathcal{F}} \left|\int_A f(\mathbf{X}, 0)d\mathcal{P}_0(\mathbf{X}) - \int_A f(\mathbf{X}, 0)d\mathcal{P}_1(\mathbf{X})\right| \\
&\leq \mathrm{TV}(\mathcal{P}_{0,A}, \mathcal{P}_{1,A}).
\end{aligned}
\tag{19}
$$

Hence, we have

$$
\begin{aligned}
&|\mathbb{E}(f(\mathbf{X}, 0)|S = 0, \mathbf{X} \in A) - \mathbb{E}(f(\mathbf{X}, 1)|S = 1, \mathbf{X} \in A)| \\
&\leq L(\mathbb{E}_1\|\mathbf{X} - \mathbf{T}_1(\mathbf{X})\|^2)^{1/2} + \mathrm{TV}(\mathcal{P}_{0,A}, \mathcal{P}_{1,A}) + U'(A, \mathcal{P}_1)\delta.
\end{aligned}
\tag{20}
$$

We can similarly derive

$$
\begin{aligned}
&|\mathbb{E}(f(\mathbf{X}, 0)|S = 0, \mathbf{X} \in A) - \mathbb{E}(f(\mathbf{X}, 1)|S = 1, \mathbf{X} \in A)| \\
&\leq L(\mathbb{E}_0\|\mathbf{X} - \mathbf{T}_0(\mathbf{X})\|^2)^{1/2} + \mathrm{TV}(\mathcal{P}_{0,A}, \mathcal{P}_{1,A}) + U'(A, \mathcal{P}_0)\delta.
\end{aligned}
\tag{21}
$$

Letting $U := \max\{U'(A, \mathcal{P}_0), U'(A, \mathcal{P}_1)\}$ completes the proof. $\square$

**Lemma A.3** (Optimal transport map between two Gaussians). *For mean vectors $\mu_{\mathbf{X}}, \mu_{\mathbf{Y}} \in \mathbb{R}^d$ and covariance matrices $\Sigma_{\mathbf{X}}, \Sigma_{\mathbf{Y}} \in \mathbb{R}^{d \times d}$, the OT map from $\mathcal{N}(\mu_{\mathbf{X}}, \Sigma_{\mathbf{X}})$ to $\mathcal{N}(\mu_{\mathbf{Y}}, \Sigma_{\mathbf{Y}})$ is given as $\mathbf{T}^{OT}(\mathbf{x}) = \mathbf{W}^{OT}\mathbf{x} + \mathbf{b}^{OT}$ where $\mathbf{W}^{OT} = \Sigma_{\mathbf{X}}^{-\frac{1}{2}} \left( \Sigma_{\mathbf{X}}^{\frac{1}{2}} \Sigma_{\mathbf{Y}} \Sigma_{\mathbf{X}}^{\frac{1}{2}} \right)^{\frac{1}{2}} \Sigma_{\mathbf{X}}^{-\frac{1}{2}}$ and $\mathbf{b}^{OT} = \mu_{\mathbf{Y}} - \mathbf{W}^{OT}\mu_{\mathbf{X}}$.*

*Proof.* Consider the centered Gaussians, i.e., $\mu_{\mathbf{X}} = \mu_{\mathbf{Y}}$ at first. Based on Theorem 4 of Olkin & Pukelsheim (1982), we have that $\mathcal{W}_2 \left( \mathcal{N}(0, \Sigma_{\mathbf{X}}), \mathcal{N}(0, \Sigma_{\mathbf{Y}}) \right) = Tr(\Sigma_{\mathbf{X}} + \Sigma_{\mathbf{Y}} - 2 \left( \Sigma_{\mathbf{X}}^{1/2} \Sigma_{\mathbf{Y}} \Sigma_{\mathbf{X}}^{1/2} \right)^{1/2}) = \|\Sigma_{\mathbf{X}}^{1/2} - \Sigma_{\mathbf{Y}}^{1/2}\|_F^2$ where $\|\cdot\|$ is the Frobenius norm. Correspondingly, Knott & Smith (1984) derived the optimal transport map as $\mathbf{x} \mapsto \Sigma_{\mathbf{X}}^{-1/2} \left( \Sigma_{\mathbf{X}}^{-1/2} \Sigma_{\mathbf{Y}} \Sigma_{\mathbf{X}}^{1/2} \right)^{1/2} \Sigma_{\mathbf{X}}^{-1/2} \mathbf{x}$.

Combining these results, we can extend the OT map formula of Gaussians with nonzero means as follows. Since $\mathbb{E}\|\mathbf{X} - \mathbf{Y}\|^2 = \mathbb{E}\|(\mathbf{X} - \mu_{\mathbf{X}}) - (\mathbf{Y} - \mu_{\mathbf{Y}}) + (\mu_{\mathbf{X}} - \mu_{\mathbf{Y}})\|^2 = \mathbb{E}\|(\mathbf{X} - \mu_{\mathbf{X}}) - (\mathbf{Y} - \mu_{\mathbf{Y}})\|^2 + \|\mu_{\mathbf{X}} - \mu_{\mathbf{Y}}\|^2$, the Wasserstein distance is given as $\mathbf{W}_2 \left( \mathcal{N}(\mu_{\mathbf{X}}, \Sigma_{\mathbf{X}}), \mathcal{N}(\mu_{\mathbf{Y}}, \Sigma_{\mathbf{Y}}) \right) = \|\mu_{\mathbf{X}} - \mu_{\mathbf{Y}}\|^2 + \|\Sigma_{\mathbf{X}}^{1/2} - \Sigma_{\mathbf{Y}}^{1/2}\|_F^2$, and so the corresponding optimal transport map is also given as $\mathbf{x} \mapsto \Sigma_{\mathbf{X}}^{-1/2} \left( \Sigma_{\mathbf{X}}^{-1/2} \Sigma_{\mathbf{Y}} \Sigma_{\mathbf{X}}^{1/2} \right)^{1/2} \Sigma_{\mathbf{X}}^{-1/2} \mathbf{x} + \mu_{\mathbf{Y}} - \Sigma_{\mathbf{X}}^{-1/2} \left( \Sigma_{\mathbf{X}}^{-1/2} \Sigma_{\mathbf{Y}} \Sigma_{\mathbf{X}}^{1/2} \right)^{1/2} \Sigma_{\mathbf{X}}^{-1/2} \mu_{\mathbf{X}}$, which completes the proof. $\square$

**Proposition 4.3** (Counterfactual fairness and the OT map) For all $A$ having $(\mathbf{I}_d - A)^{-1}$, $\mathbf{W}_s$ becomes $\sigma_{s'} \sigma_s^{-1} \mathbf{I}_d$. That is, $\tilde{\mathbf{x}}_s^{\mathrm{CF}} = \tilde{\mathbf{x}}_s^{\mathrm{OT}}$.

*Proof of Proposition 4.3.* Once we observe $\mathbf{x}_0$, the randomness $\epsilon_0$ is observed as $\epsilon_0 = B^{-1}\mathbf{x}_0 - \mu_0$. By replacing the sensitive attribute on the randomness $\epsilon_0$, we obtain $\sigma_1^{-1}(B^{-1}\tilde{\mathbf{x}}_0^{\mathrm{CF}} - \mu_1) = \sigma_0^{-1}(B^{-1}\mathbf{x}_0 - \mu_0)$. Then, its counterfactual becomes $\tilde{\mathbf{x}}_0^{\mathrm{CF}} = B\mu_1 + \sigma_1 \sigma_0^{-1}\mathbf{I}_d(\mathbf{x}_0 - B\mu_0)$. Then, we prove Proposition 4.3 by showing the if and only if condition as follows.

$$
\begin{aligned}
\mathbf{W}_0 &= (\sigma_0^2 BDB^\top)^{-1/2} \left( (\sigma_0^2 BDB^\top)^{1/2} \sigma_1^2 BDB^\top (\sigma_0^2 BDB^\top)^{1/2} \right)^{1/2} (\sigma_0^2 BDB^\top)^{-1/2} \\
&= \sigma_1 \sigma_0^{-1} (BDB^\top)^{-1/2} \left( (BDB^\top)^{1/2} BDB^\top (BDB^\top)^{1/2} \right)^{1/2} (BDB^\top)^{-1/2} \\
&= \sigma_1 \sigma_0^{-1} (BDB^\top)^{-1/2} \left( (BDB^\top)^2 \right)^{1/2} (BDB^\top)^{-1/2} \\
&= \sigma_1 \sigma_0^{-1} \mathbf{I}_d.
\end{aligned}
\tag{22}
$$

The same result can be done for $\mathbf{x}_1$. Hence, we conclude $\mathbf{W}_s = \sigma_{s'} \sigma_s^{-1} \mathbf{I}_d$. $\square$

## B  Disadvantage of high transport cost

This section presents an example of two completely different group-fair models where one is unreasonable and the other is reasonable, particularly in terms of subset fairness. This example suggests that *not all group-fair models are acceptable*, thereby emphasizing the necessity of finding group-fair models corresponding to favorable implicit transport maps.

Suppose that the distribution of the input variable is given as $\mathbf{X}|S = s \sim \text{Unif}(0, 1)$, for $s \in \{0, 1\}$. Consider the following two classification models: $\hat{f}(\mathbf{x}, s) = \frac{\text{sign}(2\mathbf{x}-1)(1-2s)+1}{2}$ and $\tilde{f}(\mathbf{x}, s) = \frac{\text{sign}(2\mathbf{x}-1)+1}{2}$.

It is clear that both $\hat{f}$ and $\tilde{f}$ are perfectly fair, i.e., $\mathcal{P}_{\hat{f}_0} = \mathcal{P}_{\hat{f}_1}$ and $\mathcal{P}_{\tilde{f}_0} = \mathcal{P}_{\tilde{f}_1}$. However, $\hat{f}$ has a notable unfairness issue in its treatments of individuals within the subset $\{\mathbf{x} \geq 1/2\}$ (as well as $\{\mathbf{x} < 1/2\}$); for when $\mathbf{x} > 1/2$, $\hat{f}$ assigns label 1 for all individuals of $s = 0$ while it assigns label 0 for all individuals of $s = 1$. This indicates that $\hat{f}$ discriminates against individuals in the subset $\{\mathbf{x} \geq 1/2\}$ (and also $\{\mathbf{x} < 1/2\}$). In contrast, $\tilde{f}$ does not exhibit such undesirable discrimination against the subsets. Hence, we can say that $\hat{f}$ has less discrimination on the subsets than $\hat{f}$. Figure 7 provides a comparative illustration of $\hat{f}$ and $\tilde{f}$.

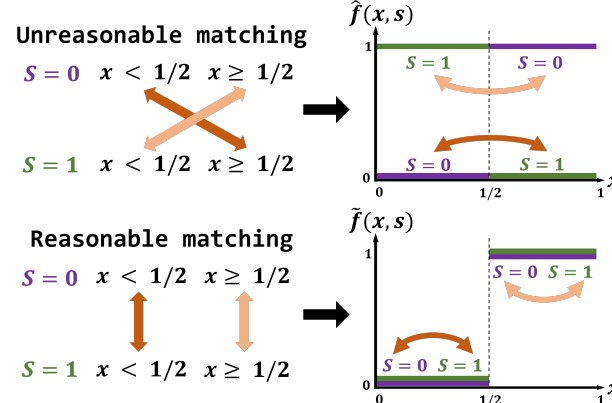

Figure 7: (Top) A group-fair model with the risk of discrimination on subsets. (Bottom) A group-fair model without the risk of discrimination on subsets.

The observed discrimination of $\hat{f}$ on subsets can be attributed to the unreasonable fair matching function of $\hat{f}$. It turns out that the fair matching function of $\hat{f}$ is $\mathbf{T}^{\hat{f}}(\mathbf{x}) = \mathbf{x} - \frac{\text{sign}((2\mathbf{x}-1)(1-2s))}{2}$. This function matches an individual in $\{\mathbf{x} < \frac{1}{2}, S = s\}$ with one in $\{\mathbf{x} \geq \frac{1}{2}, S = s'\}$, who are far apart from each other. In contrast, the fair matching function of $\tilde{f}$ is $\mathbf{T}^{\tilde{f}}(\mathbf{x}) = \mathbf{x}$. This example emphasizes the need of group-fair models whose fair matching function have low transport costs.

## C  The Kantorovich problem

As shortly introduced in Section 2.4, the Kantorovich problem is to find the optimal coupling (i.e., joint distribution) between two given distributions. It can be mathematically expressed as the following. For two given distributions $\mathcal{Q}_1$ and $\mathcal{Q}_2$, $\inf_{\pi \in \Pi(\mathcal{Q}_1, \mathcal{Q}_2)} \mathbb{E}_{\mathbf{X}, \mathbf{Y} \sim \pi}(c(\mathbf{X}, \mathbf{Y}))$ where $\Pi(\mathcal{Q}_1, \mathcal{Q}_2)$ is the set of all joint measures with marginals $\mathcal{Q}_1$ and $\mathcal{Q}_2$. Let $c$ be the $L_2$ cost function.

For given two empirical distributions on $\mathcal{D}_0 = \{\mathbf{x}_i^{(0)}\}_{i=1}^{n_0}$ and $\mathcal{D}_1 = \{\mathbf{x}_j^{(1)}\}_{j=1}^{n_1}$, we first define the cost matrix as $\mathbf{C} := [c_{i,j}] \in \mathbb{R}_+^{n_0 \times n_1}$, where $c_{i,j} = \|\mathbf{x}_i^{(0)} - \mathbf{x}_j^{(1)}\|^2$. Then, the solution of the Kantorovich problem, i.e., optimal joint distribution between the two distributions, is defined by the coupling matrix $\Gamma = [\gamma_{i,j}] \in \mathbb{R}_+^{n_0 \times n_1}$, which is the minimizer of the following objective:

$$\min_{\Gamma} \|\mathbf{C} \odot \Gamma\|_1 = \min_{\gamma_{i,j}} c_{i,j}\gamma_{i,j} \text{ s.t. } \gamma_{i,j} \geq 0, \sum_{i=1}^{n_0} \gamma_{i,j} = \frac{1}{n_1}, \sum_{j=1}^{n_1} \gamma_{i,j} = \frac{1}{n_0}. \tag{23}$$

In fact, the Kantorovich problem can be solved by linear programming (Kantorovich, 2006; Villani, 2008). To solve the linear program, we use practical implementation such as `POT` library in `Python`.

# D    Implementation details

In this section, we provide detailed descriptions for the implementation of the experiments.

## D.1    Datasets

First, the URLs of these datasets are provided.

- ADULT: the Adult income dataset (Dua & Graff, 2017) can be downloaded from the UCI repository[3].

- GERMAN: the German credit dataset (Dua & Graff, 2017) can be downloaded from the UCI repository[4].

- DUTCH: the Dutch census dataset can be downloaded from the public Github of Quy et al. (2022) [5].

- BANK: the Bank marketing dataset can be downloaded from the UCI repository[6].

The following Table 6 describes the basic information of the four datasets.

Table 6: The description of the real benchmark tabular datasets: ADULT, GERMAN, BANK, and DUTCH. $\mathbf{X}, S, Y$ and $d$ denote the input features, the sensitive attribute, the target label information, and the dimension of $\mathbf{X}$, respectively. Train/Test sizes are the number of samples in the training and test datasets, respectively.

| Dataset ‖ | Variable | Description | Dataset ‖ | Variable | Description |
|---|---|---|---|---|---|
| ADULT | $\mathbf{X}$ $S$ $Y$ $d$ Train/Test size | Personal attributes Gender Outcome over \$50$k$ 101 30,136/15,086 | GERMAN | $\mathbf{X}$ $S$ $Y$ $d$ Train/Test data size | Personal attributes Gender High credit score 60 800/200 |
| BANK | $\mathbf{X}$ $S$ $Y$ $d$ Train/Test size | Personal attributes Binarized age Subscribing a term deposit 57 24,390/6,098 | DUTCH | $\mathbf{X}$ $S$ $Y$ $d$ Train/Test data size | Personal attributes Gender High-level occupation 58 48,336/12,084 |

Second, we describe pre-processing method of the datasets used. For ADULT, GERMAN, and BANK datasets, we follow the pre-processing of the implementation of IBM's AIF360 (Bellamy et al., 2018) [7]. For DUTCH dataset, we follow the pre-processing of Quy et al. (2022)'s Github[8]. Basically, continuous input variables are normalized by min-max scaling and categorical input variables are one-hot encoded. We set batch size as 1024, 200, 1024, and 512 for ADULT, GERMAN, DUTCH, and BANK datasets, respectively.

## D.2    Algorithms

This section provides more detailed descriptions of the baseline algorithms used in our experiments.

- Reduction (Agarwal et al., 2018): This algorithm is an in-processing method that learns a fair classifier with the lowest empirical fairness level $\Delta$DP. To implement this method for MLP model architecture, we employ FairTorch[9]. It minimizes cross-entropy $+ \lambda \cdot$ Reduction regularizer for a given $\lambda > 0$.

- Reg (Donini et al., 2018; Chuang & Mroueh, 2021): This method is a regularizing approach that minimizes cross-entropy $+ \lambda \cdot \Delta\overline{\text{DP}}^2$ for a given $\lambda > 0$. In Chuang & Mroueh (2021), they call this algorithm GapReg. This is also similar to the approach of Donini et al. (2018) in the sense that the model is learned with a constraint having a given level of $\Delta\overline{\text{DP}}$.

---

[3]https://archive.ics.uci.edu/ml/datasets/adult
[4]https://archive.ics.uci.edu/ml/datasets/statlog+(german+credit+data)
[5]https://github.com/tailequy/fairness_dataset/tree/main/experiments/data/dutch.csv
[6]https://archive.ics.uci.edu/dataset/222/bank+marketing
[7]https://aif360.readthedocs.io/en/stable/
[8]https://github.com/tailequy/fairness_dataset/tree/main/experiments/data/
[9]https://github.com/wbawakate/fairtorch

- Adv (Zhang et al., 2018): This algorithm is an in-processing method that regularizes the model outputs with an adversarial network so that the adversarial network is learned to predict the sensitive attribute using the model outputs as the inputs. It minimizes cross-entropy $+ \lambda \cdot$ Adversarial loss for a given $\lambda > 0$.

Note that Reduction and Adv are ones of the most popular in-processing algorithms, as widely-used libraries AIF360 (Bellamy et al., 2018) and Fairlearn (Bird et al., 2020) provide the usage and implementation of the two algorithms. Reg is a vanilla approach of adding the regularization term in the loss function to learn the most accurate model among models satisfying a given level of group fairness.

We basically train models with various fairness levels by controlling the Lagrangian multiplier $\lambda$. The values are presented in the following table.

Table 7: Hyper-parameters used for controlling fairness levels for each algorithm.

| Algorithm | $\lambda$ |
|---|---|
| Reduction | $\{0.5, 1.0, 2.0, 3.0, 4.0, 5.0, 6.0, 8.0, 10.0, 20.0, 30.0, 40.0, 50.0, 60.0, 80.0, 100.0, 150.0, 200.0, 300.0, 500.0\}$ |
| Reg | $\{0.1, 0.2, 0.3, 0.4, 0.5, 0.6, 0.7, 0.8, 0.9, 1.0, 1.2, 1.5, 1.8, 2.0, 3.0, 5.0, 10.0, 20.0, 50.0, 100.0\}$ |
| Adv | $\{0.1, 0.2, 0.3, 0.4, 0.5, 0.6, 0.7, 0.8, 0.9, 1.0, 1.1, 1.2, 1.3, 1.5, 2.0, 3.0, 5.0, 10.0, 15.0, 20.0, 30.0, 50.0, 100.0\}$ |
| FTM | $\{0.1, 0.2, 0.3, 0.4, 0.5, 0.6, 0.7, 0.8, 0.9, 1.0, 1.1, 1.2, 1.3, 1.5, 2.0, 3.0, 5.0, 10.0\}$ |

The Adam optimizer (Kingma & Ba, 2014) with an initial learning rate of 0.001 is used, and the learning rate is scheduled by multiplying 0.95 at each epoch.

### D.3  Pseudo-code

Here, we provide a `Pytorch`-style psuedo code of calculating the MDP constraint in FTM.

**Algorithm 1:** `PyTorch`-style pseudo-code of calculating the MDP constraint in FTM.

```
# xs, xt: input vectors from the source, target distribution, respectively.
# model: a classifier to be trained
import ot
# The matching constraint: matching with the OT map
weight_s = torch.ones(size=(xs.size(0), )) / xs.size(0)
weight_t = torch.ones(size=(xt.size(0), )) / xt.size(0) # identical to weight_s
M = ot.dist(xs, xt)
G = ot.emd(weight_s, weight_t, M)
matched_xs = xt[torch.argmax(G, dim=1)]
output, matched_output = model(xs), model(matched_xs)
FTM_REG = (output - matched_output).abs().mean()
```

# E  Auxillary experimental results

In this section, we provide auxillary experimental results that are not displayed in the main body.

## E.1  Fairness-prediction trade-off (Section 5.2.1)

Figure 8 shows the trade-offs between the fairness levels with respect to $\Delta$DP, $\Delta\overline{\text{DP}}$ and classification accuracy.

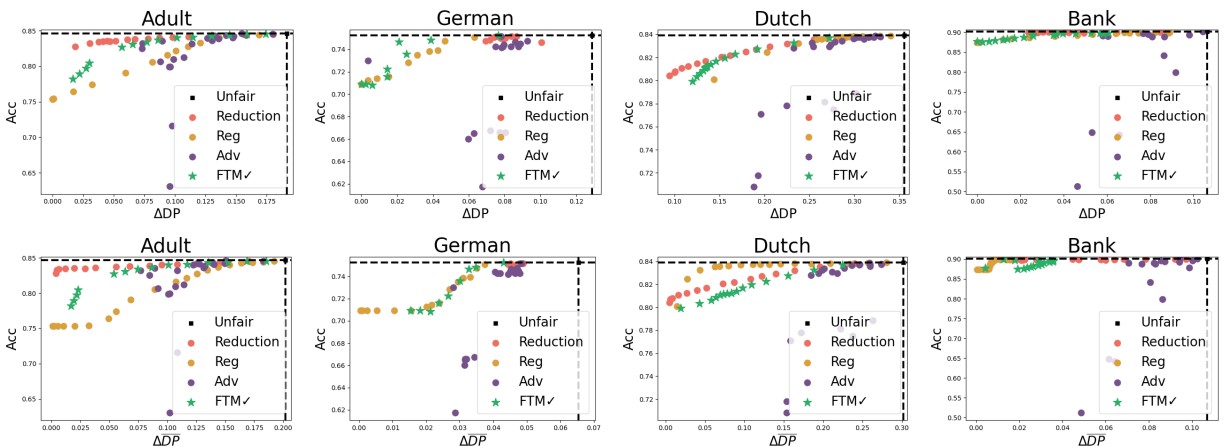

Figure 8: **Fairness-prediction trade-offs**: (Left to right) ADULT, GERMAN, DUTCH, BANK. (Top to bottom) $\Delta$DP vs. Acc, $\Delta\overline{\text{DP}}$ vs. Acc.

## E.2  Improvement in subset fairness (Section 5.2.2)

Here, we provide experimental results showing fairness levels on subsets defined by input variables. Table 8 and 9 are copies of Table 1 with standard errors.

Table 8: **Fairness on subsets defined by the input variable age**: Fairness levels on subsets defined by the input variable age on GERMAN dataset under a given $\Delta\overline{\text{DP}}$ = 0.045 with standard errors (s.e.).

|  | Algorithm ‖ | Reduction | Reg | Adv | FTM ✓ |
|---|---|---|---|---|---|
| High age | $\Delta$DP (s.e.) | 0.073 (0.015) | 0.077 (0.013) | 0.048 (0.020) | **0.045** (0.021) |
|  | $\Delta\overline{\text{DP}}$ (s.e.) | 0.049 (0.006) | 0.029 (0.008) | 0.028 (0.012) | **0.026** (0.006) |
|  | $\Delta$WDP (s.e.) | 0.053 (0.005) | 0.039 (0.003) | 0.042 (0.008) | **0.038** (0.003) |
| Low age | $\Delta$DP (s.e.) | 0.118 (0.035) | 0.116 (0.037) | 0.122 (0.047) | **0.077** (0.032) |
|  | $\Delta\overline{\text{DP}}$ (s.e.) | **0.047** (0.015) | 0.050 (0.009) | 0.053 (0.017) | **0.047** (0.007) |
|  | $\Delta$WDP (s.e.) | 0.058 (0.011) | 0.059 (0.007) | 0.061 (0.015) | **0.054** (0.006) |

Table 9: **Fairness on subsets defined by the input variable marital status**: Fairness levels on subsets defined by the input variable marital status on DUTCH dataset under a given $\Delta\overline{\text{DP}}$ = 0.12 with standard errors (s.e.).

|  | Algorithm ‖ | Reduction | Reg | Adv | FTM ✓ |
|---|---|---|---|---|---|
| Married | $\Delta$DP (s.e.) | 0.258 (0.005) | 0.372 (0.003) | 0.237 (0.083) | **0.204** (0.003) |
|  | $\Delta\overline{\text{DP}}$ (s.e.) | 0.182 (0.002) | 0.164 (0.001) | 0.187 (0.073) | **0.152** (0.002) |
|  | $\Delta$WDP (s.e.) | 0.183 (0.002) | 0.172 (0.001) | 0.193 (0.071) | **0.152** (0.002) |
| Not married | $\Delta$DP (s.e.) | **0.061** (0.007) | 0.131 (0.006) | 0.095 (0.038) | 0.068 (0.005) |
|  | $\Delta\overline{\text{DP}}$ (s.e.) | 0.045 (0.002) | 0.062 (0.003) | 0.098 (0.035) | **0.036** (0.003) |
|  | $\Delta$WDP (s.e.) | **0.045** (0.003) | 0.072 (0.002) | 0.098 (0.034) | **0.045** (0.005) |

### E.3 An additional advantage of using the marginal OT map: reducing the risk of self-fulfilling prophecy

We compare the risks of discrimination in the context of self-fulfilling prophecy in Dwork et al. (2012), a critical limitation that can arise when focusing solely on group fairness: *unqualified individuals with relatively low scores can be chosen to be qualified, while other individuals with relatively high scores are chosen to be unqualified.* To quantify the risk of self-fulfilling prophecy, we assume that the unfair model is optimal for predicting the true score of each individual. We consider the following two evaluation approaches under this assumption. For the transport map used in MDP constraint, we choose the marginal OT map.

**(Evaluation 1)** The first measure for the risk of self-fulfilling prophecy is the *Spearman's rank correlation* between unfair and fair prediction scores at each protected group: a higher rank correlation implies a lower risk of self-fulfilling prophecy. Table 10 shows that FTM has lower risks of suffering from self-fulfilling prophecy, in most cases.

Table 10: Spearman's correlation coefficients between the scores of the unfair model and group-fair models under fixed levels of $\Delta\overline{\mathrm{DP}}$ with standard errors (s.e.). **Bold** faces are the best ones, and underlined ones are the second bests.

| Dataset | ADULT | | GERMAN | |
|---|---|---|---|---|
| $\Delta\overline{\mathrm{DP}}$ | 0.10 | | 0.05 | |
| Sensitive attribute $S$ | 0 | 1 | 0 | 1 |
| Reduction (s.e.) | 0.935 (0.006) | 0.987 (0.001) | 0.996 (0.001) | 0.997 (0.001) |
| Reg (s.e.) | 0.762 (0.087) | 0.806 (0.084) | **0.997** (0.000) | **0.998** (0.000) |
| Adv (s.e.) | 0.876 (0.003) | 0.979 (0.001) | 0.986 (0.009) | 0.986 (0.010) |
| FTM ✓ (s.e.) | **0.968** (0.003) | **0.989** (0.001) | 0.993 (0.002) | 0.995 (0.001) |

| Dataset | DUTCH | | BANK | |
|---|---|---|---|---|
| $\Delta\overline{\mathrm{DP}}$ | 0.01 | | 0.02 | |
| Sensitive attribute $S$ | 0 | 1 | 0 | 1 |
| Reduction (s.e.) | 0.940 (0.001) | 0.922 (0.001) | 0.958 (0.010) | 0.978 (0.005) |
| Reg (s.e.) | 0.872 (0.003) | 0.972 (0.003) | 0.784 (0.031) | 0.974 (0.003) |
| Adv (s.e.) | 0.659 (0.171) | 0.693 (0.185) | 0.603 (0.207) | 0.505 (0.238) |
| FTM ✓ (s.e.) | **0.973** (0.002) | **0.991** (0.000) | **0.964** (0.007) | **0.979** (0.004) |

**(Evaluation 2)** For the second approach, we employ $2 \times 2$ confusion matrices to compare the predicted labels of the unfair and the group-fair models. In specific, in the privileged group $S = 1$, individuals predicted as $\hat{Y} = 0$ (i.e., unqualified) by the unfair model but $\hat{Y} = 1$ (i.e., chosen to be qualified) by the group-fair model are considered as undesirable instances in the context of self-fulfilling prophecy. Likewise, in the unprivileged group $S = 0$, individuals predicted as $\hat{Y} = 1$ by the unfair model but $\hat{Y} = 0$ by the group-fair model are similarly considered undesirable.

That is, for the risk of self-fulfilling prophecy, we count *the number of individuals whose prediction is undesirably flipped* (i.e., # of $\hat{Y} = 0$ (Unfair) $\to \hat{Y} = 1$ (Fair) for $S = 1$, and # of $\hat{Y} = 1$ (Unfair) $\to \hat{Y} = 0$ (Fair) for $S = 0$). Table 11 shows that the undesirable treatments of FTM are less observed than those of baseline methods, in most cases.

Table 11: $2 \times 2$ confusion matrices comparing the predicted labels of the unfair model and the group-fair models. The encircled numbers are the counts of undesirable instances. **Bold** faces are the best ones and underlined ones are the second bests.

| Dataset ($\Delta\overline{\mathrm{DP}}$) | | ADULT (0.05) | | GERMAN (0.05) | | DUTCH (0.15) | | BANK (0.04) | |
|---|---|---|---|---|---|---|---|---|---|
| **$S = 1$** | | Unfair | | | | | | | |
| | | $\hat{Y} = 0$ | $\hat{Y} = 1$ | $\hat{Y} = 0$ | $\hat{Y} = 1$ | $\hat{Y} = 0$ | $\hat{Y} = 1$ | $\hat{Y} = 0$ | $\hat{Y} = 1$ |
| Reduction | $\hat{Y} = 0$ | 6124 | 629 | 98 | 1 | 2170 | 662 | 5220 | 62 |
| | $\hat{Y} = 1$ | ㉒ | 1701 | ①̲ | 32 | ⑧ | 3158 | ㊇ | 579 |
| Reg | $\hat{Y} = 0$ | 6144 | 2198 | 99 | 8 | 2164 | 311 | 5265 | 229 |
| | $\hat{Y} = 1$ | ②̲ | 132 | **⓪** | 25 | ⑭ | 3509 | ⑰̲ | 412 |
| Adv | $\hat{Y} = 0$ | 6121 | 977 | 95 | 0 | 2152 | 1127 | 5255 | 516 |
| | $\hat{Y} = 1$ | ㉕ | 1353 | ④ | 33 | ㉖ | 2693 | ㉗ | 125 |
| FTM ✓ | $\hat{Y} = 0$ | 6146 | 1364 | 99 | 1 | 2174 | 862 | 5279 | 397 |
| | $\hat{Y} = 1$ | **⓪** | 966 | **⓪** | 32 | **④** | 2958 | **③** | 244 |
| **$S = 0$** | | Unfair | | | | | | | |
| | | $\hat{Y} = 0$ | $\hat{Y} = 1$ | $\hat{Y} = 0$ | $\hat{Y} = 1$ | $\hat{Y} = 0$ | $\hat{Y} = 1$ | $\hat{Y} = 0$ | $\hat{Y} = 1$ |
| Reduction | $\hat{Y} = 0$ | 3486 | ⑬̲ | 54 | **③** | 4137 | **⓪** | 129 | ⑭̲ |
| | $\hat{Y} = 1$ | 262 | 341 | 0 | 11 | 226 | 1723 | 1 | 31 |
| Reg | $\hat{Y} = 0$ | 3748 | ⑩④ | 54 | ④ | 4300 | ⑬ | 128 | ㊺ |
| | $\hat{Y} = 1$ | 0 | 250 | 0 | 10 | 63 | 1710 | 2 | 0 |
| Adv | $\hat{Y} = 0$ | 3655 | ㉜ | 53 | **③** | 3917 | ㊄ | 125 | ㉞ |
| | $\hat{Y} = 1$ | 93 | 302 | 1 | 11 | 446 | 1638 | 5 | 11 |
| FTM ✓ | $\hat{Y} = 0$ | 3719 | **⑪** | 54 | **③** | 4217 | **⑥** | 120 | **⑩** |
| | $\hat{Y} = 1$ | 29 | 343 | 0 | 11 | 146 | 1717 | 10 | 35 |

