# OpenReview forum: "Fairness Through Matching"
_TMLR — Accepted by TMLR_

### Review · Reviewer_MW9E · 2024-10-03

**Summary Of Contributions:**

This work showed that the model fairness regarding two sensitive groups, in terms of demographic parity (DP), is connected with the degree of match between the model outputs conditional on these two groups, up to a transport map on the input space. Specifically, the authors proposed Matched Demographic Parity (MDP) as a surrogate of strong DP. A small MDP guarantees a small DP and vice versa. The proposed algorithm Fairness Through Matching (FTM), which optimizes the empirical loss with a MDP penalty, can improve the subset-level fairness by properly specifying the transport map.

**Audience:**

Yes

**Claims And Evidence:**

Yes

**Requested Changes:**

Please see the weakness part above. The most critical issues are:
1. Point (i), what is the particular advantage of FTM compared to other methods? As there is no particular guide to select the optimal transport map, and ad hoc choices such as marginal or joint distribution do not offer better accuracy-fairness trade-off.
2. The assumptions and proofs of all theoretical results should be reviewed throughly.

**Strengths And Weaknesses:**

# Strength

- The authors observe an interesting connection between the DP and optimal transportation. That is, a group-fair model corresponds to an implicit transport map between the input spaces conditional on the sensitive group.
- The main idea is clearly delivered.
- The theoretical results seems to be sound, though the presentation and proof is quite sloppy.

# Weakness

(i) The biggest concern is that, Thm 3.3 and 3.5 actually reflect that the fairness of a model is equivalent to the performance of the optimal transport map. As a result, optimizing fairness is transformed to finding this optimal transport map, which does not essentially make the problem easier. To apply FTM, the practitioner has to pre-specify a mapping, which can be sub-optimal choice, e.g., an identity mapping, which leads to an undesirable accuracy-fairness trade-off. In a word, FTM seems not improving the efficiency to find a fair model. This is confirmed by the experiment results, where FTM typically is not the method with the best accuracy-fairness trade-off curve.

(ii) The writing should be improved to enhance readability, especially on the theoretical statements and proofs. I name a few below:
  1. Key definitions or quantities should be clearly defined to keep the paper self-contained, instead of referring to other works. Examples include transport map and push-forward measure in Section 2.3, and exact definition of Wasserstein distance, total variation, and K-S distance in Section 2.2.
  2. The proof of Prop 3.1 depends on Theorem 3.3, which has two assumptions. However, those assumptions are not mentioned in Prop 3.1 and its implications.
  3. Following point 2. The first condition (C1) that $F$ is bounded is automatically satisfied under this paper's setting, as $f$ predict a probability that belongs to $[0, 1]$. So (C1) is unnecessary. The second condition (C2), however, might be too strong to be practical. It requires that input $X$ has a absolutely continuous distribution, which is not satisfied for any categorical variables. For example, all datasets used in the experiments contains personal attributes such as gender, which is non-continuous.
  4. In Prop 3.1, the writing of $T_s = T_s(f)$ is confusing. I think the authors meant that for each $f$ there exist a correspond transport map. Using $T_s^f$ instead of $T_s(f)$, and define symbol instead of equal symbol, will eliminate this misunderstanding.
  5. The implication of Prop 3.1 also required more thinking. In particular, is this $T_s^f$ unique?
  6. In Def 3.4, does there exist a $T_s^f$ that achieves the minimum?
  7. The proof is too sloopy, with a bunch of undefined or not well-defined notations, such as $\Delta F_s(t)$, $\lfloor \delta \rfloor$, $F^{const}((v_{k-1}, v_k])$, etc. Without explanation, A CDF does not take an interval as the variable; $\lfloor \delta \rfloor$ is understood as the floor of $\delta$, which equals zero for $\delta<1$ and makes proof invalid.

(iii) The authors consider the setting with a single binary group and using DP as the fairness metric. Does the main observation and proposed method apply to more general scenarios? For example, multiple sensitive groups, multi-category group, and other fairness metrics such as equal opportunity, equalized odds, and calibration? The authors should discuss those limitations.

(iv) There are several papers mentioned by the authors that also apply optimal transportation to address model fairness issue. As those are highly related, the authors should clearly state the contributions of existing work and the difference from them with more details.

---

> ### Author Response · Authors · 2024-11-04
> **Author response #1**
>
> ### *Dear reviewer MW9E*: We appreciate your review comments and are truly happy to have the opportunity to improve our work. We hope our point-by-point responses provided below, along with the revised paper, will resolve your concerns.
>
> -------
> -------
> ### Weaknesses
> -------
>
> > **W1**:  The biggest concern is that, Thm 3.3 and 3.5 actually reflect that the fairness of a model is equivalent to the performance of the optimal transport map. As a result, optimizing fairness is transformed to finding this optimal transport map, which does not essentially make the problem easier. To apply FTM, the practitioner has to pre-specify a mapping, which can be a sub-optimal choice, e.g., an identity mapping, which leads to an undesirable accuracy-fairness trade-off. In a word, FTM seems not to improve the efficiency to find a fair model. This is confirmed by the experiment results, where FTM typically is not the method with the best accuracy-fairness trade-off curve.
>
> **Answer**:
> Implication of Theorem 3.3 and 3.5 (now renumbered as 3.7) is that **any transport map, not necessarily the optimal transport map,** corresponds to a group-fair model, and vice versa. Based on this mathematical finding, we propose a new algorithm called FTM **with which we can learn a group-fair model having specific properties (e.g., high subset fairness), by selecting a transport map accordingly.** We want to emphasize that **FTM is not designed to achieve the optimal trade-off between fairness and accuracy**. Rather, the focus is on **mitigating undesirable properties when achieving group fairness**, such as subset unfairness, by selecting a transport map accordingly.
>
> As you correctly point out, using FTM with a transport map that is not carefully selected could result in unreasonable group-fair models. For example, discrimination against certain subsets may occur (see Section C of Appendix for an example of a problematic group-fair model due to a poor transport map), even though group fairness is guaranteed by Theorem 3.7. Therefore, **the key to effectively using FTM lies in selecting a good transport map**.
>
> To this end, in Section 4, we recommend two types of optimal transport (OT) maps for FTM. The advantages of the two proposed maps are summarized as follows:
> 1. The marginal OT map can improve subset fairness (discussed theoretically in Theorem 4.2 and experimentally in Section 5.2.2).
> 2. The joint OT map can improve prediction accuracy and equalized odds compared to the marginal OT map (Table 3 and Figure 5 in Section 5.3.1).
>
> Furthermore, regarding your concern that FTM appears not empirically achieving the optimal fairness-prediction trade-off, we find that **using the joint OT map with large $\alpha$ can yield performance comparable to the best trade-offs** (see Figure 5 in Section 5.3.1 of the revised paper). That is, FTM can even control the fairness-prediction trade-off by controlling the transport map.
>
> In summary, the benefit of FTM is that it allows practitioners to control certain micro-level fairness (e.g., subset fairness) based on their needs, whereas existing algorithms focusing solely on group fairness may lack this flexibility.
>
> **Revision**:
> - **Section 1**: We add detailed implications of this paper in Introduction.
> - **Section 4**: We add the importance of selecting a good transport map at the beginning of Section 4.
> - **Section 5.3.1**: We add Figure 5 showing that FTM with the joint OT map can achieve competitive fairness-prediction trade-offs, compared to baseline methods.
>
> -------
> > **W2**:
> The writing should be improved to enhance readability, especially on the theoretical statements and proofs. I name a few below:
>
>
> >> **W2-1**:
> Key definitions or quantities should be clearly defined to keep the paper self-contained, instead of referring to other works. Examples include transport map and push-forward measure in Section 2.3, and exact definition of Wasserstein distance, total variation, and K-S distance in Section 2.2.
>
> **Answer**:
>
> **(1) Push-forward measure:**
> In fact, the push-forward measure is already defined in Section 2.3 as follows:
> *Here, the push-forward measure is defined by $\\mathbf{T}\_{\\#} \\mathcal{Q}\_{1}(A) = \\mathcal{Q}\_{1}(\\mathbf{T}\^{-1}(A))$ for any measurable set $A.$*
> Alternatively, we can say that the push-forward measure $\\mathbf{T}\_{\\#} \\mathcal{Q}\_{1}$ is the measure of $\\mathbf{T}(\\mathbf{X}), \\mathbf{X} \\sim \\mathcal{Q}\_{1},$ which we add in the revised paper.
>
> **(2) Wasserstein distance, Total Variation, and Kolmogorov-Smirnov distance:**
> As you comment, we add the definitions of these three distances in Section 2.2.
>
> **Revision**:
> - **Section 2.3**: We add a clearer definition of the push-forward measure.
> - **Section 2.2**: We add the exact definitions of the three distances.

---

> ### Author Response · Authors · 2024-11-04
> **Author response #2**
>
> -------
> >> **W2-2**:
> The proof of Prop 3.1 depends on Theorem 3.3, which has two assumptions. However, those assumptions are not mentioned in Prop 3.1 and its implications.
>
> **Answer**:
> Thank you for careful reading. Since Proposition 3.1 is a direct corollary of Theorem 3.3, it is necessary to state the condition (C2) before Proposition 3.1. We move (C2) to just before Proposition 3.1 (and renamed it as (C)). The purpose of assuming (C2) - now (C) - is to guarantee the existence of a transport map between two probability distributions, to make discussions easier. Furthermore, see our response to Weaknesses-2-(c) regarding the validity of Proposition 3.1 and Theorem 3.3 without the condition (C).
>
> **Revision**:
> - **Section 3.1**: We add this explanation just before Proposition 3.1.
>
> -------
> >> **W2-3**:
> Following point (b). The first condition (C1) that $f$ is bounded is automatically satisfied under this paper's setting, as $f$ predicts a probability that belongs to $[0, 1].$ So (C1) is unnecessary. The second condition (C2), however, might be too strong to be practical. It requires that input $\mathbf{X}$ has an absolutely continuous distribution, which is not satisfied for any categorical variables. For example, all datasets used in the experiments contain personal attributes such as gender, which is non-continuous.
>
> **Answer**:
> **(1) About (C1):**
> As you correctly point out, it is unnecessary to restate (C1), which we remove in the revised paper. Then, several terms are changed, e.g., $[-B, B]$ has been changed to $[0, 1]$ in the proof, the constant $C$ in Theorem 3.3 is now $2.$
>
> **(2) About (C2):**
> The condition (C2) -- now (C) -- is assumed to guarantee the existence of transport maps to make discussions easier. Importantly, we present a new theoretical result (Proposition A.2 in Section A of Appendix), similar to Theorem 3.3, which shows the existence of a stochastic transport map $\\mathbf{T}\_{s}$ satisfying $\\Delta MDP (f, \\mathbf{T}\_{s}) \le \delta,$ when the distributions are discrete.
>
> Furthermore, the existence of transport maps (and OT maps) in our experiments is also guaranteed, as the mini-batches from the two protected groups are sampled equally.
>
> **Revision**:
> - **Section 3.1**: We update the content including Proposition 3.1 and Theorem 3.3.
> - **Section A of Appendix**: We revise the proofs of Proposition 3.1 and Theorem 3.3. We add Proposition A.2, which discusses the case when the distributions are discrete.
>
> -------
> >> **W2-4**:
> In Prop 3.1, the writing of $T_{s} = T_{s}(f)$ is confusing. I think the authors meant that for each $f$ there exists a corresponding transport map. Using $T_{s}^{f}$ instead of $T_{s}(f)$, and defining the symbol instead of an equal symbol, will eliminate this misunderstanding.
>
> **Answer**:
> We modify the notation from "$\\mathbf{T}\_{s} = \\mathbf{T}\_{s} (f)$" to "$\\mathbf{T}\_{s} \in \\mathcal{T}\_{s}^{trans}$".
> Since the notation $\\mathbf{T}\_{s}\^{f}$ you suggested is already used in Definition 3.4, and we do not use the notation $\\mathbf{T}\_{s}(f)$ after Proposition 3.1, we believe this modification is sufficient to avoid further confusion.
>
> **Revision**:
> - **Proposition 3.1**: We revise the statement of Proposition 3.1 accordingly.
>
> -------
> >> **W2-5**:
> The implication of Prop 3.1 also required more thinking. In particular, is this $T_{s}^{f}$ unique?
>
> **Answer**:
> Note that Proposition 3.1 implies that there exists a mapping corresponding to a given perfectly group-fair model, but it does not imply that the mapping is unique. In fact, the $\\mathbf{T}\_{s}$ in Proposition 3.1 is not unique. Recall that $\\mathbf{T}\_{s}$ transports an input $\\mathbf{x}\_{s} \in \mathcal{X}_{s}$ to $\\mathbf{x}\_{s'} \in \\mathcal{X}\_{s'}$ to satisfy $f(\\mathbf{x}\_{s}, s) = f(\\mathbf{x}\_{s'}, s').$ Hence, if $f$ is not injective, i.e., multiple $\\mathbf{x}\_{s}$ can map to the same $f(\\mathbf{x}\_{s}, s),$ $\\mathbf{T}\_{s}$ is not unique.

---

> ### Author Response · Authors · 2024-11-04
> **Author response #3**
>
> -------
> >> **W2-6**:
> In Def 3.4, does there exist a $T_{s}^{f}$ that achieves the minimum?
>
> **Answer**:
> Yes, it can be shown by Briener's theorem. In specific, since $\\mathcal{P}\_{s}, s \in \{0, 1\}$ are absolutely continuous by (C), and the $L_{1}$ cost function used in MDP (i.e., $c(\mathbf{x}, \mathbf{y}) := \vert f(\mathbf{x}, s) - f(\mathbf{y}, s') \vert$) is lower semi-continuous and bounded from below, the minimizer of $\\Delta MDP (f, \\mathbf{T}_{s})$ uniquely exists.
>
> **Revision**:
> - **Section 3.1**: We add a discussion regarding the existence of the fair matching function just after Definition 3.4:
>   *Note that the existence of this fair matching function is guaranteed by Briener's theorem (Villani, 2008; Hütter & Rigollet, 2021). To be specific, since $\\mathcal{P}\_{s}, s \in \\{0, 1\\}$ are absolutely continuous by (C), and the cost function in MDP, i.e., $c(\mathbf{x}, \mathbf{y}) := \vert f(\mathbf{x}, s) - f(\mathbf{y}, s') \vert,$ is lower semi-continuous and bounded from below, the minimizer of $\Delta \textup{MDP} (f, \mathbf{T}_{s})$ uniquely exists.*
>
> -------
> >> **W2-7**:
> The proof is too sloppy, with a bunch of undefined or not well-defined notations, such as $\Delta F_{s}(t), \lfloor \rfloor, F^{cont}((v_{k-1}, v_{k}])$, etc. Without explanation, a CDF does not take an interval as the variable; $\lfloor \delta \rfloor$ is understood as the floor of $\delta,$ which equals zero for $\delta < 1$ and makes proof invalid.
>
> **Answer**:
> We apologize for any poor notation. We thoroughly revise the proofs, ensuring that all notations are clearly defined and the flow is easier to follow. The proofs in Section A of Appendix in the revised paper are corrected accordingly.
>
> **Revision**:
> - **Section A of Appendix**: We carefully revise the proofs of Proposition 3.1 and Theorem 3.3.
>
>
>
>
> -------
> > **W3**:
> The authors consider the setting with a single binary group and using DP as the fairness metric. Does the main observation and proposed method apply to more general scenarios? For example, multiple sensitive groups, multi-category group, and other fairness metrics such as equal opportunity, equalized odds, and calibration? The authors should discuss those limitations.
>
> **Answer**:
> We appreciate your suggestions, which will help enrich the conclusion part of this study.
>
> **(1) Other fairness measures:**
> We did not discuss the detailed FTM algorithm for Eqopp in the current version, as it is very similar to that for DP. However, we add the discussion regarding the extension of FTM to Eqopp in Section 6 of the revised paper. Moreover, note that we have already shown that using the joint OT map can contribute to improving equalized odds (see Section 5.3.1).
>
> **(2) Multiple sensitive attributes or Multi-category group:**
> For multi-category problems, the current FTM could be extended easily by modifying MDP for multi-dimensional functions accordingly. However, the extension of FTM to multiple sensitive attributes would not be easy partly because defining a good matching among multiple individuals from multiple protected groups is challenging, so we would leave this topic for future research. We add this discussion in Section 6 of the revised paper.
>
> **Revision**:
> - **Section 6**: We add the following paragraphs in the revised paper:
>   *This paper has focused on DP (demographic parity) for simplicity and clarity, but applying FTM to Eqopp (equal opportunity) is straightforward. Specifically, we only consider instances of $Y=1$ when calculating MDP. Expanding FTM to other fairness notions would be a valuable avenue for future work.*
>
>   *The scenario involving multiple sensitive attributes, which is widely explored in group fairness studies, is another potential direction. In such cases, matching individuals from more than two protected groups is required for FTM. However, matching multiple individuals from multiple protected groups is challenging, and so we leave this as a future work.*

---

> ### Author Response · Authors · 2024-11-04
> **Author response #4**
>
> -------
> > **W4**:
> There are several papers mentioned by the authors that also apply optimal transportation to address model fairness issues. As those are highly related, the authors should clearly state the contributions of existing work and the difference from them with more details.
>
> **Answer**:
> Following your suggestions, we add a detailed discussion about the related existing methods in Section 2.4.
>
> Specifically, for Gordaliza et al. (2019), we provide a conceptual comparison with FTM in Section 3.3 and an empirical comparison in Section 5.4.3. Note that FTM empirically outperforms several existing FRL (Fair Representation Learning) methods, including Gordaliza et al. (2019).
>
> **Revision**:
> - **Section 2.4**: We add the following discussion -
>   *Most of these existing algorithms (e.g., Jiang et al. (2020b); Chzhen et al. (2020); Silvia et al. (2020)) focus on applying the OT map in the prediction space, i.e., aligning two conditional distributions (distributions of $\hat{Y} | S = s, s \in \\{0, 1\\}$). These methods fundamentally differ from our approach, which focuses on applying the OT map in the input space. On the other hand, Gordaliza et al. (2019) and our approach are similar in the sense that both apply the OT map on the input space. See the detailed comparison between Gordaliza et al. (2019) and our approach in Section 3.3.*
>
> - **Section 3.3**: We add the following discussion -
>   *In particular, the FRL method in Gordaliza et al. (2019) and FTM are similar in the sense that both apply the OT map on the input space. However, Gordaliza et al. (2019) differs from ours as it learns a prediction model on the (pre-trained) fair representation space obtained by the OT map, while the prediction model of FTM is defined on the original input space and the OT map is used only in the fairness constraint.*
>
>   *Hence, the prediction model of FTM is not tied with the OT map and thus is more flexible, resulting in better prediction accuracy. See Table 6 in Section 5.4.3 for empirical evidence, showing the outperformance of FTM over several FRL methods including Gordaliza et al. (2019).*
>
> -------
> -------
> ### Requested changes
>
> -------
> > **RC1**:
> Point Weaknesses-1, what is the particular advantage of FTM compared to other methods? As there is no particular guide to select the optimal transport map, and ad hoc choices such as marginal or joint distribution do not offer better accuracy-fairness trade-off.
>
> **Answer**:
> Please see our response to **W1**.
>
> -------
> > **RC2**:
> The assumptions and proofs of all theoretical results should be reviewed thoroughly.
>
> **Answer**:
> Please see our response to **W2**.

---

### Review · Reviewer_H7Dq · 2024-10-16

**Summary Of Contributions:**

The authors introduce a regularization approach for group fairness in the binary classification setting based on a new group fairness measure called Matched Demographic Parity, which quantifies the averaged difference between predictions of two individuals coming from different protected groups who are matched by a given transport map. They provide an algorithm, called Fairness Through Matching (FTM), that learns group fair classifiers and is dependent on the choice of the transport map. Two different choices of the transport map are proposed. The first one aims at minimizing the transport cost while obtaining both group and subset fairness. The second one also considers the similarity in the target variable, in order to improve the prediction accuracy. The authors justify their approach theoretically and present empirical results from experiments on several benchmark datasets.

**Audience:**

Yes

**Claims And Evidence:**

No

**Requested Changes:**

Please, see the detailed weaknesses above.

**Strengths And Weaknesses:**

Strengths:

1. This work introduces a novel transport-based fairness measure, called Matched Demographic Parity, and proves theoretical support that establishes connections with existing group fairness measures, such as strong demographic parity.
2. The manuscript is well-structured and well-written.

Weaknesses:

The main limitations found in the manuscript are the following:
1. The motivation for the work is not sufficiently clear. In the Introduction, authors say that the existing group fairness measures, such as demographic parity, only take into account statistical disparities, with no concern for how models achieve group fairness, leading to high fair models obtained in unreasonable ways. However, this is a statement that requires more justification than the one referred to in Section C of the Appendix. In fact, the example they provide has to do with possible undesirable matchings, instead of with other concrete group fairness measures.
2. Transportation plans have been widely used to improve fairness while ensuring the accuracy of the algorithms in different settings. Authors mention in Section 2 a very brief review of some related work, but do not make it clear how they improve on it. In this sense, the experimental part of the paper lacks a more detailed comparative study with other transportation-based approaches.
3. Also related to review issues, I am missing counterfactual fairness approaches since in fact I appreciate certain similarities with the proposed definition of Matched Demographic Parity (MDP).
4. The proposed fairness measure of a model depends on a given transport map. Thus, I have concerns about the robustness and stability of such a measure in a practical setting, as well as the scalability when very different models have to be trained. On the other hand, with regard to implementation in real problems where there is usually a large disproportion between the protected groups in the data, I wonder about the limitations of this approach.
5. A confusing nomenclature for fairness notions is used throughout the paper. First of all, it is not clear what fair means for the authors. In Section 3 they mention “perfectly fair group-model” and “fair group-model” without a proper explanation. It can be deduced from the context that they refer to the strong demographic parity definition, that requires that the predictions in each protected group are equally distributed, which is equivalent to some distance between the probability distributions of $f(X)|S=0$ and $f(X)|S=1$ (e.g. Wasserstein distances, TV distance or KS distance) being zero. However, they do not state this at any point in the paper. On the other hand, another issue that leads to confusion in this regard, is that later in Section 4 they consider two other very different fairness measures: subset fairness (section 4.1.) and equalized odds (section 4.2.). This makes the objective of the work very ambiguous.
6. Results in section 3, as well as their proofs in the appendix, are interesting but it is not clear what is the benefit of using MDP instead of other strong demographic measures, such as the Total Variation or Wasserstein distances.
7. Regarding section 4:
    - Subset fairness seems to be closely related to individual fairness measures, therefore a brief comparison with the existing literature of this type would be useful.
    - Theorem 4.2. establishing an upper bound for the level of subset fairness in terms of different transport costs is promising. However, the first term of RHS is the transport cost of the minimizer of the MDP, which is a different problem. Hence, I would suggest authors to elaborate more, if possible, on the computation of the "matching function of $f$", and also to add a discussion of the transport cost of such a map. Furthermore, I would suggest to use a more concrete name for Definition 3.4., since "matching function" is quite general and is used in many different contexts.
    - In practice, the proposed algorithm for FTM is actually based on Fair Representation Learning, since they compute the OT map between the probability distributions of $X|S=0$ and $X|S=1$. Therefore, I do not see the relationship of the definition of group fairness and the subsequent result (Theorem 4.2) to the regularization approach of FTM, since the penalization is done in terms of the OT map instead of the MDP of the "matching function of $f$" as stated in the theorem. Moreover, the convenience for this choice of the OT map carrying the distribution towards their weighted Wasserstein barycenter, which is the minimizer of the Wasserstein variation, has been previously proven in Gordaliza et al. (2019). In this sense, I would ask the authors to include some references about FRL, in particular, to the results on the application of Wasserstein barycenters to fair classification.
    - The procedure for the “joint OT map” in section 4.2. is not sufficiently justified theoretically and the way to measure EO in section 5.3.1. is rather debatable. EO is a two-dimensional measure and calculating the mean of the differences $|\mathbb{E}(f_0(\mathbf{X}))|S=0, Y=y) - \mathbb{E}(f_1(\mathbf{X}))|S=1, Y=y)|  $, for $y=0,1$, may not be a very robust and informative indicator. Without sufficient support for the use of this definition, the results of experiments may not be entirely reliable.

Some additional minor comments:

8. Final paragraph in sections 4.1. and 4.2. is exactly the same, so it could be expressed in a more concise way to ease the reading.

9. Authors should clarify the order of the Wasserstein distance, since both 1 and 2 have been widely used in fair learning.

10. In Section 5: work -> works
11. Notation: I do not find it appropriate to use the symbol Delta for the fairness measure.
12. Proposition 3.1. is in fact a Corollary of Theorem 3.3. As the authors say in the proof, it is a particular case of the Theorem, for which Assumptions (C1) and (C2) are also needed.
13. Title of 3.2. would be more informative with the full name of the algorithm.

---

> ### Author Response · Authors · 2024-11-04
> **Author response #1**
>
> ### *Dear reviewer H7Dq*: We appreciate your review comments and are truly happy to have the opportunity to improve our work. We hope our point-by-point responses provided below, along with the revised paper, will resolve your concerns.
>
> -------
> -------
> ### Weaknesses
> -------
> > **W1**: The motivation for the work is not sufficiently clear. In the Introduction, authors say that the existing group fairness measures, such as demographic parity, only take into account statistical disparities, with no concern for how models achieve group fairness, leading to high fair models obtained in unreasonable ways. However, this is a statement that requires more justification than the one referred to in Section C of the Appendix. In fact, the example they provide has to do with possible undesirable matchings, instead of with other concrete group fairness measures.
>
> **Answer**:
> The main message of our paper is that **undesirable group-fair models correspond to undesirable transport maps, and thus we can avoid such undesirable models by using a desirable transport map in MDP**.
>
> Undesirable group-fair models could be learned if we focus solely on the statistical disparity between different protected groups. Several examples discussed in this paper, such as subset targeting and self-fulfilling prophecy from Dwork et al. (2012), highlight this issue. Our aim is to **explore group fairness from a micro-level and intrinsic perspective, then subsequently resolve its limitations**.
>
> To this end, we investigate the inherent properties of group-fair models by theoretically revealing the correspondence between group-fair model and transport map. This correspondence allows us to **understand the underlying mechanism of achieving group fairness in models learned by minimizing the statistical disparity such as TVDP and WDP, by identifying which pairs of individuals (from different protected groups) are treated similarly**. Moreover, we develop a method of learning group-fair models with a pre-specified transport map, allowing us to avoid undesirable properties.
>
> Implications obtained from this matching mechanism include:
> - Matching dissimilar individuals to achieve group fairness may lead to subset unfairness (see Section C).
> - Matching similar individuals helps mitigate the risk of a self-fulfilling prophecy (see Section E.3).
> - As demonstrated in Figure 4 and Table 2, such undesirable properties can emerge in group-fair models trained with existing algorithms on real-world datasets.

---

> ### Author Response · Authors · 2024-11-04
> **Author response #2**
>
> -------
> > **W2**:
> Transportation plans have been widely used to improve fairness while ensuring the accuracy of the algorithms in different settings. Authors mention in Section 2 a very brief review of some related work, but do not make it clear how they improve on it. In this sense, the experimental part of the paper lacks a more detailed comparative study with other transportation-based approaches.
>
> **Answer**:
> Following your suggestions, we add more details about existing works using the OT map for algorithmic fairness, in Section 2.4. In Section 3.3, we also add a conceptual comparison of FTM and Gordaliza et al. (2019), which are particularly similar in the sense that both apply the OT map on the input space. Furthermore, we present an **experimental result showing the outperformance of FTM over existing FRL methods including Gordaliza et al. (2019)** in Section 5.4.3.
>
> **Revision**:
> - **Section 2.4**: We add a detailed description of existing works applying the OT map for algorithmic fairness.
>   *Most of these existing algorithms (e.g., Jiang et al. (2020b); Chzhen et al. (2020); Silvia et al. (2020)) focus on applying the OT map in the prediction space, i.e., aligning two conditional distributions (distributions of $\hat{Y} | S = s, s \in \{0, 1\}$). These methods fundamentally differ from our approach, which focuses on applying the OT map in the input space. On the other hand, Gordaliza et al. (2019) and our approach are similar as both apply the OT map on the input space. See the detailed comparison between Gordaliza et al. (2019) and our approach in Section 3.3.*
>
> - **Section 3.3**: We add a conceptual comparison of FTM and FRL (Fair Representation Learning) approaches, including Gordaliza et al. (2019).
>   *In particular, Gordaliza et al. (2019) and FTM are similar in the sense that both apply the OT map on the input space. However, Gordaliza et al. (2019) differs from ours as it learns a prediction model on the (pre-trained) fair representation space obtained by the OT map, while the prediction model of FTM is defined on the original input space and the OT map is used only in the fairness constraint.*
>  *Hence, the prediction model of FTM is not tied with the OT map and thus is more flexible to results in better prediction accuracy. See Table 6 for empirical evidence, showing the outperformance of FTM over several FRL methods including Gordaliza et al. (2019).*
>
> - **Section 5.4.3**: We add an experimental result (Table 6) showing the outperformance of FTM over several existing FRL methods, including Gordaliza et al. (2019).
>
> -------
> > **W3**: Also related to review issues, I am missing counterfactual fairness approaches since in fact I appreciate certain similarities with the proposed definition of Matched Demographic Parity (MDP).
>
> **Answer**:
> As you note, FTM is related to counterfactual fairness. That is, there exists a simple SCM (Structural Causal Model) where **the counterfactual of a given input is equivalent to the transported input mapped by the (marginal) OT map**. It implies that FTM with the marginal OT map would achieve counterfactual fairness.
>
> Accordingly, we introduce counterfactual fairness in the related works section (Section 2.4). Furthermore, we provide Proposition 4.3, which shows that there exists an SCM where applying FTM with the marginal OT map would achieve counterfactual fairness (see the paragraph "Connection to Counterfactual Fairness" in Section 4.1).
>
> **Revision**:
> - **Section 2.4**: We add a paragraph introducing the notion of counterfactual fairness and related works.
> - **Section 4.1**: We add a paragraph named `Connection to Counterfactual Fairness' including Proposition 4.3.

---

> ### Author Response · Authors · 2024-11-04
> **Author response #3**
>
> -------
> > **W4**: The proposed fairness measure of a model depends on a given transport map. Thus, I have concerns about the robustness and stability of such a measure in a practical setting, as well as the scalability when very different models have to be trained. On the other hand, with regard to implementation in real problems where there is usually a large disproportion between the protected groups in the data, I wonder about the limitations of this approach.
>
> **Answer**:
> - **(Robustness and stability)**  MDP is stable with respect to the variation of a transport map. Specifically, we can show that **the difference between two $\Delta \\textup{MDP}$s for two given transport maps is bounded by the difference between the two transport maps, provided that the prediction model is Lipschitz**, as follows:
> Let $\\mathbf{T}\_{s}^{A}$ and $\\mathbf{T}\_{s}^{B}$ be given two transport maps, and assume that $f$ is $L$-Lipschitz for simplicity.
> Then, we have that
>
>   \\begin{split}
>       & \\vert \\Delta \\textup{MDP} (f, \\mathbf{T}\_{s}^{A}) - \\Delta \\textup{MDP} (f, \\mathbf{T}\_{s}^{B}) \\vert \\\\
>       & = \\left\\vert \\mathbb{E}\_{s} \\vert f(\\mathbf{X}, s) - f(\\mathbf{T}\_{s}^{A}(\\mathbf{X}), s') \\vert - \\mathbb{E}\_{s} \\vert f(\\mathbf{X}, s) - f(\\mathbf{T}\_{s}^{B}(\\mathbf{X}), s') \\vert \\right\\vert \\\\
>       & \\le \\mathbb{E}\_{s} \\vert f(\\mathbf{T}\_{s}^{A}(\\mathbf{X}), s') - f(\\mathbf{T}\_{s}^{B}(\\mathbf{X}), s') \\vert \\\\
>       & \\le L \\left( \\mathbb{E}\_{s} \\Vert \\mathbf{T}\_{s}^{A}(\\mathbf{X}) - \\mathbf{T}\_{s}^{B}(\\mathbf{X}) \\Vert^{2} \\right)^{1/2}.
>   \\end{split}
>
>
> - **(Scalability)**
> Computational complexity of FTM is not affected by the discrepancy of $f(\\cdot, 0)$ and $f(\\cdot, 1)$ as well as the discrepancy of $\\mathcal{P}\_{0}$ and $\\mathcal{P}\_{1},$ when the transport map is fixed.
> In turn, complexity in computing the marginal or joint OT map is proportional to the sample size but not to the discrepancy of $\\mathcal{P}\_{0}$ and $\\mathcal{P}\_{1}.$
> Hence, FTM can be generally applied to any datasets without much hamper.
>
> - **(Stability under the imbalance setting)**
> For this revision, we compare the stability of our method and existing algorithms under an imbalance setting for the sensitive attribute.
> For this purpose, after training/test data split, we construct an imbalanced training dataset with a 5:95 ratio (5% for $S=0$ and 95% for $S=1$) by randomly sampling data from $\\mathcal{D}\_{0}$ and fully using $\\mathcal{D}\_{1}.$
> Then, we evaluate the performances on test dataset, where the models are learned on this imbalanced training dataset.
> We conduct this random process several times.
>
>    See Table 4 in Section 5.4.1 in the revised paper for the results, which shows that FTM is stable for the imbalance setting.
>
> **Revision**:
>
> - **Section 5.4.1**: We add Table 4, showing the stability of FTM under an imbalance setting.
>
> -------
> > **W5**: A confusing nomenclature for fairness notions is used throughout the paper. First of all, it is not clear what fair means for the authors. In Section 3 they mention “perfectly fair group-model” and “fair group-model” without a proper explanation. It can be deduced from the context that they refer to the strong demographic parity definition, that requires that the predictions in each protected group are equally distributed, which is equivalent to some distance between the probability distributions of $f(X)|S=0$ and $f(X)|S=1$ (e.g. Wasserstein distances, TV distance or KS distance) being zero. However, they do not state this at any point in the paper. On the other hand, another issue that leads to confusion in this regard, is that later in Section 4 they consider two other very different fairness measures: subset fairness (section 4.1.) and equalized odds (section 4.2.). This makes the objective of the work very ambiguous.
>
> **Answer**:
> We add a definition of the perfectly group-fair model in Section 2.2:
> *Let $\\Delta = \\Delta (f)$ be a given measure of fairness. A given model $f$ is said to be group-fair (with level $\\epsilon$) if it satisfies $\\Delta (f) \le \\epsilon.$ Furthermore, if $\Delta (f) = 0,$ we say $f$ is perfectly group-fair (with respect to $\\Delta$).*
>
> We want to emphasize that **our focus is on mitigating undesirable properties when achieving group fairness**, such as subset unfairness, by selecting a transport map accordingly. In Section 4, we discuss the advantages of FTM with the two proposed transport maps. These transport maps contribute to improving other fairness notions, such as subset fairness and equalized odds, which are specific benefits (while achieving group fairness).
>
> **Revision**:
> - **Section 2.2**: We modify the title from "Measures for group fairness" to "Measures for group fairness & Definition of a group-fair model", and add the definition of group-fair model.

---

> ### Author Response · Authors · 2024-11-04
> **Author response #4**
>
> -------
> > **W6**:
> Results in section 3, as well as their proofs in the appendix, are interesting but it is not clear what is the benefit of using MDP instead of other strong demographic measures, such as the Total Variation or Wasserstein distances.
>
> **Answer**:
> Let $\Delta$ be a given (existing) fairness measure, and define $\\mathcal{F}\_{\\Delta} (\\delta) := \\{ f \\in \\mathcal{F} : \\Delta(f) \\le \\delta \\}$ as the set of group-fair models of level $\\delta$ (with respect to $\\Delta$).
> Similarly, for MDP, define $\\mathcal{F}\_{\\Delta \\textup{MDP}} (\\mathbf{T}\_{s}, \\delta) := \\{ f \\in \\mathcal{F} : \\Delta \\textup{MDP} (f, \\mathbf{T}\_{s}) \\le \delta \\}.$ Following from Theorem 3.3 and 3.5 (now renumbered as 3.7), we know that the three measures (TVDP, WDP, and MDP) are closely related:
>         $        \\mathcal{F}\_{\\Delta \\textup{TVDP}} (\\delta) \\subseteq \\cup\_{\\mathbf{T}\_{s} \\in \\mathcal{T}\_{s}^{\\textup{trans}}} \\{ f: \Delta \\textup{MDP}(f, \\mathbf{T}\_{s}) \\le 2\\delta \\} \\subseteq \\mathcal{F}\_{\\Delta \\textup{WDP}} (2\\delta).        $
>
> As discussed in Section 1 (Introduction) as well as previous works (e.g., Dwork et al., (2012)), there exist group-fair models having undesirable properties such as subset targeting or self-fulfilling prophecy. The advantage of MDP is that we can **screen out undesirable group-fair models during the learning phase, to consider desirable group fair models only**. And, using MDP achieves this goal by considering group-fair models whose transport maps have low transport costs. That is, we can search for a group-fair model only on $\\cup\_{\\mathbf{T}\_{s} \\in \\mathcal{T}\_{s}\^{\\textup{good trans}}} \\{ f: \\Delta \\textup{MDP}(f, \\mathbf{T}\_{s}) \\le 2\\delta \\},$ where  $\\mathcal{T}\_{s}\^{\\textup{good trans}} \\subseteq \\mathcal{T}\_{s}\^{\\textup{trans}}$ is the set of good transport maps. For good transport maps, we specifically consider the marginal and joint OT maps in Section 4, but other maps could be used.
> Note that this kind of screening is not possible when using existing group-fair measures.
>
> **Revision**:
> - **Section 3.2**: We add this explanation just after Theorem 3.7.
>
> -------
> > **W7**: Regarding section 4:
>
> >> **W7-1**: Subset fairness seems to be closely related to individual fairness measures, so a brief comparison with existing literature of this type would be useful.
>
> **Answer**
> FTM and individual fairness are similar in the sense that they try to treat similar individuals similarly.
> A difference is that FTM aims to treat **two individuals from different protected groups** similarly, while the individual fairness tries to treat similar individuals similarly **regardless of sensitive attribute (even when it is unknown)**.            That is, similar individuals in FTM could be dissimilar in view of individual fairness, especially when the two protected groups are significantly different.
>
> A limitation of individual fairness is that group fairness is not guaranteed. FTM can be understood as a tool to resolve this limitation by searching higher individually fair ones among group-fair models. Empirical results support this conjecture that FTM improves individual fairness compared to baseline methods for group fairness (see Table 5 in Section 5.4.2).
>
> **Revision**
> - **Section 2.4**: We add a paragraph introducing the notion of individual fairness and related works.
> - **Section 3.3**: We add paragraphs of this discussion comparing the purposes of FTM and individual fairness.
>  - **Section 5.4.2**: We build a new subsection, including Table 5 comparing FTM and baseline algorithms in view of individual fairness.

---

> ### Author Response · Authors · 2024-11-04
> **Author response #5**
>
> -------
>
> >> **W7-2**: Theorem 4.2. establishing an upper bound for the level of subset fairness in terms of different transport costs is promising. However, the first term of RHS is the transport cost of the minimizer of the MDP, which is a different problem. ...
>
> **Answer**:
>
> - **(Modified term)** We revise the term "matching function" to "fair matching function" in Definition 3.4, as it is the map that minimizes the MDP.
>
> - **(How to obtain the fair matching function)**
> Let $\\mathcal{D}\_{0} = \\{ \\mathbf{x}\_{i} : s\_{i} = 0 \\} = \\{ \\mathbf{x}\_{i}\^{(0)} \\}\_{i=1}\^{n_{0}}$ and $\\mathcal{D}\_{1} = \\{ \mathbf{x}_{j} : s\_{j} = 1 \\} = \\{ \\mathbf{x}\_{j}\^{(1)} \\}\_{i=1}\^{n\_{1}}$ be the set of inputs divided by the sensitive attribute, where $n\_{0} + n\_{1} = n$.
>
> 1. **Case of $n_{0} = n_{1}$**:
> When the sizes of two protected groups are equal ($n\_{0} = n\_{1}$), we can easily find the fair matching function by quantile matching. We first sort the scores in $\\{ f\_{s}(\\mathbf{x}) \\}_{\\mathbf{x} \\in \\mathcal{D}\_{s}}$ for each group $s \\in \\{0,1\\}.$ Then, we match the individuals having the same rank (i.e., quantile) in each set, thereby obtaining the fair matching function of $f$. Its transport cost is then subsequently computed by the mean distance between two matched individuals. This straightforward procedure is theoretically guaranteed by the definition of 1-Wasserstein distance, which is calculated by quantile matching (Jiang et al., 2020; Chzhen et al., 2020; Rachev and Ruschendorf, 1998). We theoretically prove this procedure in Proposition 3.5.
>
> 2. **Case of $n_{0} \\neq n_{1}$**:
> When the sizes differ, we can consider a stochastic fair matching function (a stochastic transport map that minimizes $\\Delta \\textup{MDP}$), which matches individuals with probability, not deterministically (see Section A for more details about the stochastic transport map).
> In fact, a stochastic transport map is equivalent to a joint distribution between two protected groups, as follows:
>
>    Denote $\\mathbb{Q}$ as a joint distribution between $\\mathcal{D}\_{0}$ and $\\mathcal{D}\_{1}.$ Let $\\mathbf{X}\_{0}$ and $\\mathbf{X}\_{1}$ be the random variables following the empirical distributions on $\\mathcal{D}\_{0}$ and $\\mathcal{D}\_{1},$ respectively. The stochastic transport map $\\mathbf{T}\_{s}$ corresponding to a given $\\mathbb{Q}$ is defined by $\\mathbf{T}\_{s}(\\mathbf{x}\_{i}\^{(s)}) = \\mathbf{x}\_{j}^{(s')}$ with probability $\\mathbb{Q}(\\mathbf{X}\_{s} = \\mathbf{x}\_{i}\^{(s)}, \\mathbf{X}\_{s'} = \\mathbf{x}\_{j}\^{(s')}).$ Once we find the stochastic fair matching function, the stochastic transport map (joint distribution $\\mathbb{Q}$) minimizing $\\Delta \\textup{MDP}(f, \\mathbb{Q}) = \\mathbb{E}\_{(\\mathbf{X}\_{0}, \\mathbf{X}\_{1}) \\sim \\mathbb{Q}} \\vert f(\\mathbf{X}\_{0}, 0) - f(\\mathbf{X}\_{1}, 1) \\vert,$ we can compute its transport cost as $\\mathbb{E}\_{(\\mathbf{X}\_{0}, \\mathbf{X}\_{1}) \\sim \\mathbb{Q}} \\Vert \\mathbf{X}\_{0} - \\mathbf{X}\_{1} \\Vert^{2}.$ Note that the minimization of $\\Delta \\textup{MDP}$ with respect to the stochastic transport map is technically equivalent to solving the Kantorovich problem, which can be easily solved by the use of linear programming. We introduce details about the Kantorovich problem in Section D of Appendix.
>
>    Or alternatively, we can apply the mini-batch sampling technique similar to the computation of OT map for FTM. We sample two random mini-batches $\\tilde{\\mathcal{D}}\_{0} \\subset \\mathcal{D}\_{0}$ and $\\tilde{\\mathcal{D}}\_{1} \\subset \\mathcal{D}\_{1}$ with identical size $m.$ Then, we follow the process in `(1) Case of $n\_{0} = n\_{1}$'. The transport cost of fair matching function can be estimated by the average of transport costs computed on many random mini-batches.
>
>    We formally present this discussion and corresponding procedures with details in Section 3.1.
>
> - **(Usage of the transport cost of the fair matching function)**
> Furthermore, the transport cost of the fair matching function can serve as a measure to assess whether a given group-fair model is desirable. For example, **when choosing a model between two group-fair models with similar levels of group fairness or/and prediction accuracies, the model with the lower transport cost would be preferred.** In fact, in Section 5.3.2, we compare the transport costs of the fair matching functions for two different group-fair models.
>
> **Revision**:
>
> - **Section 3.1**:
> We modify the term "matching function" to "fair matching function" in Definition 3.4.
> We add a paragraph "Practical computation of the fair matching function", including Proposition 3.5, which shows the practical computation of fair matching function.
> We add Remark 3.6 to discuss the usage of the transport cost of the fair matching function.
>
> - **Section D of Appendix**:
> We build a new section introducing the Kantorovich problem, with details.

---

> ### Author Response · Authors · 2024-11-04
> **Author response #6**
>
> -------
> >> **W7-3**: In practice, the proposed algorithm for FTM is actually based on Fair Representation Learning, since they compute the OT map between the probability distributions of $X|S=0$ and $X|S=1$. Therefore, I do not see the relationship of the definition of group fairness and the subsequent result (Theorem 4.2) to the regularization approach of FTM, since the penalization is done in terms of the OT map instead of the MDP of the "matching function of $f$" as stated in the theorem. Moreover, the convenience for this choice of the OT map carrying the distribution towards their weighted Wasserstein barycenter, which is the minimizer of the Wasserstein variation, has been previously proven in Gordaliza et al. (2019). In this sense, I would ask the authors to include some references about FRL, in particular, to the results on the application of Wasserstein barycenters to fair classification.
>
>
> **Answer**:
>
> First, the role of transport map in FTM and the role of aligning two conditional representation distributions in FRL are different.
> FTM builds a prediction model in the original input space, using transport map in the MDP constraint only.
> On the contrary, FRL methods first learn fair representation by aligning the conditional distributions of representation, and then use the learned fair representation as a input.
>
> Furthermore, the motivations of FTM and FRL also fundamentally differ.
> FTM is designed to learn group-fair models with desirable properties (e.g., higher fairness on subsets).
> In contrast, FRL methods aims to obtain fair representations which can be used for downstream tasks requiring fairness.
>
> We add this conceptual comparison in Section 3.3.
>
> Also, see our response to Weaknesses-2 for the specific comparison of FTM and the FRL method from Gordaliza et al. (2019).
> We also empirically show that FTM is superior to several FRL methods including Gordaliza et al. (2019), in Table 6 of the revised paper.
>
> **Revision**:
>
> - **Section 3.3**: We add this explanation in the paragraph named "Fair representation learning".
>
> - **Section 5.4.3**: We add an experimental result (Table 6) showing the outperformance of FTM over several FRL methods, including Gordaliza et al. (2019).

---

> ### Author Response · Authors · 2024-11-04
> **Author response #7**
>
> -------
> >> **W7-4**: The procedure for the “joint OT map” in section 4.2. is not sufficiently justified theoretically and the way to measure EO in section 5.3.1. is rather debatable. EO is a two-dimensional measure and calculating the mean of the differences $|\mathbb{E}(f_{0}(X)|S=0, Y=y) - \mathbb{E}(f_{1}(X)|S=1, Y=y)|$, for $y=0,1,$ may not be a very robust and informative indicator. Without sufficient support for the use of this definition, the results of experiments may not be entirely reliable.
>
> **Answer**:
>
> - **(About the measure)**
> First, the measure used in this paper, i.e., the mean (or sum) of the differences (difference in TPR + difference in FPR), has been also considered in previous works (e.g., Donini et al., 2018; Chuang & Mroueh, 2021).
> Note that the differences in TPR and FPR are defined by
> $\\Delta \\textup{TPR}(f) := \\vert \mathbb{P}( C\_{f\_{0}}(\\mathbf{X}) = 1 \\vert Y = 1, S = 0 ) - \\mathbb{P}( C\_{f\_{1}}(\\mathbf{X}) = 1 \\vert Y = 1, S = 1 ) \vert$ and
> $\\Delta \\textup{FPR}(f) := \\vert \\mathbb{P}( C\_{f\_{0}}(\\mathbf{X}) = 1 \\vert Y = 0, S = 0 ) - \\mathbb{P}( C\_{f\_{1}}(\\mathbf{X}) = 1 \vert Y = 0, S = 1 ) \\vert,$ respectively.
>
>    However, as you note, it would be more convincing to report both the differences in $\\Delta \\textup{TPR}$ and $\\Delta \\textup{FPR}.$ Hence, we modify Table 3 to additionally report $\\Delta \\textup{TPR}$ and $\\Delta \\textup{FPR}.$  It still shows that the joint OT map can improve EO, when compared to the marginal OT map.
>
> - **(The relationship between the joint OT map and EO)**
> We can decompose $\\Delta \\textup{MDP}(f, \\mathbf{T}\_{s})$ as
> $ \\Delta \\textup{MDP}(f, \\mathbf{T}\_{s})  = w\_{0} \\mathbb{E}\_{s} ( \\vert f(\\mathbf{X}, s) - f(\\mathbf{T}\_{s}(\\mathbf{X}), s') \\vert \\vert Y = 0 )   + w\_{1} \\mathbb{E}\_{s} ( \\vert f(\\mathbf{X}, s) - f(\\mathbf{T}\_{s}(\\mathbf{X}), s') \\vert \\vert Y = 1 ),$
> where $w\_{y} := \\mathbb{P}(Y = y \\vert S = s), y \\in \\{ 0, 1\\}.$
> Note that by the definition of the joint OT map, we have for almost all $\\mathbf{x}$ with respect to the distribution of $\\mathbf{X} \\vert S = s,$
> $\\mathcal{P} (Y = y \\vert \\mathbf{X} = \\mathbf{x}, S = s) = \\mathcal{P}(Y = y \\vert \\mathbf{X} = \\mathbf{T}_{s}(\\mathbf{x}), S = s')$ for all $y \\in \\{0, 1\\}$ when $\\alpha \\rightarrow \\infty.$
>
>    Then, we have that  $C \\max\\{ \\Delta \\overline{\\textup{TPR}}(f), \\Delta \\overline{\\textup{FPR}}(f) \\}    \\le w\_{0} \Delta \\overline{\\textup{TPR}}(f) + w\_{1} \\Delta \\overline{\\textup{FPR}}(f)   \\le \\Delta \\textup{MDP}(f, \\mathbf{T}\_{s}),$   where $C = min\\{w_{0}, w_{1}\\}$ is a constant.  Here, $\\Delta \\overline{\\textup{TPR}}(f) = \vert \\mathbb{E}( f\_{0}(\\mathbf{X}) \\vert Y = 1, S = 0 ) - \\mathbb{E}( f\_{1}(\mathbf{X}) \vert Y = 1, S = 1 ) \vert$    and $\\Delta \\overline{\\textup{FPR}}(f) = \\vert \\mathbb{E}( f\_{0}(\\mathbf{X}) \\vert Y = 0, S = 0 ) - \\mathbb{E}( f\_{1}(\mathbf{X}) \\vert Y = 0, S = 1 ) \\vert$    are the smooth versions of $\\Delta {\\textup{TPR}}(f)$ and $\\Delta {\\textup{FPR}}(f),$ respectively.
>
>    Hence, we can conclude that using the joint OT map with large $\alpha$ can control EO.
>
> **Revision**:
>
> - **Section 4.2**: We add the theoretical relationship between FTM with the joint OT map and EO.
> - **Section 5.3.1**: We modify Table 3 by adding the values of $\\Delta \\textup{TPR}$ and $\\Delta \\textup{FPR}.$
>
> -------
> > **W8**: Final paragraph in sections 4.1. and 4.2. is exactly the same, so it could be expressed in a more concise way to ease the reading.
>
> **Answer**:
> As you note, the empirical FTM algorithm given a transport map is identical for the both cases. To avoid redundancy, we add a new subsection (i.e., Section 4.3) to present the final learning objective function, while removing the two redundant paragraphs.
>
> **Revision**:
> - **Section 4.3**: We add a new subsection presenting the empirical FTM algorithm given a pre-specified transport map.
>
> -------
>
> > **W9**: . Authors should clarify the order of the Wasserstein distance, since both 1 and 2 have been widely used in fair learning.
>
> **Answer**:
> The order of Wasserstein distance, used for $\\Delta \\textup{WDP}$ in this paper, is 1. That is, for given two probability distributions $\\mathcal{Q}\_{1}$ and $\\mathcal{Q}\_{2},$ the (1-)Wasserstein distance is defined as $\\mathcal{W}(\\mathcal{Q}\_{1}, \\mathcal{Q}\_{2}) := \inf\_{\\gamma \\in \\Gamma(\\mathcal{Q}\_{1}, \\mathcal{Q}\_{2})} \\mathbb{E}\_{(x, y) \\sim \\gamma} \\Vert x - y \\Vert\_{1},$ where $ \\Gamma(\\mathcal{Q}\_{1}, \\mathcal{Q}\_{2})$ is the set of joint probability distributions of marginals $ \\mathcal{Q}\_{1} $ and $\\mathcal{Q}\_{2}.$ As you suggest, we provide this detailed definition in Section 2.2.
>
> **Revision**:
>
> - **Section 2.2**: We add the concrete definition of 1-Wasserstein distance, which is used for $\\Delta \\textup{WDP}$ in this paper.

---

> ### Author Response · Authors · 2024-11-04
> **Author response #8**
>
> -------
>
> > **W10**:  In Section 5: work $\rightarrow$ works
>
> **Answer**:
>
> Thank you for the careful reading.  We modify it accordingly.
>
> **Revision**:
>
> - **Section 5**: We modify as: "... empirically work well to learn ..." $\\rightarrow$ "... empirically works well to learn ..."
>
> -------
>
> > **W11**: Notation: I do not find it appropriate to use the symbol Delta for the fairness measure.
>
> **Answer**:
>
> We use the notation $\\Delta$ to clearly distinguish between the concept and measure of group fairness.  For example, "DP" represents the concept of statistical parity between predictions for two protected groups, while "$\\Delta \\textup{DP}$" denotes a specific measure of DP.  Moreover, note that the notation $\\Delta$ (or 'D', representing the difference) is commonly used in many prior studies (Donini et al., 2018; Madras et al., 2018; Chuang \\& Mroueh, 2021; Kim et al., 2022).
>
> -------
>
> > **W12**: Proposition 3.1. is in fact a Corollary of Theorem 3.3. As the authors say in the proof, it is a particular case of the Theorem, for which Assumptions (C1) and (C2) are also needed.
>
> **Answer**:
> Thank you for careful reading.   Since Proposition 3.1 is a direct corollary of Theorem 3.3, it is necessary to state the condition (C2) before Proposition 3.1.  We move (C2) to just before Proposition 3.1.
> Note that the purpose of assuming (C2) is for easier discussion to guarantee the existence of a transport map between two probability distributions.
>
> Furthermore, in fact, (C1) is not needed because we have already assumed the bound of $f$. Hence, we remove (C1) and renamed (C2) as (C).
>
> **Revision**:
>
> - **Section 3.1**: We add this explanation just before Proposition 3.1.
>
> -------
> > **W13**: Title of 3.2. would be more informative with the full name of the algorithm.
>
>
> **Answer**:
>
> Thank you for the suggestion.   We modify the title of Section 3.2 as "FTM (Fairness Through Matching): learning a group-fair model with a given transport map", accordingly.
>
> **Revision**:
>
> - **Section 3.2**: We modify the title of this section from
>   "FTM: learning a group-fair model for a given transport map"
>   to
>   "Fairness Through Matching (FTM): learning a group-fair model with a given transport map".

---

> > ### Comment · Reviewer_H7Dq · 2024-11-21
> > **Response to authors**
> >
> > I would like to thank the authors for the detailed response to all the issues raised. I sincerely believe that the paper has gained value as a result of their careful review.

---

### Review · Reviewer_i5aw · 2024-10-23

**Summary Of Contributions:**

The authors proposed that all fairness models can be reformulated as transportation problems, and vice versa. This offers a new perspective on addressing group fairness. They further suggested achieving group fairness by solving the reformulated transportation problem. Experimental results on several widely used fairness research datasets demonstrate comparable performance to mainstream baseline methods.

**Audience:**

Yes

**Broader Impact Concerns:**

The authors addressed broader impact very well at the end of the paper. I don't have further concerns regarding this topic.

**Claims And Evidence:**

Yes

**Requested Changes:**

1. Is it reasonable to add accuracy numbers in Table 2?

2. Is it reasonable to show the improvement on Equalized Opportunity [1]?


[1] Equality of Opportunity in Supervised Learning

**Strengths And Weaknesses:**

Strengths:

1. The paper is well organized and easy to follow.

2. The new point of view to formulate fairness model using transportation problem is novel to the best of my knowledge.

3. Experimental results show it promising to improve fairness through solving the re-formulation into transportation problem.

Weakness:

1. The experiment results seem not consistently outperforming baseline methods, as shown in Figure 2.

---

> ### Author Response · Authors · 2024-11-04
> **Author response**
>
> ### *Dear reviewer i5aw*: We appreciate your review comments and are truly happy to have the opportunity to improve our work. We hope our point-by-point responses provided below, along with the revised paper, will resolve your concerns.
>
> -------
> -------
> ### Weaknesses
> -------
> > **W1**:
> The experiment results seem not consistently outperforming baseline methods, as shown in Figure 2.
>
> **Answer**:
> As you note, FTM with the marginal OT map does not consistently outperform the baselines. However, we observe that **FTM with the joint OT map of large $\alpha$ can yield performance comparable to the best trade-offs** (see Figure 5 in Section 5.3.1 of the revised paper).
>
> In summary, the benefit of FTM is that it allows practitioners to control specific micro-level fairness (e.g., subset fairness) based on their needs, without hampering prediction accuracy much, whereas existing algorithms focused solely on group fairness may lack this flexibility.
>
> **Revision**:
> - **Section 5.3.1**: We add Figure 5, which demonstrates that FTM with the joint OT map can achieve competitive fairness-prediction trade-offs compared to the best trade-offs.
>
> -------
> -------
> ### Requested changes
> -------
> > **RC1**:
> Is it reasonable to add accuracy numbers in Table 2?
>
> **Answer**:
> The purpose of presenting Table 2—now Table 3 in the revised paper—is to demonstrate the flexibility of FTM by appropriately choosing the transport map. For example, to improve prediction accuracy, we recommend using the joint OT map (introduced in Section 4.2) instead of the marginal OT map (introduced in Section 4.1), if the prediction accuracy of a group-fair model learned by FTM with the marginal OT map is suboptimal.
>
> However, as shown in Figure 6 in Section 5.3.2, FTM with the joint OT map has a higher transport cost than FTM with the marginal OT map, indicating a trade-off between transport cost and prediction accuracy.
>
> -------
> > **RC2**:
> Is it reasonable to show the improvement on Equalized Opportunity [1]?
> [1] Equality of Opportunity in Supervised Learning
>
> **Answer**:
> The reason why we provide the equalized odds (EO) results is to highlight a benefit of using the joint OT map. The joint OT map matches individuals based on both input features and output labels, increasing the chance of matching individuals with similar input features $\mathbf{X}$ and same labels $Y$. As a result, the level of equalized odds is expected to be improved with the joint OT map compared to the marginal OT map, which we empirically confirm in Table 3.
>
> Furthermore, we include the results for equal opportunity (i.e., $\Delta \textup{TPR}$) in Table 3 of the revised paper.
>
> **Revision**:
> - **Section 5.3.1**: We modify Table 3 by adding the values of $\Delta \textup{TPR}$ (and $\Delta \textup{FPR}$ also).

---

### Decision · Action_Editor_SjDj · 2024-12-20

**Recommendation:** Accept as is

**Comment:**

Three reviewers recommended acceptance.

The reviewers raised several concerns, including the need for more details about existing works that use the optimal transport map for algorithmic fairness, particularly the work of Gordaliza et al. (2019), which is notably similar in applying the optimal transport map to the input space. Additionally, they requested further experimental results comparing the proposed method with fair representation learning approaches, including Gordaliza et al. (2019). Improvements to the writing were also suggested to enhance readability, especially regarding the theoretical statements and proofs.

These concerns were sufficiently addressed during the rebuttal period and are reflected in the revised manuscript.

**Audience:**

Yes.

**Claims And Evidence:**

The paper examines the inherent properties of group-fair models by establishing a theoretical link between group-fair models and transport maps. This link explains how models minimizing statistical disparities (e.g., TVDP and WDP) achieve group fairness by identifying which individuals from different protected groups are treated similarly. While matching dissimilar individuals may cause subset unfairness, aligning similar ones reduces the risk of self-fulfilling prophecies. Subset unfairness is a common issue in group-fair models trained with current algorithms on real-world data.

The paper then proposes a method of learning group-fair models with a pre-specified transport map. The proposed method prioritizes mitigating subset unfairness over optimizing the fairness-accuracy trade-off by carefully selecting transport maps. Unlike existing algorithms that focus solely on group fairness, the proposed method offers flexibility for practitioners to address micro-level fairness (e.g., subset fairness) based on specific needs. Experimental results validate these claims.